# ACCELERATING DISTRIBUTED STOCHASTIC OPTIMIZATION VIA SELF-REPELLENT RANDOM WALKS

**Jie Hu**[*1]**, Vishwaraj Doshi**[*2]**, Do Young Eun**[1]
[1]North Carolina State University, [2]IQVIA Inc.
{jhu29,dyeun}@ncsu.edu, vishwaraj.doshi@iqvia.com

## ABSTRACT

We study a family of distributed stochastic optimization algorithms where gradients are sampled by a token traversing a network of agents in random-walk fashion. Typically, these random-walks are chosen to be Markov chains that asymptotically sample from a desired target distribution, and play a critical role in the convergence of the optimization iterates. In this paper, we take a novel approach by replacing the standard *linear* Markovian token by one which follows a *nonlinear* Markov chain - namely the Self-Repellent Radom Walk (SRRW). Defined for any given 'base' Markov chain, the SRRW, parameterized by a positive scalar $\alpha$, is less likely to transition to states that were highly visited in the past, thus the name. In the context of MCMC sampling on a graph, a recent breakthrough in Doshi et al. (2023) shows that the SRRW achieves $O(1/\alpha)$ decrease in the asymptotic variance for sampling. We propose the use of a 'generalized' version of the SRRW to drive token algorithms for distributed stochastic optimization in the form of stochastic approximation, termed SA-SRRW. We prove that the optimization iterate errors of the resulting SA-SRRW converge to zero almost surely and prove a central limit theorem, deriving the explicit form of the resulting asymptotic covariance matrix corresponding to iterate errors. This asymptotic covariance is always smaller than that of an algorithm driven by the base Markov chain and decreases at rate $O(1/\alpha^2)$ - the performance benefit of using SRRW thereby *amplified* in the stochastic optimization context. Empirical results support our theoretical findings.

## 1 INTRODUCTION

Stochastic optimization algorithms solve optimization problems of the form

$$\boldsymbol{\theta}^* \in \underset{\boldsymbol{\theta} \in \mathbb{R}^d}{\arg\min} f(\boldsymbol{\theta}), \quad \text{where } f(\boldsymbol{\theta}) \triangleq \mathbb{E}_{X \sim \boldsymbol{\mu}}[F(\boldsymbol{\theta}, X)] = \sum_{i \in \mathcal{N}} \mu_i F(\boldsymbol{\theta}, i), \tag{1}$$

with the objective function $f : \mathbb{R}^d \to \mathbb{R}$ and $X$ taking values in a finite state space $\mathcal{N}$ with distribution $\boldsymbol{\mu} \triangleq [\mu_i]_{i \in \mathcal{N}}$. Leveraging partial gradient information per iteration, these algorithms have been recognized for their scalability and efficiency with large datasets (Bottou et al., 2018; Even, 2023). For any given *noise sequence* $\{X_n\}_{n \geq 0} \subset \mathcal{N}$, and step size sequence $\{\beta_n\}_{n \geq 0} \subset \mathbb{R}_+$, most stochastic optimization algorithms can be classified as stochastic approximations (SA) of the form

$$\boldsymbol{\theta}_{n+1} = \boldsymbol{\theta}_n + \beta_{n+1} H(\boldsymbol{\theta}_n, X_{n+1}), \quad \forall\, n \geq 0, \tag{2}$$

where, roughly speaking, $H(\boldsymbol{\theta}, i)$ contains gradient information $\nabla_{\boldsymbol{\theta}} F(\theta, i)$, such that $\boldsymbol{\theta}^*$ solves $\mathbf{h}(\boldsymbol{\theta}) \triangleq \mathbb{E}_{X \sim \boldsymbol{\mu}}[H(\boldsymbol{\theta}, X)] = \sum_{i \in \mathcal{N}} \mu_i H(\boldsymbol{\theta}, i) = \mathbf{0}$. Such SA iterations include the well-known stochastic gradient descent (SGD), stochastic heavy ball (SHB) (Gadat et al., 2018; Li et al., 2022), and some SGD-type algorithms employing additional auxiliary variables (Barakat et al., 2021).[1] These algorithms typically have the stochastic noise term $X_n$ generated by *i.i.d.* random variables with probability distribution $\boldsymbol{\mu}$ in each iteration. In this paper, we study a stochastic optimization algorithm where the noise sequence governing access to the gradient information is generated from general stochastic processes in place of *i.i.d.* random variables.

---

[*]Equal contributors.
[1]Further illustrations of stochastic optimization algorithms of the form (2) are deferred to Appendix A.

This is commonly the case in distributed learning, where $\{X_n\}$ is a (typically Markovian) random walk, and should asymptotically be able to sample the gradients from the desired probability distribution $\boldsymbol{\mu}$. This is equivalent to saying that the random walker's *empirical distribution* converges to $\boldsymbol{\mu}$ almost surely (a.s.); that is, $\mathbf{x}_n \triangleq \frac{1}{n+1} \sum_{k=0}^{n} \boldsymbol{\delta}_{X_k} \xrightarrow[n \to \infty]{a.s.} \boldsymbol{\mu}$ for any initial $X_0 \in \mathcal{N}$, where $\boldsymbol{\delta}_{X_k}$ is the delta measure whose $X_k$'th entry is one, the rest being zero. Such convergence is most commonly achieved by employing the Metropolis Hastings random walk (MHRW) which can be designed to sample from any *target* measure $\boldsymbol{\mu}$ and implemented in a scalable manner (Sun et al., 2018). Unsurprisingly, convergence characteristics of the employed Markov chain affect that of the SA sequence (2), and appear in both finite-time and asymptotic analyses. Finite-time bounds typically involve the second largest eigenvalue in modulus (SLEM) of the Markov chain's transition kernel $\mathbf{P}$, which is critically connected to the mixing time of a Markov chain (Levin & Peres, 2017); whereas asymptotic results such as central limit theorems (CLT) involve asymptotic covariance matrices that embed information regarding the entire spectrum of $\mathbf{P}$, i.e., all eigenvalues as well as eigenvectors (Brémaud, 2013), which are key to understanding the sampling efficiency of a Markov chain. Thus, the choice of random walker can significantly impact the performance of (2), and simply ensuring that it samples from $\boldsymbol{\mu}$ asymptotically is not enough to achieve optimal algorithmic performance. In this paper, we take a closer look at the distributed stochastic optimization problem through the lens of a *non-linear* Markov chain, known as the *Self Repellent Random Walk* (SRRW), which was shown in Doshi et al. (2023) to achieve asymptotically minimal sampling variance for large values of $\alpha$, a positive scalar controlling the strength of the random walker's self-repellence behaviour. Our proposed modification of (2) can be implemented within the settings of decentralized learning applications in a scalable manner, while also enjoying significant performance benefit over distributed stochastic optimization algorithms driven by vanilla Markov chains.

**Token Algorithms for Decentralized Learning.** In decentralized learning, agents like smartphones or IoT devices, each containing a subset of data, collaboratively train models on a graph $\mathcal{G}(\mathcal{N}, \mathcal{E})$ by sharing information locally without a central server (McMahan et al., 2017). In this setup, $N = |\mathcal{N}|$ agents correspond to nodes $i \in \mathcal{N}$, and an edge $(i, j) \in \mathcal{E}$ indicates direct communication between agents $i$ and $j$. This decentralized approach offers several advantages compared to the traditional centralized learning setting, promoting data privacy and security by eliminating the need for raw data to be aggregated centrally and thus reducing the risk of data breach or misuse (Bottou et al., 2018; Nedic, 2020). Additionally, decentralized approaches are more scalable and can handle vast amounts of heterogeneous data from distributed agents without overwhelming a central server, alleviating concerns about single point of failure (Vogels et al., 2021).

Among decentralized learning approaches, the class of 'Token' algorithms can be expressed as stochastic approximation iterations of the type (2), wherein the sequence $\{X_n\}$ is realized as the sample path of a token that stochastically traverses the graph $\mathcal{G}$, carrying with it the iterate $\boldsymbol{\theta}_n$ for any time $n \geq 0$ and allowing each visited node (agent) to incrementally update $\boldsymbol{\theta}_n$ using locally available gradient information. Token algorithms have gained popularity in recent years (Hu et al., 2022; Triastcyn et al., 2022; Hendrikx, 2023), and are provably more communication efficient (Even, 2023) when compared to consensus-based algorithms - another popular approach for solving distributed optimization problems (Boyd et al., 2006; Morral et al., 2017; Olshevsky, 2022). The construction of token algorithms means that they do not suffer from expensive costs of synchronization and communication that are typical of consensus-based approaches, where all agents (or a subset of agents selected by a coordinator (Boyd et al., 2006; Wang et al., 2019)) on the graph are required to take simultaneous actions, such as communicating on the graph at each iteration. While decentralized Federated learning has indeed helped mitigate the communication overhead by processing multiple SGD iterations prior to each aggregation (Lalitha et al., 2018; Ye et al., 2022; Chellapandi et al., 2023), they still cannot overcome challenges such as synchronization and straggler issues.

**Self Repellent Random Walk.** As mentioned earlier, sample paths $\{X_n\}$ of token algorithms are usually generated using Markov chains with $\boldsymbol{\mu} \in \text{Int}(\Sigma)$ as their limiting distribution. Here, $\Sigma$ denotes the $N$-dimensional probability simplex, with $\text{Int}(\Sigma)$ representing its interior. A recent work by Doshi et al. (2023) pioneers the use of *non-linear Markov chains* to, in some sense, improve upon *any given* time-reversible Markov chain with transition kernel $\mathbf{P}$ whose stationary distribution is $\boldsymbol{\mu}$.

They show that the non-linear transition kernel[2] $\mathbf{K}[\cdot] : \text{Int}(\Sigma) \to [0, 1]^{N \times N}$, given by

$$K_{ij}[\mathbf{x}] \triangleq \frac{P_{ij}(x_j/\mu_j)^{-\alpha}}{\sum_{k \in \mathcal{N}} P_{ik}(x_k/\mu_k)^{-\alpha}}, \quad \forall\, i, j \in \mathcal{N}, \tag{3}$$

for any $\mathbf{x} \in \text{Int}(\Sigma)$, when simulated as a self-interacting random walk (Del Moral & Miclo, 2006; Del Moral & Doucet, 2010), can achieve smaller asymptotic variance than the base Markov chain when sampling over a graph $\mathcal{G}$, for all $\alpha > 0$. The argument $\mathbf{x}$ for the kernel $\mathbf{K}[\boldsymbol{x}]$ is taken to be the empirical distribution $\mathbf{x}_n$ at each time step $n \geq 0$. For instance, if node $j$ has been visited more often than other nodes so far, the entry $x_j$ becomes larger (than target value $\mu_j$), resulting in a smaller transition probability from $i$ to $j$ under $\mathbf{K}[\mathbf{x}]$ in (3) compared to $P_{ij}$. This ensures that a random walker prioritizes more seldom visited nodes in the process, and is thus 'self-repellent'. This effect is made more drastic by increasing $\alpha$, and leads to asymptotically near-zero variance at a rate of $O(1/\alpha)$. Moreover, the polynomial function $(x_i/\mu_i)^{-\alpha}$ chosen to encode self-repellent behaviour is shown in Doshi et al. (2023) to be the only one that allows the SRRW to inherit the so-called 'scale-invariance' property of the underlying Markov chain – a necessary component for the scalable implementation of a random walker over a large network without requiring knowledge of any graph-related global constants. Conclusively, such attributes render SRRW especially suitable for distributed optimization.[3]

**Effect of Stochastic Noise - Finite time and Asymptotic Approaches.** Most contemporary token algorithms driven by Markov chains are analyzed using the finite-time bounds approach for obtaining insights into their convergence rates (Sun et al., 2018; Doan et al., 2019; 2020; Triastcyn et al., 2022; Hendrikx, 2023). However, as also explained in Even (2023), in most cases these bounds are overly dependent on mixing time properties of the specific Markov chain employed therein. This makes them largely ineffective in capturing the exact contribution of the underlying random walk in a manner which is qualitative enough to be used for algorithm design; and performance enhancements are typically achieved via application of techniques such as variance reduction (Defazio et al., 2014; Schmidt et al., 2017), momentum/Nesterov's acceleration (Gadat et al., 2018; Li et al., 2022), adaptive step size (Kingma & Ba, 2015; Reddi et al., 2018), which work by modifying the algorithm iterations themselves, and never consider potential improvements to the stochastic input itself.

Complement to finite-time approaches, asymptotic analysis using CLT has proven to be an excellent tool to approach the design of stochastic algorithms (Hu et al., 2022; Devraj & Meyn, 2017; Morral et al., 2017; Chen et al., 2020a; Mou et al., 2020; Devraj & Meyn, 2021). Hu et al. (2022) shows how asymptotic analysis can be used to compare the performance of SGD algorithms for various stochastic inputs using their notion of efficiency ordering, and, as mentioned in Devraj & Meyn (2017), the asymptotic benefits from minimizing the limiting covariance matrix are known to be a good predictor of finite-time algorithmic performance, also observed empirically in Section 4.

From the perspective of both finite-time analysis as well as asymptotic analysis of token algorithms, it is now well established that employing 'better' Markov chains can enhance the performance of stochastic optimization algorithm. For instance, Markov chains with smaller SLEMs yield tighter finite-time upper bounds (Sun et al., 2018; Ayache & El Rouayheb, 2021; Even, 2023). Similarly, Markov chains with smaller asymptotic variance for MCMC sampling tasks also provide better performance, resulting in smaller covariance matrix of SGD algorithms (Hu et al., 2022). Therefore, with these breakthrough results via SRRW achieving near-zero sampling variance, it is within reason to ask: *Can we achieve near-zero variance in distributed stochastic optimization driven by SRRW-like token algorithms on any general graph?*[4] In this paper, we answer in the affirmative.

**SRRW Driven Algorithm and Analysis Approach.** For any ergodic time-reversible Markov chain with transition probability matrix $\mathbf{P} \triangleq [P_{ij}]_{i,j \in \mathcal{N}}$ and stationary distribution $\boldsymbol{\mu} \in \text{Int}(\Sigma)$, we consider a general step size version of the SRRW stochastic process analysed in Doshi et al. (2023) and

---

[2]Here, non-linearity in the transition kernel implies that $\mathbf{K}[\mathbf{x}]$ takes probability distribution $\mathbf{x}$ as the argument (Andrieu et al., 2007), as opposed to the kernel being a linear operator $\mathbf{K}[\mathbf{x}] = \mathbf{P}$ for a constant stochastic matrix $\mathbf{P}$ in a standard (linear) Markovian setting.

[3]Recently, Guo et al. (2020) introduce an optimization scheme, which designs self-repellence into the perturbation of the gradient descent iterates (Jin et al., 2017; 2018; 2021) with the goal of escaping saddle points. This notion of self-repellence is distinct from the SRRW, which is a probability kernel designed specifically for a token to sample from a target distribution $\boldsymbol{\mu}$ over a set of nodes on an arbitrary graph.

[4]This near-zero sampling variance implies a significantly smaller variance than even an *i.i.d.* sampling counterpart, while adhering to graph topological constraints of token algorithms.

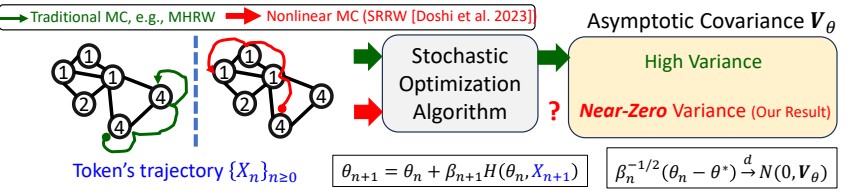

Figure 1: Visualization of token algorithms using SRRW versus traditional MC in distributed learning. Our CLT analysis, extended from SRRW itself to distributed stochastic approximation, leads to near-zero variance for the SA iteration $\boldsymbol{\theta}_n$. Node numbers on the left denote visit counts.

use it to drive the noise sequence in (2). Our SA-SRRW algorithm is as follows:

$$\text{Draw:} \qquad X_{n+1} \sim \mathbf{K}_{X_n,\cdot}[\mathbf{x}_n] \tag{4a}$$

$$\text{Update:} \qquad \mathbf{x}_{n+1} = \mathbf{x}_n + \gamma_{n+1}(\boldsymbol{\delta}_{X_{n+1}} - \mathbf{x}_n), \tag{4b}$$

$$\boldsymbol{\theta}_{n+1} = \boldsymbol{\theta}_n + \beta_{n+1} H(\boldsymbol{\theta}_n, X_{n+1}), \tag{4c}$$

where $\{\beta_n\}$ and $\{\gamma_n\}$ are step size sequences decreasing to zero, and $\mathbf{K}[\mathbf{x}]$ is the SRRW kernel in (3). Current non-asymptotic analyses require globally Lipschitz mean field function (Chen et al., 2020b; Doan, 2021; Zeng et al., 2021; Even, 2023) and is thus inapplicable to SA-SRRW since the mean field function of the SRRW iterates (4b) is only locally Lipschitz (details deferred to Appendix B). Instead, we successfully obtain non-trivial results by taking an asymptotic CLT-based approach to analyze (4). This goes beyond just analyzing the asymptotic *sampling covariance*[5] as in Doshi et al. (2023), the result therein forming a special case of ours by setting $\gamma_n = 1/(n+1)$ and considering only (4a) and (4b), that is, in the absence of optimization iteration (4c). Specifically, we capture the effect of SRRW's hyper-parameter $\alpha$ on the asymptotic speed of convergence of the *optimization* error term $\boldsymbol{\theta}_n - \boldsymbol{\theta}^*$ to zero via explicit deduction of its asymptotic covariance matrix. See Figure 1 for illustration.

**Our Contributions.**

**1.** Given any time-reversible 'base' Markov chain with transition kernel $\mathbf{P}$ and stationary distribution $\boldsymbol{\mu}$, we generalize first and second order convergence results of $\mathbf{x}_n$ to target measure $\boldsymbol{\mu}$ (Theorems 4.1 and 4.2 in Doshi et al., 2023) to a class of weighted empirical measures, through the use of more general step sizes $\gamma_n$. This includes showing that the asymptotic *sampling* covariance terms decrease to zero at rate $O(1/\alpha)$, thus quantifying the effect of self-repellent on $\mathbf{x}_n$. Our generalization is not for the sake thereof and is shown in Section 3 to be crucial for the design of step sizes $\beta_n, \gamma_n$.

**2.** Building upon the convergence results for iterates $\mathbf{x}_n$, we analyze the algorithm (4) driven by the SRRW kernel in (3) with step sizes $\beta_n$ and $\gamma_n$ separated into three disjoint cases:

(i) $\beta_n = o(\gamma_n)$, and we say that $\boldsymbol{\theta}_n$ is on the *slower* timescale compared to $\mathbf{x}_n$;
(ii) $\beta_n = \gamma_n$, and we say that $\boldsymbol{\theta}_n$ and $\mathbf{x}_n$ are on the same timescale;
(iii) $\gamma_n = o(\beta_n)$, and we say that $\boldsymbol{\theta}_n$ is on the *faster* timescale compared to $\mathbf{x}_n$.

For any $\alpha \geq 0$ and let $k = 1, 2$ and $3$ refer to the corresponding cases (i), (ii) and (iii), we show that

$$\boldsymbol{\theta}_n \xrightarrow[n\to\infty]{a.s.} \boldsymbol{\theta}^* \quad \text{and} \quad (\boldsymbol{\theta}_n - \boldsymbol{\theta}^*)/\sqrt{\beta_n} \xrightarrow[n\to\infty]{dist.} N\left(\mathbf{0}, \mathbf{V}_{\boldsymbol{\theta}}^{(k)}(\alpha)\right),$$

featuring distinct asymptotic covariance matrices $\mathbf{V}_{\boldsymbol{\theta}}^{(1)}(\alpha), \mathbf{V}_{\boldsymbol{\theta}}^{(2)}(\alpha)$ and $\mathbf{V}_{\boldsymbol{\theta}}^{(3)}(\alpha)$, respectively. The three matrices coincide when $\alpha = 0$,[6]. Moreover, the derivation of the CLT for cases (i) and (iii), for which (4) corresponds to *two-timescale* SA with *controlled* Markov noise, is the first of its kind and thus a key technical contribution in this paper, as expanded upon in Section 3.

**3.** For case (i), we show that $\mathbf{V}_{\boldsymbol{\theta}}^{(1)}(\alpha)$ decreases to zero (in the sense of Loewner ordering introduced in Section 2.1) as $\alpha$ increases, with rate $O(1/\alpha^2)$. This is especially surprising, since the asymptotic performance benefit from using the SRRW kernel with $\alpha$ in (3), to drive the noise terms $X_n$, is *amplified* in the context of distributed learning and estimating $\boldsymbol{\theta}^*$; compared to the sampling case, for which the rate is $O(1/\alpha)$ as mentioned earlier. For case (iii), we show that $\mathbf{V}_{\boldsymbol{\theta}}^{(3)}(\alpha) = \mathbf{V}_{\boldsymbol{\theta}}^{(3)}(0)$ for all $\alpha \geq 0$, implying that using the SRRW in this case provides no asymptotic benefit than the

---

[5]Sampling covariance corresponds to only the empirical distribution $\mathbf{x}_n$ in (4b).
[6]The $\alpha = 0$ case is equivalent to simply running the base Markov chain, since from (3) we have $\mathbf{K}[\cdot] = \mathbf{P}$, thus bypassing the SRRW's effect and rendering all three cases nearly the same.

original base Markov chain, and thus performs worse than case (i). In summary, we deduce that $\mathbf{V}_{\boldsymbol{\theta}}^{(1)}(\alpha_2) <_L \mathbf{V}_{\boldsymbol{\theta}}^{(1)}(\alpha_1) <_L \mathbf{V}_{\boldsymbol{\theta}}^{(1)}(0) = \mathbf{V}_{\boldsymbol{\theta}}^{(3)}(0) = \mathbf{V}_{\boldsymbol{\theta}}^{(3)}(\alpha)$ for all $\alpha_2 > \alpha_1 > 0$ and $\alpha > 0$.[7]

**4.** We numerically simulate our SA-SRRW algorithm on various real-world datasets, focusing on a binary classification task, to evaluate its performance across all three cases. By carefully choosing the function $H$ in SA-SRRW, we test the SGD and algorithms driven by SRRW. Our findings consistently highlight the superiority of case (i) over cases (ii) and (iii) for diverse $\alpha$ values, even in their finite time performance. Notably, our tests validate the variance reduction at a rate of $O(1/\alpha^2)$ for case (i), suggesting it as the best algorithmic choice among the three cases.

## 2 PRELIMINARIES AND MODEL SETUP

In Section 2.1, we first standardize the notations used throughout the paper, and define key mathematical terms and quantities used in our theoretical analyses. Then, in Section 2.2, we consolidate the model assumptions of our SA-SRRW algorithm (4). We then go on to discuss our assumptions, and provide additional interpretations of our use of generalized step-sizes.

### 2.1 BASIC NOTATIONS AND DEFINITIONS

Vectors are denoted by lower-case bold letters, e.g., $\mathbf{v} \triangleq [v_i] \in \mathbb{R}^D$, and matrices by upper-case bold, e.g., $\mathbf{M} \triangleq [M_{ij}] \in \mathbb{R}^{D \times D}$. $\mathbf{M}^{-T}$ is the transpose of the matrix inverse $\mathbf{M}^{-1}$. The diagonal matrix $\mathbf{D_v}$ is formed by vector $\mathbf{v}$ with $v_i$ as the $i$'th diagonal entry. Let $\mathbf{1}$ and $\mathbf{0}$ denote vectors of all ones and zeros, respectively. The identity matrix is represented by $\mathbf{I}$, with subscripts indicating dimensions as needed. A matrix is *Hurwitz* if all its eigenvalues possess strictly negative real parts. $\mathbb{1}_{\{\cdot\}}$ denotes an indicator function with condition in parentheses. We use $\|\cdot\|$ to denote both the Euclidean norm of vectors and the spectral norm of matrices. Two symmetric matrices $\mathbf{M}_1, \mathbf{M}_2$ follow Loewner ordering $\mathbf{M}_1 <_L \mathbf{M}_2$ if $\mathbf{M}_2 - \mathbf{M}_1$ is positive semi-definite and $\mathbf{M}_1 \neq \mathbf{M}_2$. This slightly differs from the conventional definition with $\leq_L$, which allows $\mathbf{M}_1 = \mathbf{M}_2$.

Throughout the paper, the matrix $\mathbf{P} \triangleq [P_{i,j}]_{i,j \in \mathcal{N}}$ and vector $\boldsymbol{\mu} \triangleq [\mu_i]_{i \in \mathcal{N}}$ are used exclusively to denote an $N \times N$-dimensional transition kernel of an ergodic Markov chain, and its stationary distribution, respectively. Without loss of generality, we assume $P_{ij} > 0$ if and only if $a_{ij} > 0$. Markov chains satisfying the *detailed balance equation*, where $\mu_i P_{ij} = \mu_j P_{ji}$ for all $i, j \in \mathcal{N}$, are termed *time-reversible*. For such chains, we use $(\lambda_i, \mathbf{u}_i)$ (resp. $(\lambda_i, \mathbf{v}_i)$) to denote the $i$'th left (resp. right) eigenpair where the eigenvalues are ordered: $-1 < \lambda_1 \leq \cdots \leq \lambda_{N-1} < \lambda_N = 1$, with $\mathbf{u}_N = \boldsymbol{\mu}$ and $\mathbf{v}_N = \mathbf{1}$ in $\mathbb{R}^N$. We assume eigenvectors to be normalized such that $\mathbf{u}_i^T \mathbf{v}_i = 1$ for all $i$, and we have $\mathbf{u}_i = \mathbf{D}_{\boldsymbol{\mu}} \mathbf{v}_i$ and $\mathbf{u}_i^T \mathbf{v}_j = 0$ for all $i, j \in \mathcal{N}$. We direct the reader to Aldous & Fill (2002, Chapter 3.4) for a detailed exposition on spectral properties of time-reversible Markov chains.

### 2.2 SA-SRRW: KEY ASSUMPTIONS AND DISCUSSIONS

**Assumptions:** All results in our paper are proved under the following assumptions.

(A1) The function $H : \mathbb{R}^D \times \mathcal{N} \to \mathbb{R}^D$, is a continuous at every $\boldsymbol{\theta} \in \mathbb{R}^D$, and there exists a positive constant $L$ such that $\|H(\boldsymbol{\theta}, i)\| \leq L(1 + \|\boldsymbol{\theta}\|)$ for every $\boldsymbol{\theta} \in \mathbb{R}^D, i \in \mathcal{N}$.

(A2) Step sizes $\beta_n$ and $\gamma_n$ follow $\beta_n = (n+1)^{-b}$, and $\gamma_n = (n+1)^{-a}$, where $a, b \in (0.5, 1]$.

(A3) Roots of function $\mathbf{h}(\cdot)$ are disjoint, which comprise the globally attracting set $\Theta \triangleq \left\{ \boldsymbol{\theta}^* | \mathbf{h}(\boldsymbol{\theta}^*) = \mathbf{0}, \nabla \mathbf{h}(\boldsymbol{\theta}^*) + \frac{\mathbb{1}_{\{b=1\}}}{2} \mathbf{I} \text{ is Hurwitz} \right\} \neq \emptyset$ of the associated ordinary differential equation (ODE) for iteration (4c), given by $d\boldsymbol{\theta}(t)/dt = \mathbf{h}(\boldsymbol{\theta}(t))$.

(A4) For any $(\boldsymbol{\theta}_0, \mathbf{x}_0, X_0) \in \mathbb{R}^D \times \text{Int}(\Sigma) \times \mathcal{N}$, the iterate sequence $\{\boldsymbol{\theta}_n\}_{n \geq 0}$ (resp. $\{\mathbf{x}_n\}_{n \geq 0}$) is $\mathbb{P}_{\boldsymbol{\theta}_0, \mathbf{x}_0, X_0}$-almost surely contained within a compact subset of $\mathbb{R}^D$ (resp. $\text{Int}(\Sigma)$).

**Discussions on Assumptions:** Assumption A1 requires $H$ to only be locally Lipschitz albeit with linear growth, and is less stringent than the globally Lipschitz assumption prevalent in optimization literature (Li & Wai, 2022; Hendrikx, 2023; Even, 2023).

---

[7]In particular, this is the reason why we advocate for a more general step size $\gamma_n = (n+1)^{-a}$ in the SRRW iterates with $a < 1$, allowing us to choose $\beta_n = (n+1)^{-b}$ with $b \in (a, 1]$ to satisfy $\beta_n = o(\gamma_n)$ for case (i).

Assumption A2 is the general umbrella assumption under which cases (i), (ii) and (iii) mentioned in Section 1 are extracted by setting: (i) $a < b$, (ii) $a = b$, and (iii) $a > b$. Cases (i) and (iii) render $\boldsymbol{\theta}_n, \mathbf{x}_n$ on different timescales; the polynomial form of $\beta_n, \gamma_n$ widely assumed in the two-timescale SA literature (Mokkadem & Pelletier, 2006; Zeng et al., 2021; Hong et al., 2023). Case (ii) characterizes the SA-SRRW algorithm (4) as a single-timescale SA with polynomially decreasing step size, and is among the most common assumptions in the SA literature (Borkar, 2022; Fort, 2015; Li et al., 2023). In all three cases, the form of $\gamma_n$ ensures $\gamma_n \le 1$ such that the SRRW iterates $\mathbf{x}_n$ in (4b) is within $\mathrm{Int}(\Sigma)$, ensuring that $\mathbf{K}[\mathbf{x}_n]$ is well-defined for all $n \ge 0$.

In Assumption A3, limiting dynamics of SA iterations $\{\boldsymbol{\theta}_n\}_{n\ge 0}$ closely follow trajectories $\{\boldsymbol{\theta}(t)\}_{t\ge 0}$ of their associated ODE, and assuming the existence of globally stable equilibria is standard (Borkar, 2022; Fort, 2015; Li et al., 2023). In optimization problems, this is equivalent to assuming the existence of at most countably many local minima.

Assumption A4 assumes almost sure boundedness of iterates $\boldsymbol{\theta}_n$ and $\mathbf{x}_n$, which is a common assumption in SA algorithms (Kushner & Yin, 2003; Chen, 2006; Borkar, 2022; Karmakar & Bhatnagar, 2018; Li et al., 2023) for the stability of the SA iterations by ensuring the well-definiteness of all quantities involved. Stability of the weighted empirical measure $\mathbf{x}_n$ of the SRRW process is practically ensured by studying (4b) with a truncation-based procedure (see Doshi et al., 2023, Remark 4.5 and Appendix E for a comprehensive explanation), while that for $\boldsymbol{\theta}_n$ is usually ensured either as a by-product of the algorithm design, or via mechanisms such as projections onto a compact subset of $\mathbb{R}^D$, depending on the application context. We now provide additional discussions regarding the step-size assumptions and their implications on the SRRW iteration (4b).

**SRRW with General Step Size:** As shown in Benaim & Cloez (2015, Remark 1.1), albeit for a completely different non-linear Markov kernel driving the algorithm therein, iterates $\mathbf{x}_n$ of (4b) can also be expressed as *weighted* empirical measures of $\{X_n\}_{n\ge 0}$, in the following form:

$$\mathbf{x}_n = \frac{\sum_{i=1}^n \omega_i \boldsymbol{\delta}_{X_i} + \omega_0 \mathbf{x}_0}{\sum_{i=0}^n \omega_i}, \quad \text{where } \omega_0 = 1, \text{ and } \omega_n = \frac{\gamma_n}{\prod_{i=1}^n (1 - \gamma_i)}, \tag{5}$$

for all $n > 0$. For the special case when $\gamma_n = 1/(n+1)$ as in Doshi et al. (2023), we have $\omega_n = 1$ for all $n \ge 0$ and $\mathbf{x}_n$ is the typical, unweighted empirical measure. For the additional case considered in our paper, when $a < 1$ for $\gamma_n$ as in assumption A2, we can approximate $1 - \gamma_n \approx e^{-\gamma_n}$ and $\omega_n \approx n^{-a} e^{n^{(1-a)}/(1-a)}$. This implies that $\omega_n$ will increase at sub-exponential rate, giving more weight to recent visit counts and allowing it to quickly 'forget' the poor initial measure $\mathbf{x}_0$ and shed the correlation with the initial choice of $X_0$. This 'speed up' effect by setting $a < 1$ is guaranteed in case (i) irrespective of the choice of $b$ in Assumption A2, and in Section 3 we show how this can lead to further reduction in covariance of optimization error $\boldsymbol{\theta}_n = \boldsymbol{\theta}^*$ in the asymptotic regime.

**Additional assumption for case (iii):** Before moving on to Section 3, we take another look at the case when $\gamma_n = o(\beta_n)$, and replace A3 with the following, stronger assumption only for case (iii).

(A3′) For any $\mathbf{x} \in \mathrm{Int}(\Sigma)$, there exists a function $\rho : \mathrm{Int}(\Sigma) \to \mathbb{R}^D$ such that $\|\rho(\mathbf{x})\| \le L_2(1 + \|\mathbf{x}\|)$ for some $L_2 > 0$, $\mathbb{E}_{i\sim\boldsymbol{\pi}[\mathbf{x}]}[H(\rho(\mathbf{x}), i)] = 0$ and $\mathbb{E}_{i\sim\boldsymbol{\pi}[\mathbf{x}]}[\nabla H(\rho(\mathbf{x}), i)] + \frac{\mathbb{1}_{\{b=1\}}}{2}\mathbf{I}$ is Hurwitz.

While Assumption A3′ for case (iii) is much stronger than A3, it is not detrimental to the overall results of our paper, since case (i) is of far greater interest as impressed upon in Section 1. This is discussed further in Appendix C.

## 3   ASYMPTOTIC ANALYSIS OF THE SA-SRRW ALGORITHM

In this section, we provide the main results for the SA-SRRW algorithm (4). We first present the a.s. convergence and the CLT result for SRRW with generalized step size, extending the results in Doshi et al. (2023). Building upon this, we present the a.s. convergence and the CLT result for the SA iterate $\boldsymbol{\theta}_n$ under different settings of step sizes. We then shift our focus to the analysis of the different asymptotic covariance matrices emerging out of the CLT result, and capture the effect of $\alpha$ and the step sizes, particularly in cases (i) and (iii), on $\boldsymbol{\theta}_n - \boldsymbol{\theta}^*$ via performance ordering.

**Almost Sure convergence and CLT:** The following result establishes first and second order convergence of the sequence $\{\mathbf{x}_n\}_{n\ge 0}$, which represents the weighted empirical measures of the SRRW process $\{X_n\}_{n\ge 0}$, based on the update rule in (4b).

**Lemma 3.1.** *Under Assumptions A1, A2 and A4, for the SRRW iterates* (4b), *we have*

$$\mathbf{x}_n \xrightarrow[n\to\infty]{a.s.} \boldsymbol{\mu}, \quad and \quad \gamma_n^{-1/2}(\mathbf{x}_n - \boldsymbol{\mu}) \xrightarrow[n\to\infty]{dist.} N(\mathbf{0}, \mathbf{V}_{\mathbf{x}}(\alpha)),$$

$$where \quad \mathbf{V}_{\mathbf{x}}(\alpha) = \sum_{i=1}^{N-1} \frac{1}{2\alpha(1+\lambda_i)+2-\mathbb{1}_{\{a=1\}}} \cdot \frac{1+\lambda_i}{1-\lambda_i} \mathbf{u}_i \mathbf{u}_i^T. \tag{6}$$

*Moreover, for all $\alpha_2 > \alpha_1 > 0$, we have $\mathbf{V}_{\mathbf{x}}(\alpha_2) <_L \mathbf{V}_{\mathbf{x}}(\alpha_1) <_L \mathbf{V}_{\mathbf{x}}(0)$.*

Lemma 3.1 shows that the SRRW iterates $\mathbf{x}_n$ converges to the target distribution $\boldsymbol{\mu}$ a.s. even under the general step size $\gamma_n = (n+1)^{-a}$ for $a \in (0.5, 1]$. We also observe that the asymptotic covariance matrix $\mathbf{V}_{\mathbf{x}}(\alpha)$ decreases at rate $O(1/\alpha)$. Lemma 3.1 aligns with Doshi et al. (2023, Theorem 4.2 and Corollary 4.3) for the special case of $a = 1$, and is therefore more general. Critically, it helps us establish our next result regarding the first-order convergence for the optimization iterate sequence $\{\boldsymbol{\theta}_n\}_{n\geq 0}$ following update rule (4c), as well as its second-order convergence result, which follows shortly after. The proofs of Lemma 3.1 and our next result, Theorem 3.2, are deferred to Appendix D. In what follows, $k = 1, 2$, and 3 refer to cases (i), (ii), and (iii) in Section 2.2, respectively. All subsequent results are proven under Assumptions A1 to A4, with A3′ replacing A3 only when the step sizes $\beta_n, \gamma_n$ satisfy case (iii).

**Theorem 3.2.** *For $k \in \{1, 2, 3\}$, and any initial $(\boldsymbol{\theta}_0, \mathbf{x}_0, X_0) \in \mathbb{R}^D \times Int(\Sigma) \times \mathcal{N}$, we have $\boldsymbol{\theta}_n \to \boldsymbol{\theta}^*$ as $n \to \infty$ for some $\boldsymbol{\theta}^* \in \Theta$, $\mathbb{P}_{\boldsymbol{\theta}_0, \mathbf{x}_0, X_0}$-almost surely.*

In the stochastic optimization context, the above result ensures convergence of iterates $\boldsymbol{\theta}_n$ to a local minimizer $\boldsymbol{\theta}^*$. Loosely speaking, the first-order convergence of $\mathbf{x}_n$ in Lemma 3.1 as well as that of $\boldsymbol{\theta}_n$ are closely related to the convergence of trajectories $\{\mathbf{z}(t) \triangleq (\boldsymbol{\theta}(t), \mathbf{x}(t))\}_{t\geq 0}$ of the (coupled) mean-field ODE, written in a matrix-vector form as

$$\frac{d}{dt}\mathbf{z}(t) = \mathbf{g}(\mathbf{z}(t)) \triangleq \begin{bmatrix} \mathbf{H}(\boldsymbol{\theta}(t))^T \boldsymbol{\pi}[\mathbf{x}(t)] \\ \boldsymbol{\pi}[\mathbf{x}(t)] - \mathbf{x}(t) \end{bmatrix} \in \mathbb{R}^{D+N}. \tag{7}$$

where matrix $\mathbf{H}(\boldsymbol{\theta}) \triangleq [H(\boldsymbol{\theta}, 1), \cdots, H(\boldsymbol{\theta}, N)]^T \in \mathbb{R}^{N\times D}$ for any $\boldsymbol{\theta} \in \mathbb{R}^D$. Here, $\boldsymbol{\pi}[\mathbf{x}] \in Int(\Sigma)$ is the stationary distribution of the SRRW kernel $\mathbf{K}[\mathbf{x}]$ and is shown in Doshi et al. (2023) to be given by $\pi_i[\mathbf{x}] \propto \sum_{j\in\mathcal{N}} \mu_i P_{ij}(x_i/\mu_i)^{-\alpha}(x_j/\mu_j)^{-\alpha}$. The Jacobian matrix of (7) when evaluated at equilibria $\mathbf{z}^* = (\boldsymbol{\theta}^*, \boldsymbol{\mu})$ for $\boldsymbol{\theta}^* \in \Theta$ captures the behaviour of solutions of the mean-field in their vicinity, and plays an important role in the asymptotic covariance matrices arising out of our CLT results. We evaluate this Jacobian matrix $\mathbf{J}(\alpha)$ as a function of $\alpha \geq 0$ to be given by

$$\mathbf{J}(\alpha) \triangleq \nabla g(\mathbf{z}^*) = \begin{bmatrix} \nabla\mathbf{h}(\boldsymbol{\theta}^*) & -\alpha\mathbf{H}(\boldsymbol{\theta}^*)^T(\mathbf{P}^T+\mathbf{I}) \\ \mathbf{0}_{N\times D} & 2\alpha\boldsymbol{\mu}\mathbf{1}^T - \alpha\mathbf{P}^T - (\alpha+1)\mathbf{I} \end{bmatrix} \triangleq \begin{bmatrix} \mathbf{J}_{11} & \mathbf{J}_{12}(\alpha) \\ \mathbf{J}_{21} & \mathbf{J}_{22}(\alpha) \end{bmatrix}. \tag{8}$$

The derivation of $\mathbf{J}(\alpha)$ is referred to Appendix E.1.[8] Here, $\mathbf{J}_{21}$ is a zero matrix since $\boldsymbol{\pi}[\mathbf{x}] - \mathbf{x}$ is devoid of $\boldsymbol{\theta}$. While matrix $\mathbf{J}_{22}(\alpha)$ is exactly of the form in Doshi et al. (2023, Lemma 3.4) to characterize the SRRW performance, our analysis includes an additional matrix $\mathbf{J}_{12}(\alpha)$, which captures the effect of $\mathbf{x}(t)$ on $\boldsymbol{\theta}(t)$ in the ODE (7), which translates to the influence of our generalized SRRW empirical measure $\mathbf{x}_n$ on the SA iterates $\boldsymbol{\theta}_n$ in (4).

For notational simplicity, and without loss of generality, all our remaining results are stated while conditioning on the event that $\{\boldsymbol{\theta}_n \to \boldsymbol{\theta}^*\}$, for some $\boldsymbol{\theta}^* \in \Theta$. We also adopt the shorthand notation $\mathbf{H}$ to represent $\mathbf{H}(\boldsymbol{\theta}^*)$. Our main CLT result is as follows, with its proof deferred to Appendix E.

**Theorem 3.3.** *For any $\alpha \geq 0$, we have: (a) There exists $\mathbf{V}^{(k)}(\alpha)$ for all $k \in \{1, 2, 3\}$ such that*

$$\begin{bmatrix} \beta_n^{-1/2}(\boldsymbol{\theta}_n - \boldsymbol{\theta}^*) \\ \gamma_n^{-1/2}(\mathbf{x}_n - \boldsymbol{\mu}) \end{bmatrix} \xrightarrow[n\to\infty]{dist.} N\left(\mathbf{0}, \mathbf{V}^{(k)}(\alpha)\right).$$

*(b) For $k = 2$, matrix $\mathbf{V}^{(2)}(\alpha)$ solves the Lyapunov equation $\mathbf{J}(\alpha)\mathbf{V}^{(2)}(\alpha) + \mathbf{V}^{(2)}(\alpha)\mathbf{J}(\alpha)^T + \mathbb{1}_{\{b=1\}}\mathbf{V}^{(2)}(\alpha) = -\mathbf{U}$, where the Jacobian matrix $\mathbf{J}(\alpha)$ is in (8), and*

$$\mathbf{U} \triangleq \sum_{i=1}^{N-1} \frac{1+\lambda_i}{1-\lambda_i} \cdot \begin{bmatrix} \mathbf{H}^T\mathbf{u}_i\mathbf{u}_i^T\mathbf{H} & \mathbf{H}^T\mathbf{u}_i\mathbf{u}_i^T \\ \mathbf{u}_i\mathbf{u}_i^T\mathbf{H} & \mathbf{u}_i\mathbf{u}_i^T \end{bmatrix} \triangleq \begin{bmatrix} \mathbf{U}_{11} & \mathbf{U}_{12} \\ \mathbf{U}_{21} & \mathbf{U}_{22} \end{bmatrix}. \tag{9}$$

*(c) For $k \in \{1, 3\}$, $\mathbf{V}^{(k)}(\alpha)$ becomes block diagonal, which is given by*

$$\mathbf{V}^{(k)}(\alpha) = \begin{bmatrix} \mathbf{V}_{\boldsymbol{\theta}}^{(k)}(\alpha) & \mathbf{0}_{D\times N} \\ \mathbf{0}_{N\times D} & \mathbf{V}_{\mathbf{x}}(\alpha) \end{bmatrix}, \tag{10}$$

---

[8]The Jacobian $\mathbf{J}(\alpha)$ is $(D+N)\times(D+N)$−dimensional, with $\mathbf{J}_{11} \in \mathbb{R}^{D\times D}$ and $\mathbf{J}_{22}(\alpha) \in \mathbb{R}^{N\times N}$. Following this, all matrices written in a block form, such as matrix $\mathbf{U}$ in (9), will inherit the same dimensional structure.

*where $\mathbf{V_x}(\alpha)$ is as in (6), and $\mathbf{V}_{\boldsymbol{\theta}}^{(1)}(\alpha)$ and $\mathbf{V}_{\boldsymbol{\theta}}^{(3)}(\alpha)$ can be written in the following explicit form:*

$$\mathbf{V}_{\boldsymbol{\theta}}^{(1)}(\alpha) = \int_0^{\infty} e^{t(\nabla_{\boldsymbol{\theta}}\mathbf{h}(\boldsymbol{\theta}^*) + \frac{\mathbb{1}_{\{b=1\}}}{2}\boldsymbol{I})}\mathbf{U}_{\boldsymbol{\theta}}(\alpha)e^{t(\nabla_{\boldsymbol{\theta}}\mathbf{h}(\boldsymbol{\theta}^*) + \frac{\mathbb{1}_{\{b=1\}}}{2}\boldsymbol{I})^T}dt,$$

$$\mathbf{V}_{\boldsymbol{\theta}}^{(3)}(\alpha) = \int_0^{\infty} e^{t\nabla_{\boldsymbol{\theta}}\mathbf{h}(\boldsymbol{\theta}^*)}\mathbf{U}_{11}e^{t\nabla_{\boldsymbol{\theta}}\mathbf{h}(\boldsymbol{\theta}^*)}dt,$$

$$where \quad \mathbf{U}_{\boldsymbol{\theta}}(\alpha) = \sum_{i=1}^{N-1}\frac{1}{(\alpha(1+\lambda_i)+1)^2}\cdot\frac{1+\lambda_i}{1-\lambda_i}\mathbf{H}^T\mathbf{u}_i\mathbf{u}_i^T\mathbf{H}. \tag{11}$$

For $k \in \{1,3\}$, SA-SRRW in (4) is a *two-timescale SA* with *controlled* Markov noise. While a few works study the CLT of *two-timescale* SA with the stochastic input being a martingale-difference (i.i.d.) noise (Konda & Tsitsiklis, 2004; Mokkadem & Pelletier, 2006), a CLT result covering the case of controlled Markov noise (e.g., $k \in \{1,3\}$), a far more general setting than martingale-difference noise, is still an open problem. Thus, we here prove our CLT for $k \in \{1,3\}$ from scratch by a series of careful decompositions of the Markovian noise, ultimately into a martingale-difference term and several *non-vanishing* noise terms through repeated application of the Poisson equation (Benveniste et al., 2012; Fort, 2015). Although the form of the resulting asymptotic covariance looks similar to that for the martingale-difference case in (Konda & Tsitsiklis, 2004; Mokkadem & Pelletier, 2006) at first glance, they are not equivalent. Specifically, $\mathbf{V}_{\boldsymbol{\theta}}^{(k)}(\alpha)$ captures both the effect of SRRW hyper-parameter $\alpha$, as well as that of the underlying base Markov chain via eigen-pairs $(\lambda_i, \mathbf{u}_i)$ of its transition probability matrix $\mathbf{P}$ in matrix $\mathbf{U}$, whereas the latter only covers the martingale-difference noise terms as a special case.

When $k = 2$, that is, $\beta_n = \gamma_n$, algorithm (4) can be regarded as a single-timescale SA algorithm. In this case, we utilize the CLT in Fort (2015, Theorem 2.1) to obtain the implicit form of $\mathbf{V}^{(2)}(\alpha)$ as shown in Theorem 3.3. However, $\mathbf{J}_{12}(\alpha)$ being non-zero for $\alpha > 0$ restricts us from obtaining an explicit form for the covariance term corresponding to SA iterate errors $\boldsymbol{\theta}_n - \boldsymbol{\theta}^*$. On the other hand, for $k \in \{1,3\}$, the nature of two-timescale structure causes $\boldsymbol{\theta}_n$ and $\mathbf{x}_n$ to become asymptotically independent with zero correlation terms inside $\mathbf{V}^{(k)}(\alpha)$ in (10), and we can explicitly deduce $\mathbf{V}_{\boldsymbol{\theta}}^{(k)}(\alpha)$. We now take a deeper dive into $\alpha$ and study its effect on $\mathbf{V}_{\boldsymbol{\theta}}^{(k)}(\alpha)$.

**Covariance Ordering of SA-SRRW:** We refer the reader to Appendix F for proofs of all remaining results. We begin by focusing on case (i) and capturing the impact of $\alpha$ on $\mathbf{V}_{\boldsymbol{\theta}}^{(1)}(\alpha)$.

**Proposition 3.4.** *For all $\alpha_2 > \alpha_1 > 0$, we have $\mathbf{V}_{\boldsymbol{\theta}}^{(1)}(\alpha_2) <_L \mathbf{V}_{\boldsymbol{\theta}}^{(1)}(\alpha_1) <_L \mathbf{V}_{\boldsymbol{\theta}}^{(1)}(0)$. Furthermore, $\mathbf{V}_{\boldsymbol{\theta}}^{(1)}(\alpha)$ decreases to zero at a rate of $O(1/\alpha^2)$.*

Proposition 3.4 proves a monotonic reduction (in terms of Loewner ordering) of $\mathbf{V}_{\boldsymbol{\theta}}^{(1)}(\alpha)$ as $\alpha$ increases. Moreover, the decrease rate $O(1/\alpha^2)$ surpasses the $O(1/\alpha)$ rate seen in $\mathbf{V_x}(\alpha)$ and the sampling application in Doshi et al. (2023, Corollary 4.7), and is also empirically observed in our simulation in Section 4.[9] Suppose we consider the same SA now driven by an *i.i.d.* sequence $\{X_n\}$ with the same marginal distribution $\boldsymbol{\mu}$. Then, our Proposition 3.4 asserts that a token algorithm employing SRRW (walk on a graph) with large enough $\alpha$ on a *general* graph can actually produce better SA iterates with its asymptotic covariance going down to zero, than a 'hypothetical situation' where the walker is able to access any node $j$ with probability $\mu_j$ from anywhere in one step (more like a random jumper). This can be seen by noting that for large time $n$, the scaled MSE $\mathbb{E}[\|\boldsymbol{\theta}_n - \boldsymbol{\theta}^*\|^2]/\beta_n$ is composed of the diagonal entries of the covariance matrix $\mathbf{V}_{\boldsymbol{\theta}}$, which, as we discuss in detail in Appendix F.2, are decreasing in $\alpha$ as a consequence of the Loewner ordering in Proposition 3.4. For large enough $\alpha$, the scaled MSE for SA-SRRW becomes smaller than its *i.i.d.* counterpart, which is always a constant. Although Doshi et al. (2023) alluded this for sampling applications with $\mathbf{V_x}(\alpha)$, we broaden its horizons to distributed optimization problem with $\mathbf{V}_{\boldsymbol{\theta}}(\alpha)$ using tokens on graphs. Our subsequent result concerns the performance comparison between cases (i) and (iii).

**Corollary 3.5.** *For any $\alpha > 0$, we have $\mathbf{V}_{\boldsymbol{\theta}}^{(1)}(\alpha) <_L \mathbf{V}_{\boldsymbol{\theta}}^{(3)}(\alpha) = \mathbf{V}_{\boldsymbol{\theta}}^{(3)}(0)$.*

We show that case (i) is asymptotically better than case (iii) for $\alpha > 0$. In view of Proposition 3.4 and Corollary 3.5, the advantages of case (i) become prominent.

---

[9]Further insights of $O(1/\alpha^2)$ are tied to the two-timescale structure, particularly $\beta_n = o(\gamma_n)$ in case (i), which places $\boldsymbol{\theta}_n$ on the slow timescale so that the correlation terms $\mathbf{J}_{12}(\alpha), \mathbf{J}_{22}(\alpha)$ in the Jacobian matrix $\mathbf{J}(\alpha)$ in (8) come into play. Technical details are referred to Appendix E.2, where we show the form of $\mathbf{U}_{\boldsymbol{\theta}}(\alpha)$.

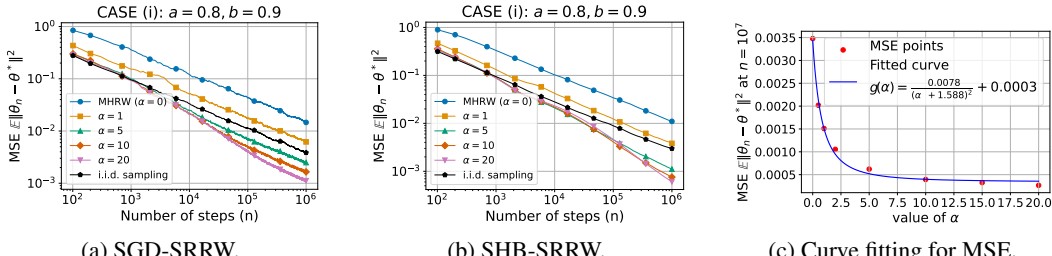

(a) SGD-SRRW.  (b) SHB-SRRW.  (c) Curve fitting for MSE.

Figure 2: Simulation results under case (i): (a) and (b) show the performance of SGD-SRRW and SHB-SRRW for various $\alpha$ values. (c) shows that MSE decreases at $O(1/\alpha^2)$ speed.

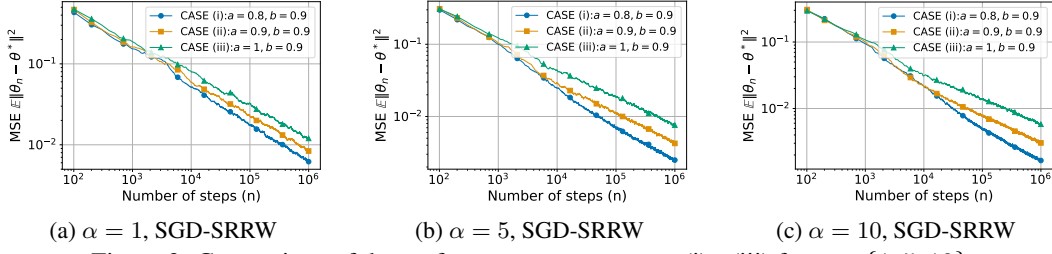

(a) $\alpha = 1$, SGD-SRRW  (b) $\alpha = 5$, SGD-SRRW  (c) $\alpha = 10$, SGD-SRRW

Figure 3: Comparison of the performance among cases (i) - (iii) for $\alpha \in \{1, 5, 10\}$.

## 4 SIMULATION

In this section, we simulate our SA-SRRW algorithm on the wikiVote graph (Leskovec & Krevl, 2014), comprising 889 nodes and 2914 edges. We configure the SRRW's base Markov chain $\mathbf{P}$ as the MHRW with a uniform target distribution $\boldsymbol{\mu} = \frac{1}{N}\mathbf{1}$. For distributed optimization, we consider the following $L_2$ regularized binary classification problem:

$$\min_{\boldsymbol{\theta} \in \mathbb{R}^D} \left\{ f(\boldsymbol{\theta}) \triangleq \frac{1}{N} \sum_{i=1}^{N} \log\left(1 + e^{\boldsymbol{\theta}^T \mathbf{s}_i}\right) - y_i\left(\boldsymbol{\theta}^T \mathbf{s}_i\right) + \frac{\kappa}{2}\|\boldsymbol{\theta}\|^2 \right\}, \tag{12}$$

where $\{(\mathbf{s}_i, y_i)\}_{i=1}^{N}$ is the *ijcnn1* dataset (with 22 features, i.e., $\mathbf{s}_i \in \mathbb{R}^{22}$) from LIBSVM (Chang & Lin, 2011), and penalty parameter $\kappa = 1$. Each node in the wikiVote graph is assigned one data point, thus 889 data points in total. We perform SRRW driven SGD (SGD-SRRW) and SRRW driven stochastic heavy ball (SHB-SRRW) algorithms (see (13) in Appendix A for its algorithm). We fix the step size $\beta_n = (n+1)^{-0.9}$ for the SA iterates and adjust $\gamma_n = (n+1)^{-a}$ in the SRRW iterates to cover all three cases discussed in this paper: (i) $a = 0.8$; (ii) $a = 0.9$; (iii) $a = 1$. We use mean square error (MSE), i.e., $\mathbb{E}[\|\boldsymbol{\theta}_n - \boldsymbol{\theta}^*\|^2]$, to measure the error on the SA iterates.

Our results are presented in Figures 2 and 3, where each experiment is repeated 100 times. Figures 2a and 2b, based on wikiVote graph, highlight the consistent performance ordering across different $\alpha$ values for both algorithms over almost all time (not just asymptotically). Notably, curves for $\alpha \geq 5$ outperform that of the *i.i.d.* sampling (in black) even under the graph constraints. Figure 2c on the smaller *Dolphins* graph (Rossi & Ahmed, 2015) - 62 nodes and 159 edges - illustrates that the points of $(\alpha, \text{MSE})$ pair arising from SGD-SRRW at time $n = 10^7$ align with a curve in the form of $g(x) = \frac{c_1}{(x+c_2)^2} + c_3$ to showcase $O(1/\alpha^2)$ rates. This smaller graph allows for longer simulations to observe the asymptotic behaviour. Additionally, among the three cases examined at identical $\alpha$ values, Figures 3a - 3c confirm that case (i) performs consistently better than the rest, underscoring its superiority in practice. Further results, including those from non-convex functions and additional datasets, are deferred to Appendix H due to space constraints.

## 5 CONCLUSION

In this paper, we show both theoretically and empirically that the SRRW as a drop-in replacement for Markov chains can provide significant performance improvements when used for token algorithms, where the *acceleration* comes purely from the careful analysis of the stochastic input of the algorithm, without changing the optimization iteration itself. Our paper is an instance where the asymptotic analysis approach allows the design of better algorithms despite the usage of unconventional noise sequences such as nonlinear Markov chains like the SRRW, for which traditional finite-time analytical approaches fall short, thus advocating their wider adoption.

## 6 ACKNOWLEDGMENTS AND DISCLOSURE OF FUNDING

We thank the anonymous reviewers for their constructive comments, especially Reviewer DxLx for raising future directions. This work was supported in part by National Science Foundation under Grant Nos. CNS-2007423 and IIS-1910749.

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

## A  EXAMPLES OF STOCHASTIC ALGORITHMS OF THE FORM (2).

In the literature of stochastic optimizations, many SGD variants have been proposed by introducing an auxiliary variable to improve convergence. In what follows, we present two SGD variants with decreasing step size that can be presented in the form of (2): SHB (Gadat et al., 2018; Li et al., 2022) and momentum-based algorithm (Barakat et al., 2021; Barakat & Bianchi, 2021).

$$\begin{cases} \boldsymbol{\theta}_{n+1} = \boldsymbol{\theta}_n - \beta_{n+1}\mathbf{m}_n \\ \mathbf{m}_{n+1} = \mathbf{m}_n + \beta_{n+1}(\nabla F(\boldsymbol{\theta}_n, X_{n+1}) - \mathbf{m}_n), \end{cases} \qquad \begin{cases} \mathbf{v}_{n+1} = \mathbf{v}_n + \beta_{n+1}(\nabla F(\boldsymbol{\theta}_n, X_{n+1})^2 - \mathbf{v}_n), \\ \mathbf{m}_{n+1} = \mathbf{m}_n + \beta_{n+1}(\nabla F(\boldsymbol{\theta}_n, X_{n+1}) - \mathbf{m}_n), \\ \boldsymbol{\theta}_{n+1} = \boldsymbol{\theta}_n - \beta_{n+1}\mathbf{m}_n/\sqrt{\mathbf{v}_n + \epsilon}, \end{cases}$$

$$\text{(a). SHB} \qquad\qquad \text{(b). Momentum-based Algorithm} \tag{13}$$

where $\epsilon > 0$, $\boldsymbol{\theta}_n, \mathbf{m}_n, \mathbf{v}_n, \nabla F(\boldsymbol{\theta}, X) \in \mathbb{R}^d$, and the square and square root in (13) (b) are element-wise operators.[10]

For SHB, we introduce an augmented variable $\mathbf{z}_n$ and function $H(\mathbf{z}_n, X_{n+1})$ defined as follows:

$$\mathbf{z}_n \triangleq \begin{bmatrix} \boldsymbol{\theta}_n \\ \mathbf{m}_n \end{bmatrix} \in \mathbb{R}^{2d}, \quad H(\mathbf{z}_n, X_{n+1}) \triangleq \begin{bmatrix} -\mathbf{m}_n \\ \nabla F(\boldsymbol{\theta}_n, X_{n+1}) - \mathbf{m}_n \end{bmatrix} \in \mathbb{R}^{2d}.$$

For the general momentum-based algorithm, we define

$$\mathbf{z}_n \triangleq \begin{bmatrix} \mathbf{v}_n \\ \mathbf{m}_n \\ \boldsymbol{\theta}_n \end{bmatrix} \in \mathbb{R}^{3d}, \quad H(\mathbf{z}_n, X) \triangleq \begin{bmatrix} \nabla F(\boldsymbol{\theta}_n, X_{n+1})^2 - \mathbf{v}_n \\ \nabla F(\boldsymbol{\theta}_n, X_{n+1}) - \mathbf{m}_n \\ -\mathbf{m}_n/\sqrt{\mathbf{v}_n + \epsilon} \end{bmatrix} \in \mathbb{R}^{3d}.$$

Thus, we can reformulate both algorithms in (13) as $\mathbf{z}_{n+1} = \mathbf{z}_n + \beta_{n+1}H(\mathbf{z}_n, X_{n+1})$. This augmentation approach was previously adopted in (Gadat et al., 2018; Barakat et al., 2021; Barakat & Bianchi, 2021; Li et al., 2022) to analyze the asymptotic performance of algorithms in (13) using an *i.i.d.* sequence $\{X_n\}_{n\geq 0}$. Consequently, the general SA iteration (2) includes these SGD variants. However, we mainly focus on the CLT for the general SA driven by SRRW in this paper. Pursuing the explicit CLT results of these SGD variants with specific form of function $H(\boldsymbol{\theta}, X)$ driven by the SRRW sequence $\{X_n\}$ is out of the scope of this paper.

When we numerically test the SHB algorithm in Section 4, we use the exact form of (13) (a) and the stochastic sequence $\{X_n\}$ is now driven by the SRRW. Specifically, we consider MHRW with transition kernel $\mathbf{P}$ as the base Markov chain of the SRRW process, e.g.,

$$P_{ij} = \begin{cases} \min\left\{\frac{1}{d_i}, \frac{1}{d_j}\right\} & \text{if node } j \text{ is the neighbor of node } i, \\ 0 & \text{otherwise}, \end{cases}$$

$$P_{ii} = 1 - \sum_{j\in\mathcal{N}} P_{ij}.$$

Then, at each time step $n$,

$$\text{Draw:} \quad X_{n+1} \sim \mathbf{K}_{X_n,\cdot}[\mathbf{x}_n],$$

$$\text{where} \quad K_{ij}[\mathbf{x}] \triangleq \frac{P_{ij}(x_j/\mu_j)^{-\alpha}}{\sum_{k\in\mathcal{N}} P_{ik}(x_k/\mu_k)^{-\alpha}}, \quad \forall\, i, j \in \mathcal{N},$$

$$\text{Update:} \quad \mathbf{x}_{n+1} = \mathbf{x}_n + \gamma_{n+1}(\boldsymbol{\delta}_{X_{n+1}} - \mathbf{x}_n),$$

$$\boldsymbol{\theta}_{n+1} = \boldsymbol{\theta}_n - \beta_{n+1}\mathbf{m}_n,$$

$$\mathbf{m}_{n+1} = \mathbf{m}_n + \beta_{n+1}(\nabla F(\boldsymbol{\theta}_n, X_{n+1}) - \mathbf{m}_n).$$

---

[10]For ease of expression, we simplify the original SHB and momentum-based algorithms from Gadat et al. (2018); Li et al. (2022); Barakat et al. (2021); Barakat & Bianchi (2021), setting all tunable parameters to 1 and resulting in (13).

## B  DISCUSSION ON MEAN FIELD FUNCTION OF SRRW ITERATES (4b)

Non-asymptotic analyses have seen extensive attention recently in both *single-timescale* SA literature (Sun et al., 2018; Karimi et al., 2019; Chen et al., 2020b; 2022) and *two-timescale* SA literature (Doan, 2021; Zeng et al., 2021). Specifically, single-timescale SA has the following form:

$$\mathbf{x}_{n+1} = \mathbf{x}_n + \beta_{n+1} H(\mathbf{x}_n, X_{n+1}),$$

and function $h(\mathbf{x}) \triangleq \mathbb{E}_{X \sim \boldsymbol{\mu}}[H(\mathbf{x}, X)]$ is the mean field of function $H(\mathbf{x}, X)$. Similarly, for two-timescale SA, we have the following recursions:

$$\mathbf{x}_{n+1} = \mathbf{x}_n + \beta_{n+1} H_1(\mathbf{x}_n, \mathbf{y}_n, X_{n+1}),$$
$$\mathbf{y}_{n+1} = \mathbf{y}_n + \gamma_{n+1} H_2(\mathbf{x}_n, \mathbf{y}_n, X_{n+1}),$$

where $\{\beta_n\}$ and $\{\gamma_n\}$ are on different timescales, and function $h_i(\mathbf{x}, \mathbf{y}) \triangleq \mathbb{E}_{X \sim \boldsymbol{\mu}}[H_i(\mathbf{x}, \mathbf{y}, X)]$ is the mean field of function $H_i(\mathbf{x}, \mathbf{y}, X)$ for $i = \{1, 2\}$. All the aforementioned works require the mean field function $h(\mathbf{x})$ in the single-timescale SA (or $h_1(\mathbf{x}, \mathbf{y})$, $h_2(\mathbf{x}, \mathbf{y})$ in the two-timescale SA) to be globally Lipschitz with a Lipschitz constant $L$ to proceed with the derivation of finite-time bounds including the constant $L$.

Here, we show that the mean field function $\boldsymbol{\pi}[\mathbf{x}] - \mathbf{x}$ in the SRRW iterates (4b) is not globally Lipschitz, where $\boldsymbol{\pi}[\mathbf{x}]$ is the stationary distribution of the SRRW kernel $\mathbf{K}[\mathbf{x}]$ defined in (3). To this end, we show that each entry of Jacobian matrix of $\boldsymbol{\pi}[\mathbf{x}] - \mathbf{x}$ goes unbounded because a multivariate function is Lipschitz if and only if it has bounded partial derivatives. Note that from Doshi et al. (2023, Proposition 2.1), for the $i$-th entry of $\boldsymbol{\pi}[\mathbf{x}]$, we have

$$\boldsymbol{\pi}_i[\mathbf{x}] = \frac{\sum_{j \in \mathcal{N}} \mu_i P_{ij} (x_i/\mu_i)^{-\alpha} (x_j/\mu_j)^{-\alpha}}{\sum_{i \in \mathcal{N}} \sum_{j \in \mathcal{N}} \mu_i P_{ij} (x_i/\mu_i)^{-\alpha} (x_j/\mu_j)^{-\alpha}}. \tag{16}$$

Then, the Jacobian matrix of the mean field function $\boldsymbol{\pi}[\mathbf{x}] - \mathbf{x}$, which has been derived in Doshi et al. (2023, Proof of Lemma 3.4 in Appendix B), is given as follows:

$$\frac{\partial(\boldsymbol{\pi}_i[\mathbf{x}] - x_i)}{\partial x_j}$$
$$= \frac{2\alpha}{x_j} \cdot \frac{(\sum_{k \in \mathcal{N}} \mu_i P_{ik} (x_i/\mu_i)^{-\alpha} (x_k/\mu_k)^{-\alpha})(\sum_{k \in \mathcal{N}} \mu_j P_{jk} (x_j/\mu_j)^{-\alpha} (x_k/\mu_k)^{-\alpha})}{(\sum_{l \in \mathcal{N}} \sum_{k \in \mathcal{N}} \mu_l P_{lk} (x_l/\mu_l)^{-\alpha} (x_k/\mu_k)^{-\alpha})^2} \tag{17}$$
$$- \frac{\alpha}{x_j} \cdot \frac{\mu_i P_{ij} (x_i/\mu_i)^{-\alpha} (x_j/\mu_j)^{-\alpha}}{\sum_{l \in \mathcal{N}} \sum_{k \in \mathcal{N}} \mu_l P_{lk} (x_l/\mu_l)^{-\alpha} (x_k/\mu_k)^{-\alpha}}$$

for $i, j \in \mathcal{N}, i \neq j$, and

$$\frac{\partial(\boldsymbol{\pi}_i[\mathbf{x}] - x_i)}{\partial x_i}$$
$$= \frac{2\alpha}{x_i} \cdot \frac{(\sum_{k \in \mathcal{N}} \mu_i P_{ik} (x_i/\mu_i)^{-\alpha} (x_k/\mu_k)^{-\alpha})^2}{(\sum_{l \in \mathcal{N}} \sum_{k \in \mathcal{N}} \mu_l P_{lk} (x_l/\mu_l)^{-\alpha} (x_k/\mu_k)^{-\alpha})^2} \tag{18}$$
$$- \frac{\alpha}{x_i} \cdot \frac{\sum_{k \in \mathcal{N}} \mu_i P_{ik} (x_i/\mu_i)^{-\alpha} (x_k/\mu_k)^{-\alpha} + \mu_i P_{ii} (x_i/\mu_i)^{-2\alpha}}{\sum_{l \in \mathcal{N}} \sum_{k \in \mathcal{N}} \mu_l P_{lk} (x_l/\mu_l)^{-\alpha} (x_k/\mu_k)^{-\alpha}} - 1$$

for $i \in \mathcal{N}$. Since the empirical distribution $\mathbf{x} \in \text{Int}(\Sigma)$, we have $x_i \in (0, 1)$ for all $i \in \mathcal{N}$. For fixed $i$, assume $x_i = x_j$ and as they approach zero, the terms $(x_i/\mu_i)^{-\alpha}$, $(x_j/\mu_j)^{-\alpha}$ dominate the fraction in (17) and both the numerator and the denominator of the fraction have the same order in terms of $x_i, x_j$. Thus, we have

$$\frac{\partial(\boldsymbol{\pi}_i[\mathbf{x}] - x_i)}{\partial x_j} = O\left(\frac{1}{x_j}\right)$$

such that the $(i, j)$-th entry of the Jacobian matrix can go unbounded as $x_j \to 0$. Consequently, $\boldsymbol{\pi}[\mathbf{x}] - \mathbf{x}$ is not globally Lipschitz for $\mathbf{x} \in \text{Int}(\Sigma)$.

## C  DISCUSSION ON ASSUMPTION A3′

When $\gamma_n = o(\beta_n)$, iterates $\mathbf{x}_n$ has smaller step size compared to $\boldsymbol{\theta}_n$, thus converges 'slower' than $\boldsymbol{\theta}_n$. From Assumption A3′, $\boldsymbol{\theta}_n$ will intuitively converge to some point $\rho(\mathbf{x})$ with the current value $\mathbf{x}$ from the iteration $\mathbf{x}_n$, i.e., $\mathbb{E}_{X \sim \boldsymbol{\pi}[\mathbf{x}]}[H(\rho(\mathbf{x}), X)] = 0$, while the Hurwitz condition is to ensure the stability around $\rho(\mathbf{x})$. We can see that Assumption A3 is less stringent than A3′ in that it only assumes such condition when $\mathbf{x} = \boldsymbol{\mu}$ such that $\rho(\boldsymbol{\mu}) = \boldsymbol{\theta}^*$ rather than for all $\mathbf{x} \in \text{Int}(\Sigma)$.

One special instance of Assumption A3′ is by assuming the linear SA, e.g., $H(\boldsymbol{\theta}, i) = A_i \boldsymbol{\theta} + b_i$. In this case, $\mathbb{E}_{X \sim \boldsymbol{\pi}[\mathbf{x}]}[H(\rho(\mathbf{x}), X)] = 0$ is equivalent to $\mathbb{E}_{i \sim \boldsymbol{\pi}[\mathbf{x}]}[A_i]\rho(\mathbf{x}) + \mathbb{E}_{i \sim \boldsymbol{\pi}[\mathbf{x}]}[b_i] = 0$. Under the condition that for every $\mathbf{x} \in \text{Int}(\Sigma)$, matrix $\mathbb{E}_{i \sim \boldsymbol{\pi}[\mathbf{x}]}[A_i]$ is invertible, we then have

$$\rho(\mathbf{x}) = -(\mathbb{E}_{i \sim \boldsymbol{\pi}[\mathbf{x}]}[A_i])^{-1} \cdot \mathbb{E}_{i \sim \boldsymbol{\pi}[\mathbf{x}]}[b_i].$$

However, this condition is quite strict. Loosely speaking, $\mathbb{E}_{i \sim \boldsymbol{\pi}[\mathbf{x}]}[A_i]$ being invertible for any $\mathbf{x}$ is similar to saying that any convex combination of $\{A_i\}$ is invertible. For example, if we assume $\{A_i\}_{i \in \mathcal{N}}$ are negative definite and they all share the same eigenbasis $\{\mathbf{u}_i\}$, e.g., $A_i = \sum_{j=1}^{D} \lambda_j^i \mathbf{u}_i \mathbf{u}_i^T$ and $\lambda_j^i < 0$ for all $i \in \mathcal{N}, j \in [D]$. Then, $\mathbb{E}_{i \sim \boldsymbol{\pi}[\mathbf{x}]}[A_i]$ is invertible.

Another example for Assumption A3′ is when $H(\boldsymbol{\theta}, i) = H(\boldsymbol{\theta}, j)$ for all $i, j \in \mathcal{N}$, which implies that each agent in the distributed learning has the same local dataset to collaboratively train the model. In this example, $\rho(\mathbf{x}) = \boldsymbol{\theta}^*$ such that

$$\mathbb{E}_{i \sim \boldsymbol{\pi}[\mathbf{x}]}[H(\rho(\mathbf{x}), i)] = h(\boldsymbol{\theta}^*) = 0,$$

$$\mathbb{E}_{i \sim \boldsymbol{\pi}[\mathbf{x}]}[\nabla H(\rho(\mathbf{x}), i)] + \frac{\mathbb{1}_{\{b=1\}}}{2}\mathbf{I} = \nabla h(\boldsymbol{\theta}^*) + \frac{\mathbb{1}_{\{b=1\}}}{2}\mathbf{I} \quad \text{being Hurwitz.}$$

## D  PROOF OF LEMMA 3.1 AND LEMMA 3.2

In this section, we demonstrate the almost sure convergence of both $\boldsymbol{\theta}_n$ and $\mathbf{x}_n$ together. This proof naturally incorporates the almost certain convergence of the SRRW iteration in Lemma 3.1, since $\mathbf{x}_n$ is independent of $\boldsymbol{\theta}_n$ (as indicated in (4)), allowing us to separate out its asymptotic results. The same reason applies to the CLT analysis of SRRW iterates and we refer the reader to Section E.1 for the CLT result of $\mathbf{x}_n$ in Lemma 3.1.

We will use different techniques for different settings of step sizes in Assumption A2. Specifically, for step sizes $\gamma_n = (n + 1)^{-a}, \beta_n = (n + 1)^{-b}$, we consider the following scenarios:

**Scenario 1:** We consider case(ii): $1/2 < a = b \leq 1$, and will apply the almost sure convergence result of the single-timescale stochastic approximation in Theorem G.8 and verify all the conditions therein.

**Scenario 2:** We consider both case(i): $1/2 < a < b \leq 1$ and case (iii): $1/2 < b < a \leq 1$. In these two cases, step sizes $\gamma_n, \beta_n$ decrease at different rates, thereby putting iterates $\mathbf{x}_n, \boldsymbol{\theta}_n$ on different timescales and resulting in a two-timescale structure. We will apply the existing almost sure convergence result of the two-timescale stochastic approximation with iterate-dependent Markov chain in Yaji & Bhatnagar (2020, Theorem 4) where our SA-SRRW algorithm can be regarded as a special instance.[11]

### D.1  SCENARIO 1

In Scenario 1, we have $\beta_n = \gamma_n$. First, we rewrite (4) as

$$\begin{bmatrix} \boldsymbol{\theta}_{n+1} \\ \mathbf{x}_{n+1} \end{bmatrix} = \begin{bmatrix} \boldsymbol{\theta}_n \\ \mathbf{x}_n \end{bmatrix} + \gamma_{n+1} \begin{bmatrix} H(\boldsymbol{\theta}_n, X_{n+1}) \\ \boldsymbol{\delta}_{X_{n+1}} - \mathbf{x}_n \end{bmatrix}. \tag{19}$$

---

[11]However, Yaji & Bhatnagar (2020) paper only analysed the almost sure convergence. The central limit theorem analysis remains unknown in the literature for the two-timescale stochastic approximation with iterate-dependent Markov chains. Thus, our CLT analysis in Section E for this two-timescale structure with iterate-dependent Markov chain is still novel and recognized as our contribution.

By augmentations, we define the variable $\mathbf{z}_n \triangleq \begin{bmatrix} \boldsymbol{\theta}_n \\ \mathbf{x}_n \end{bmatrix} \in \mathbb{R}^{(N+D)\times 1}$ and the function $G(\mathbf{z}_n, i) \triangleq \begin{bmatrix} H(\boldsymbol{\theta}_n, i) \\ \boldsymbol{\delta}_i - \mathbf{x}_n \end{bmatrix} \in \mathbb{R}^{(N+d)\times 1}$. In addition, we define a new Markov chain $\{Y_n\}_{n\geq 0}$ in the same state space $\mathcal{N}$ as SRRW sequence $\{X_n\}_{n\geq 0}$. With slight abuse of notation, the transition kernel of $\{Y_n\}$ is denoted by $\mathbf{K}'[\mathbf{z}_n] \equiv \mathbf{K}[\mathbf{x}_n]$ and its stationary distribution $\boldsymbol{\pi}'[\mathbf{z}_n] \equiv \boldsymbol{\pi}[\mathbf{x}_n]$, where $\mathbf{K}[\mathbf{x}_n]$ and $\boldsymbol{\pi}(\mathbf{x}_n)$ are the transition kernel and its corresponding stationary distribution of SRRW, with $\boldsymbol{\pi}[\mathbf{x}]$ of the form

$$\pi_i[\mathbf{x}] \propto \sum_{j\in\mathcal{N}} \mu_i P_{ij}(x_i/\mu_i)^{-\alpha}(x_j/\mu_j)^{-\alpha}. \tag{20}$$

Recall that $\boldsymbol{\mu}$ is the fixed point, i.e., $\boldsymbol{\pi}[\boldsymbol{\mu}] = \boldsymbol{\mu}$, and $\mathbf{P}$ is the base Markov chain inside SRRW (see (3)). Then, the mean field

$$g(\mathbf{z}) = \mathbb{E}_{Y\sim\boldsymbol{\pi}'(\mathbf{z})}[G(\mathbf{z}, Y)] = \begin{bmatrix} \sum_{i\in\mathcal{N}} \pi_i[\mathbf{x}]H(\boldsymbol{\theta}, i) \\ \boldsymbol{\pi}[\mathbf{x}] - \mathbf{x} \end{bmatrix},$$

and $\mathbf{z}^* = (\boldsymbol{\theta}^*, \boldsymbol{\mu})$ for $\boldsymbol{\theta}^* \in \Theta$ in Assumption A3 is the root of $g(\mathbf{z})$, i.e., $g(\mathbf{z}^*) = 0$. The augmented iteration (19) becomes

$$\mathbf{z}_{n+1} = \mathbf{z}_n + \gamma_{n+1} G(\mathbf{z}_n, Y_{n+1}) \tag{21}$$

with the goal of solving $g(\mathbf{z}) = 0$. Therefore, we can treat (21) as an SA algorithm driven by a Markov chain $\{Y_n\}_{n\geq 0}$ with its kernel $\mathbf{K}'[\mathbf{z}]$ and stationary distribution $\boldsymbol{\pi}'[\mathbf{z}]$, which has been widely studied in the literature (e.g., Delyon (2000); Benveniste et al. (2012); Fort (2015); Li et al. (2023)). In what follows, we demonstrate that for any initial point $\mathbf{z}_0 = (\boldsymbol{\theta}_0, \mathbf{x}_0) \in \mathbb{R}^D \times \text{Int}(\Sigma)$, the SRRW iteration $\{\mathbf{x}_n\}_{n\geq 0}$ will almost surely converge to the target distribution $\boldsymbol{\mu}$, and the SA iteration $\{\boldsymbol{\theta}_n\}_{n\geq 0}$ will almost surely converge to the set $\Theta$.

Now we verify conditions C1 - C4 in Theorem G.8. Our assumption A4 is equivalent to condition C1 and assumption A2 corresponds to condition C2. For condition C3, we set $\nabla w(\mathbf{z}) \equiv -g(\mathbf{z})$, and the set $S \equiv \{\mathbf{z}^*|\boldsymbol{\theta}^* \in \Theta, \mathbf{x}^* = \boldsymbol{\mu}\}$, including disjoint points. For condition C4, since $\mathbf{K}'[\mathbf{z}]$, or equivalently $\mathbf{K}[\mathbf{x}]$, is ergodic and time-reversible for a given $\mathbf{z}$, as shown in the SRRW work Doshi et al. (2023), it automatically ensures a solution to the Poisson equation, which has been well discussed in Chen et al. (2020a, Section 2) and Benveniste et al. (2012); Meyn (2022). To show (97) and (98) in condition C4, for each given $\mathbf{z}$ and any $i \in \mathcal{N}$, we need to give the explicit solution $m_{\mathbf{z}}(i)$ to the Poisson equation $m_{\mathbf{z}}(i) - (\mathbf{K}'_{\mathbf{z}}m_{\mathbf{z}})(i) = G(\mathbf{z}, i) - g(\mathbf{z})$ in (96). The notation $(\mathbf{K}'_{\mathbf{z}}m_{\mathbf{z}})(i)$ is defined as follows.

$$(\mathbf{K}'_{\mathbf{z}}m_{\mathbf{z}})(i) = \sum_{j\in\mathcal{N}} \mathbf{K}'_{\mathbf{z}}(i,j)m(\mathbf{z}, j).$$

Let $\mathbf{G}(\mathbf{z}) \triangleq [G(\mathbf{z}, 1), \cdots, G(\mathbf{z}, N)]^T \in \mathbb{R}^{N\times D}$. We use $[\mathbf{A}]_{:,i}$ to denote the $i$-th column of matrix $\mathbf{A}$. Then, we let $m_{\mathbf{z}}(i)$ such that

$$m_{\mathbf{z}}(i) = \sum_{k=0}^{\infty} \left([\mathbf{G}(\mathbf{z})(\mathbf{K}'[\mathbf{z}]^k)^T]_{[:,i]} - g(\mathbf{z})\right) = \sum_{k=0}^{\infty}[\mathbf{G}(\mathbf{z})((\mathbf{K}'[\mathbf{z}]^k)^T - \boldsymbol{\pi}'[\mathbf{z}]\mathbf{1}^T)]_{[:,i]}. \tag{22}$$

In addition,

$$(\mathbf{K}'_{\mathbf{z}}m_{\mathbf{z}})(i) = \sum_{k=1}^{\infty}[\mathbf{G}(\mathbf{z})(\mathbf{K}'[\mathbf{z}]^T - \boldsymbol{\pi}'[\mathbf{z}]\mathbf{1}^T)]_{[:,i]}. \tag{23}$$

We can check that the $m_{\mathbf{z}}(i)$ form in (22) is indeed the solution of the above Poisson equation. Now, by induction, we get $\mathbf{K}'[\mathbf{z}]^k - \mathbf{1}\boldsymbol{\pi}'[\mathbf{z}]^T = (\mathbf{K}'[\mathbf{z}] - \mathbf{1}\boldsymbol{\pi}'[\mathbf{z}]^T)^k$ for $k \geq 1$ and for $k = 0$, $\mathbf{K}'[\mathbf{z}]^0 - \mathbf{1}\boldsymbol{\pi}'[\mathbf{z}]^T = (\mathbf{K}'[\mathbf{z}] - \mathbf{1}\boldsymbol{\pi}'[\mathbf{z}]^T)^0 - \mathbf{1}\boldsymbol{\pi}'[\mathbf{z}]^T$. Then,

$$\begin{aligned}
m_{\mathbf{z}}(i) &= \sum_{k=0}^{\infty}[\mathbf{G}(\mathbf{z})(\mathbf{K}'[\mathbf{z}]^T - \boldsymbol{\pi}'[\mathbf{z}]\mathbf{1}^T)^k]_{[:,i]} - g(\mathbf{z}) \\
&= \left[\mathbf{G}(\mathbf{z})\sum_{k=0}^{\infty}(\mathbf{K}'[\mathbf{z}]^T - \boldsymbol{\pi}'[\mathbf{z}]\mathbf{1}^T)^k\right]_{[:,i]} - g(\mathbf{z}) \\
&= \left[\mathbf{G}(\mathbf{z})(\mathbf{I} - \mathbf{K}'[\mathbf{z}]^T + \boldsymbol{\pi}'[\mathbf{z}]\mathbf{1}^T)^{-1}\right]_{[:,i]} - g(\mathbf{z}) \\
&= \sum_{j\in\mathcal{N}}(\mathbf{I} - \mathbf{K}'[\mathbf{z}] + \mathbf{1}\boldsymbol{\pi}'[\mathbf{z}]^T)^{-1}(i,j)G(\mathbf{z}, j) - g(\mathbf{z}).
\end{aligned} \tag{24}$$

Here, $(\mathbf{I} - \mathbf{K}'[\mathbf{z}] + \mathbf{1}\boldsymbol{\pi}'[\mathbf{z}]^T)^{-1}$ is well defined because $\mathbf{K}'[\mathbf{z}]$ is ergodic and time-reversible for any given $\mathbf{z}$ (proved in Doshi et al. (2023, Appendix A)). Now that both functions $H(\boldsymbol{\theta}, i)$ and $\boldsymbol{\delta}_i - \mathbf{x}$ are bounded for each compact subset of $\mathbb{R}^D \times \Sigma$ by our assumption A1, function $G(\mathbf{z}, i)$ is also bounded within the compact subset of its domain. Thus, function $m_{\mathbf{z}}(i)$ is bounded, and (97) is verified. Moreover, for a fixed $i \in \mathcal{N}$,

$$\sum_{j \in \mathcal{N}} (\mathbf{I} - \mathbf{K}'[\mathbf{z}] + \mathbf{1}\boldsymbol{\pi}'[\mathbf{z}]^T)^{-1}(i, j)\boldsymbol{\delta}_j = (\mathbf{I} - \mathbf{K}'[\mathbf{z}] + \mathbf{1}\boldsymbol{\pi}'[\mathbf{z}]^T)^{-1}_{[:,i]} = (\mathbf{I} - \mathbf{K}[\mathbf{x}] + \mathbf{1}\boldsymbol{\pi}[\mathbf{x}]^T)^{-1}_{[:,i]}$$

and this vector-valued function is continuous in $\mathbf{x}$ because $\mathbf{K}[\mathbf{x}], \boldsymbol{\pi}[\mathbf{x}]$ are continuous. We then rewrite (24) as

$$m_{\mathbf{z}}(i) = \begin{bmatrix} \sum_{j \in \mathcal{N}} (\mathbf{I} - \mathbf{K}[\mathbf{x}] + \mathbf{1}\boldsymbol{\pi}[\mathbf{x}]^T)^{-1}(i, j)H(\mathbf{x}, j) \\ (\mathbf{I} - \mathbf{K}[\mathbf{x}]^T + \boldsymbol{\pi}[\mathbf{x}]\mathbf{1}^T)^{-1}_{[:,i]} \end{bmatrix} - \begin{bmatrix} \sum_{i \in \mathcal{N}} \pi_i[\mathbf{x}]H(\boldsymbol{\theta}, i) \\ \boldsymbol{\pi}[\mathbf{x}] - \mathbf{x} \end{bmatrix}.$$

With continuous functions $H(\boldsymbol{\theta}, i), \mathbf{K}[\mathbf{x}], \boldsymbol{\pi}[\mathbf{x}]$, we have $m_{\mathbf{z}}(i)$ continuous with respect to $\mathbf{z}$, so does $(\mathbf{K}'_{\mathbf{z}}m_{\mathbf{z}})(i)$. This implies that functions $m_{\mathbf{z}}(i)$ and $(\mathbf{K}'_{\mathbf{z}}m_{\mathbf{z}})(i)$ are locally Lipschitz, which satisfies (98) with $\phi_{\mathcal{C}}(x) = C_{\mathcal{C}}x$ for some constant $C_{\mathcal{C}}$ that depends on the compact set $\mathcal{C}$. Therefore, condition C4 is checked, and we can apply Theorem G.8 to show the almost convergence result of (19), i.e., almost surely,

$$\lim_{n \to \infty} \mathbf{x}_n = \boldsymbol{\mu}, \quad \text{and} \quad \limsup_{n \to \infty} \inf_{\boldsymbol{\theta}^* \in \Theta} \|\boldsymbol{\theta}_n - \boldsymbol{\theta}^*\| = 0.$$

Therefore, the almost sure convergence of $\mathbf{x}_n$ in Lemma 3.1 is also proved. This finishes the proof in Scenario 1.

## D.2 SCENARIO 2

Now in this subsection, we consider the steps sizes $\gamma_n, \beta_n$ with $1/2 < a < b \leq 1$ and $1/2 < b < a \leq 1$. We will frequently use assumptions (B1) - (B5) in Section G.3 and Theorem G.10 to prove the almost sure convergence.

### D.2.1 CASE (I): $1/2 < a < b \leq 1$

In case (i), $\boldsymbol{\theta}_n$ is on the slow timescale and $\mathbf{x}_n$ is on the fast timescale because iteration $\boldsymbol{\theta}_n$ has smaller step size than $\mathbf{x}_n$, making $\boldsymbol{\theta}_n$ converge slower than $\mathbf{x}_n$. Here, we consider the two-timescale SA of the form:

$$\boldsymbol{\theta}_{n+1} = \boldsymbol{\theta}_n + \beta_{n+1}H(\boldsymbol{\theta}, X_{n+1}),$$
$$\mathbf{x}_{n+1} = \mathbf{x}_n + \gamma_{n+1}(\boldsymbol{\delta}_{X_{n+1}} - \mathbf{x}).$$

Now, we verify assumptions (B1) - (B5) listed in Section G.3.

- Assumptions (B1) and (B5) are satisfied by our assumptions A2 and A4.

- Our assumption A3 shows that the function $H(\boldsymbol{\theta}, X)$ is continuous and differentiable w.r.t $\boldsymbol{\theta}$ and grows linearly with $\|\boldsymbol{\theta}\|$. In addition, $\boldsymbol{\delta}_X - \mathbf{x}$ also satisfies this property. Therefore, (B2) is satisfied.

- Now that the function $\boldsymbol{\pi}[\mathbf{x}] - \mathbf{x}$ is independent of $\boldsymbol{\theta}$, we can set $\rho(\boldsymbol{\theta}) = \boldsymbol{\mu}$ for any $\boldsymbol{\theta} \in \mathbb{R}^D$ such that $\boldsymbol{\pi}[\boldsymbol{\mu}] - \boldsymbol{\mu} = 0$ from Doshi et al. (2023, Proposition 3.1), and

$$\nabla_{\mathbf{x}}(\boldsymbol{\pi}(\mathbf{x}) - \mathbf{x})|_{\mathbf{x}=\boldsymbol{\mu}} = 2\alpha\mathbf{u}\mathbf{1}^T - \alpha\mathbf{P}^T - (\alpha + 1)\mathbf{I}$$

from Doshi et al. (2023, Lemma 3.4), which is Hurwitz. Furthermore, $\rho(\boldsymbol{\theta}) = \boldsymbol{\mu}$ inherently satisfies the condition $\|\rho(\boldsymbol{\theta})\| \leq L_2(1 + \|\boldsymbol{\theta}\|)$ for any $L_2 \geq \|\boldsymbol{\mu}\|$. Thus, conditions (i) - (iii) in (B3) are satisfied. Additionally, $\sum_{i \in \mathcal{N}} \pi_i[\rho(\boldsymbol{\theta})]H(\boldsymbol{\theta}, i) = \sum_{i \in \mathcal{N}} \pi_i[\mathbf{x}] = \mathbf{h}(\boldsymbol{\theta})$ such that for $\boldsymbol{\theta}^* \in \Theta$ defined in assumption A3, $\sum_{i \in \mathcal{N}} \pi_i[\rho(\boldsymbol{\theta}^*)]H(\boldsymbol{\theta}^*, i) = \mathbf{h}(\boldsymbol{\theta}^*) = 0$, and $\nabla_{\boldsymbol{\theta}}\mathbf{h}(\boldsymbol{\theta}^*)$ is Hurwitz. Therefore, (B3) is checked.

- Assumption (B4) is verified by the nature of SRRW, i.e., its transition kernel $\mathbf{K}[\mathbf{x}]$ and the corresponding stationary distribution $\boldsymbol{\pi}[\mathbf{x}]$ with $\boldsymbol{\pi}[\boldsymbol{\mu}] = \boldsymbol{\mu}$.

Consequently, assumptions (B1) - (B5) are satisfied by our assumptoins A1 - A4 and by Theorem G.10, we have $\lim_{n\to\infty} \mathbf{x}_n = \boldsymbol{\mu}$ and $\boldsymbol{\theta}_n \to \Theta$ almost surely.

Next, we consider $1/2 < b < a \leq 1$. As discussed before, (B1), (B2), (B4) and (B5) are satisfied by our assumptions A1 - A4 and the properties of SRRW. The only difference for this step size setting, compared to the previous one $1/2 < a < b \leq 1$, is that the roles of $\boldsymbol{\theta}_n, \mathbf{x}_n$ are now flipped, that is, $\boldsymbol{\theta}_n$ is now on the fast timescale while $\mathbf{x}_n$ is on the slow timescale. By a much stronger Assumption A3′, for any $\mathbf{x} \in \text{Int}(\Sigma)$, (i) $\mathbb{E}_{X\sim\boldsymbol{\pi}[\mathbf{x}]}[H(\rho(\mathbf{x}), X)] = 0$; (ii) $\mathbb{E}_{X\sim\boldsymbol{\pi}[\mathbf{x}]}[\nabla H(\rho(\mathbf{x}), X)]$ is Hurwitz; (iii) $\|\rho(\mathbf{x})\| \leq L_2(1 + \|\mathbf{x}\|)$. Hence, conditions (i) - (iii) in (B3) are satisfied. Moreover, we have $\boldsymbol{\pi}[\boldsymbol{\mu}] - \boldsymbol{\mu} = 0$, $\nabla(\boldsymbol{\pi}[\mathbf{x}] - \mathbf{x})|_{\mathbf{x}=\boldsymbol{\mu}}$ being Hurwitz, as mentioned in the previous part. Therefore, (B3) is verified. Accordingly, (B1) - (B5) are checked by our assumptions A1, A2, A3′, A4. By Theorem G.10, we have $\lim_{n\to\infty} \mathbf{x}_n = \boldsymbol{\mu}$ and $\boldsymbol{\theta}_n \to \Theta$ almost surely.

# E  PROOF OF THEOREM 3.3

This section is devoted to the proof of Theorem 3.3, which also includes the proof of the CLT results for the SRRW iteration $\mathbf{x}_n$ in Lemma 3.1. We will use different techniques depending on the step sizes in Assumption A2. Specifically, for step sizes $\gamma_n = (n + 1)^{-a}, \beta_n = (n + 1)^{-b}$, we will consider three cases: case (i): $\beta_n = o(\gamma_n)$; case (ii): $\beta_n = \gamma_n$; and case (iii): $\gamma_n = o(\beta_n)$. For case (ii), we will use the existing CLT result for single-timescale SA in Theorem G.9. For cases (i) and (iii), we will construct our own CLT analysis for the two-timescale structure. We start with case (ii).

## E.1  CASE (II): $\beta_n = \gamma_n$

In this part, we stick to the notations for single-timescale SA studied in Section D.1. To utilize Theorem G.9, apart from Conditions C1 - C4 that have been checked in Section D.1, we still need to check conditions C5 and C6 listed in Section G.2.

Assumption A3 corresponds to condition C5. For condition C6, we need to obtain the explicit form of function $Q_{\mathbf{z}}$ to the Poisson equation defined in (96), that is,

$$Q_{\mathbf{z}}(i) - (\mathbf{K}'_{\mathbf{z}}Q_{\mathbf{z}})(i) = \psi(\mathbf{z}, i) - \mathbb{E}_{j\sim\boldsymbol{\pi}[\mathbf{z}]}[\psi(\mathbf{z}, j)]$$

where

$$\psi(\mathbf{z}, i) \triangleq \sum_{j\in\mathcal{N}} \mathbf{K}'_{\mathbf{z}}(i, j) m_{\mathbf{z}}(j) m_{\mathbf{z}}(j)^T - (\mathbf{K}'_{\mathbf{z}} m_{\mathbf{z}})(i)(\mathbf{K}'_{\mathbf{z}} m_{\mathbf{z}})(i)^T.$$

Following the similar steps in the derivation of $m_{\mathbf{z}}(i)$ from (22) to (24), we have

$$Q_{\mathbf{z}}(i) = \sum_{j\in\mathcal{N}} (\mathbf{I} - \mathbf{K}'[\mathbf{z}] + \mathbf{1}\boldsymbol{\pi}'[\mathbf{z}]^T)^{-1}(i, j) m_{\mathbf{z}}(j) - \pi'_j[\mathbf{z}] m_{\mathbf{z}}(j).$$

We also know that $Q_{\mathbf{z}}(i)$ and $(\mathbf{K}'_{\mathbf{z}}Q_{\mathbf{z}})(i)$ are continuous in $\mathbf{z}$ for any $i \in \mathcal{N}$. For any $\mathbf{z}$ in a compact set $\Omega$, $Q_{\mathbf{z}}(i)$ and $(\mathbf{K}'_{\mathbf{z}}Q_{\mathbf{z}})(i)$ are bounded because function $m_{\mathbf{z}}(i)$ is bounded. Therefore, C6 is checked. By Theorem G.9, assume $\mathbf{z}_n = \begin{bmatrix} \boldsymbol{\theta}_n \\ \mathbf{x}_n \end{bmatrix}$ converges to a point $\mathbf{z}^* = \begin{bmatrix} \boldsymbol{\theta}^* \\ \boldsymbol{\mu} \end{bmatrix}$ for $\boldsymbol{\theta}^* \in \Theta$, we have

$$\gamma_n^{-1/2}(\mathbf{z}_n - \mathbf{z}^*) \xrightarrow[n\to\infty]{dist.} N(0, \mathbf{V}), \tag{26}$$

where $\mathbf{V}$ is the solution of the following Lyapunov equation

$$\mathbf{V}\left(\frac{\mathbb{1}_{\{b=1\}}}{2}\mathbf{I} + \nabla g(\mathbf{z}^*)^T\right) + \left(\frac{\mathbb{1}_{\{b=1\}}}{2}\mathbf{I} + \nabla g(\mathbf{z}^*)\right)\mathbf{V} + \mathbf{U} = 0, \tag{27}$$

and $\mathbf{U} = \sum_{i\in\mathcal{N}} \mu_i \left(m_{\mathbf{z}^*}(i) m_{\mathbf{z}^*}(i)^T - (\mathbf{K}_{\mathbf{z}^*} m_{\mathbf{z}^*})(i)(\mathbf{K}_{\mathbf{z}^*} m_{\mathbf{z}^*})(i)^T\right).$

By algebraic calculations of derivative of $\boldsymbol{\pi}[\mathbf{x}]$ with respect to $\mathbf{x}$ in (20),[12] we can rewrite $\nabla g(\mathbf{z}^*)$ in terms of $\mathbf{x}, \boldsymbol{\theta}$, i.e.,

$$
\mathbf{J}(\alpha) \triangleq \nabla g(\mathbf{z}^*) = \begin{bmatrix} \frac{\partial \sum_{i \in \mathcal{N}} \pi_i[\mathbf{x}] H(\boldsymbol{\theta},i)}{\partial \boldsymbol{\theta}} & \frac{\partial \sum_{i \in \mathcal{N}} \pi_i[\mathbf{x}] H(\boldsymbol{\theta},i)}{\partial \mathbf{x}} \\ \frac{\partial (\boldsymbol{\pi}[\mathbf{x}] - \mathbf{x})}{\partial \boldsymbol{\theta}} & \frac{\partial \boldsymbol{\pi}[\mathbf{x}] - \mathbf{x}}{\partial \mathbf{x}} \end{bmatrix}_{\mathbf{z}=\mathbf{z}^*}
$$

$$
= \begin{bmatrix} \nabla \mathbf{h}(\boldsymbol{\theta}^*) & -\alpha \mathbf{H}^T(\mathbf{P}^T + \mathbf{I}) \\ \mathbf{0} & 2\alpha \boldsymbol{\mu} \mathbf{1}^T - \alpha \mathbf{P}^T - (\alpha+1)\mathbf{I} \end{bmatrix} \triangleq \begin{bmatrix} \mathbf{J}_{11} & \mathbf{J}_{12}(\alpha) \\ \mathbf{J}_{21} & \mathbf{J}_{22}(\alpha) \end{bmatrix},
$$

where matrix $\mathbf{H} = [H(\boldsymbol{\theta}^*, 1), \cdots, H(\boldsymbol{\theta}, N)]^T$. Then, we further clarify the matrix $\mathbf{U}$. Note that

$$
m_{\mathbf{z}^*}(i) = \sum_{k=0}^{\infty} [\mathbf{G}(\mathbf{z}^*)((\mathbf{P}^k)^T - \boldsymbol{\mu}\mathbf{1}^T)]_{[:,i]} = \sum_{k=0}^{\infty} [\mathbf{G}(\mathbf{z}^*)(\mathbf{P}^k)^T]_{[:,i]} = \mathbb{E}\left[\sum_{k=0}^{\infty} [G(\mathbf{z}^*, X_k)]\,\middle|\, X_0 = i\right],
$$
(28)

where the first equality holds because $\mathbf{K}'[\boldsymbol{\mu}] = \mathbf{P}$ from the definition of SRRW kernel (3), the second equality stems from $\mathbf{G}(\mathbf{z}^*)\boldsymbol{\mu} = g(\mathbf{z}^*) = 0$, and the last term is a conditional expectation over the *base* Markov chain $\{X_k\}_{k \geq 0}$ (with transition kernel $\mathbf{P}$) conditioned on $X_0 = i$. Similarly, with $(\mathbf{K}'_{\mathbf{z}} m_{\mathbf{z}})(i)$ in the form of (23), we have

$$
(\mathbf{K}'_{\mathbf{z}} m_{\mathbf{z}})(i) = \mathbb{E}\left[\sum_{k=1}^{\infty} [G(\mathbf{z}^*, X_k)]\,\middle|\, X_0 = i\right].
$$

From the form '$\sum_{i \in \mathcal{N}} \mu_i$' inside the matrix $\mathbf{U}$, the Markov chain $\{X_k\}_{k \geq 0}$ is in its stationary regime from the beginning, i.e., $X_k \sim \boldsymbol{\mu}$ for any $k \geq 0$. Hence,

$$
\mathbf{U} = \mathbb{E}\left[\left(\sum_{k=0}^{\infty} [G(\mathbf{z}^*, X_k)]\right)\left(\sum_{k=0}^{\infty} [G(\mathbf{z}^*, X_k)]\right)^T\right]
$$

$$
- \mathbb{E}\left[\left(\sum_{k=1}^{\infty} [G(\mathbf{z}^*, X_k)]\right)\left(\sum_{k=1}^{\infty} [G(\mathbf{z}^*, X_k)]\right)^T\right]
$$

$$
= \mathbb{E}\left[G(\mathbf{z}^*, X_0)G(\mathbf{z}^*, X_0)^T\right] + \mathbb{E}\left[G(\mathbf{z}^*, X_0)\left(\sum_{k=1}^{\infty} G(\mathbf{z}^*, X_k)\right)^T\right] \qquad (29)
$$

$$
+ \mathbb{E}\left[\left(\sum_{k=1}^{\infty} G(\mathbf{z}^*, X_k)\right)G(\mathbf{z}^*, X_0)^T\right]
$$

$$
= \mathrm{Cov}(G(\mathbf{z}^*, X_0), G(\mathbf{z}^*, X_0))
$$

$$
+ \sum_{k=1}^{\infty} [\mathrm{Cov}(G(\mathbf{z}^*, X_0), G(\mathbf{z}^*, X_k)) + \mathrm{Cov}(G(\mathbf{z}^*, X_k), G(\mathbf{z}^*, X_0))],
$$

where the covariance between $G(\mathbf{z}^*, X_0)$ and $G(\mathbf{z}^*, X_k)$ for the Markov chain $\{X_n\}$ in the stationary regime is $\mathrm{Cov}(G(\mathbf{z}^*, X_0), G(\mathbf{z}^*, X_k))$. By Brémaud (2013, Theorem 6.3.7), it is demonstrated that $\mathbf{U}$ is the sampling covariance of the base Markov chain $\mathbf{P}$ for the test function $G(\mathbf{z}^*, \cdot)$. Moreover, Brémaud (2013, equation (6.34)) states that this sampling covariance $\mathbf{U}$ can be rewritten in the following form:

$$
\mathbf{U} = \sum_{i=1}^{N-1} \mathbf{G}(\mathbf{z}^*)^T \mathbf{u}_i \mathbf{u}_i \mathbf{G}(\mathbf{z}^*) = \sum_{i=1}^{N-1} \frac{1+\lambda_i}{1-\lambda_i} \begin{bmatrix} \mathbf{H}^T \mathbf{u}_i \mathbf{u}_i^T \mathbf{H} & \mathbf{H}^T \mathbf{u}_i \mathbf{u}_i^T \\ \mathbf{u}_i \mathbf{u}_i^T \mathbf{H} & \mathbf{u}_i \mathbf{u}_i^T \end{bmatrix} \triangleq \begin{bmatrix} \mathbf{U}_{11} & \mathbf{U}_{12} \\ \mathbf{U}_{21} & \mathbf{U}_{22} \end{bmatrix}, \quad (30)
$$

where $\{(\lambda_i, \mathbf{u}_i)\}_{i \in \mathcal{N}}$ is the eigenpair of the transition kernel $\mathbf{P}$ of the ergodic and time-reversible base Markov chain. This completes the proof of case 1.

---

[12]One may refer to Doshi et al. (2023, Appendix B, Proof of Lemma 3.4) for the computation of $\frac{\partial \boldsymbol{\pi}[\mathbf{x}] - \mathbf{x}}{\partial \mathbf{x}}$.

*Remark* E.1. For the CLT result (26), we can look further into the asymptotic covariance matrix $\mathbf{V}$ as in (27). For convenience, we denote $\mathbf{V} = \begin{bmatrix} \mathbf{V}_{11} & \mathbf{V}_{12} \\ \mathbf{V}_{21} & \mathbf{V}_{22} \end{bmatrix}$ and $\mathbf{U}$ in the form of (30) such that

$$\begin{bmatrix} \mathbf{V}_{11} & \mathbf{V}_{12} \\ \mathbf{V}_{21} & \mathbf{V}_{22} \end{bmatrix} \left( \frac{\mathbb{1}_{\{b=1\}}}{2} \mathbf{I} + \mathbf{J}(\alpha)^T \right) + \left( \frac{\mathbb{1}_{\{b=1\}}}{2} \mathbf{I} + \mathbf{J}(\alpha) \right) \begin{bmatrix} \mathbf{V}_{11} & \mathbf{V}_{12} \\ \mathbf{V}_{21} & \mathbf{V}_{22} \end{bmatrix} + \mathbf{U} = 0. \tag{31}$$

For the SRRW iteration $\mathbf{x}_n$, from (26) we know that $\gamma_n^{-1/2}(\mathbf{x}_n - \boldsymbol{\mu}) \xrightarrow[n\to\infty]{dist.} N(\mathbf{0}, \mathbf{V}_4)$. Thus, in this remark, we want to obtain the closed form of $\mathbf{V}_{22}$. By algebraic computations of the bottom-right sub-block matrix, we have

$$\left( 2\alpha\boldsymbol{\mu}\mathbf{1}^T - \alpha\mathbf{P}^T - \left( \alpha + 1 - \frac{\mathbb{1}_{\{a=1\}}}{2} \right) \mathbf{I} \right) \mathbf{V}_{22}$$

$$+ \mathbf{V}_{22} \left( 2\alpha\boldsymbol{\mu}\mathbf{1}^T - \alpha\mathbf{P}^T - \left( \alpha + 1 - \frac{\mathbb{1}_{\{a=1\}}}{2} \right) \mathbf{I} \right)^T$$

$$+ \mathbf{U}_{22} = 0.$$

By using result of the closed form solution to the Lyapunov equation (e.g., Lemma G.1) and the eigendecomposition of $\mathbf{P}$, we have

$$\mathbf{V}_{22} = \sum_{i=1}^{N-1} \frac{1}{2\alpha(1 + \lambda_i) + 2 - \mathbb{1}_{\{a=1\}}} \cdot \frac{1 + \lambda_i}{1 - \lambda_i} \mathbf{u}_i \mathbf{u}_i^T. \tag{32}$$

### E.2 CASE (I): $\beta_n = o(\gamma_n)$

In this part, we mainly focus on the CLT of the SA iteration $\boldsymbol{\theta}_n$ because the SRRW iteration $\mathbf{x}_n$ is independent of $\boldsymbol{\theta}_n$ and its CLT result has been shown in Remark E.1.

#### E.2.1 DECOMPOSITION OF SA-SRRW ITERATION (4)

We slightly abuse the math notation and define the function

$$\mathbf{h}(\boldsymbol{\theta}, \mathbf{x}) \triangleq \mathbb{E}_{i \sim \boldsymbol{\pi}[\mathbf{x}]} H(\boldsymbol{\theta}, i) = \sum_{i \in \mathcal{N}} \pi_i[\mathbf{x}] H(\boldsymbol{\theta}, i)$$

such that $\mathbf{h}(\boldsymbol{\theta}, \boldsymbol{\mu}) \equiv \mathbf{h}(\boldsymbol{\theta})$. Then, we reformulate (25) as

$$\boldsymbol{\theta}_{n+1} = \boldsymbol{\theta}_n + \beta_{n+1}\mathbf{h}(\boldsymbol{\theta}_n, \mathbf{x}_n) + \beta_{n+1}(H(\boldsymbol{\theta}_n, X_{n+1}) - \mathbf{h}(\boldsymbol{\theta}_n, \mathbf{x}_n)). \tag{33a}$$

$$\mathbf{x}_{n+1} = \mathbf{x}_n + \gamma_{n+1}(\boldsymbol{\pi}[\mathbf{x}_n] - \mathbf{x}_n) + \gamma_{n+1}(\boldsymbol{\delta}_{X_{n+1}}) - \boldsymbol{\pi}[\mathbf{x}_n]). \tag{33b}$$

There exist functions $q_{\mathbf{x}} : \mathcal{N} \to \mathbb{R}^N, \tilde{H}_{\boldsymbol{\theta},\mathbf{x}} : \mathcal{N} \to \mathbb{R}^D$ satisfying the following Poisson equations

$$\boldsymbol{\delta}_i - \boldsymbol{\pi}(\mathbf{x}) = q_{\mathbf{x}}(i) - (\mathbf{K}_{\mathbf{x}} q_{\mathbf{x}})(i) \tag{34a}$$

$$H(\boldsymbol{\theta}, i) - \mathbf{h}(\boldsymbol{\theta}, \mathbf{x}) = \tilde{H}_{\boldsymbol{\theta},\mathbf{x}}(i) - (\mathbf{K}_{\mathbf{x}}\tilde{H}_{\boldsymbol{\theta},\mathbf{x}})(i), \tag{34b}$$

for any $\boldsymbol{\theta} \in \mathbb{R}^D, \mathbf{x} \in \text{Int}(\Sigma)$ and $i \in \mathcal{N}$, where $(\mathbf{K}_{\mathbf{x}} q_{\mathbf{x}})(i) \triangleq \sum_{j \in \mathcal{N}} K_{ij}[\mathbf{x}] q_{\mathbf{x}}(j), (\mathbf{K}_{\mathbf{x}}\tilde{H}_{\boldsymbol{\theta},\mathbf{x}})(j) \triangleq \sum_{j \in \mathcal{N}} K_{ij}[\mathbf{x}] \tilde{H}_{\boldsymbol{\theta},\mathbf{x}}(j)$. The existence and explicit form of the solutions $q_{\mathbf{x}}, \tilde{H}_{\boldsymbol{\theta},\mathbf{x}}$, which are continuous w.r.t $\mathbf{x}, \boldsymbol{\theta}$, follow the similar steps that can be found in Section D.1 from (22) to (24). Thus, we can further decompose (33) into

$$\boldsymbol{\theta}_{n+1} = \boldsymbol{\theta}_n + \beta_{n+1}\mathbf{h}(\boldsymbol{\theta}_n, \mathbf{x}_n) + \beta_{n+1} \underbrace{(\tilde{H}_{\boldsymbol{\theta}_n,\mathbf{x}_n}(X_{n+1}) - (\mathbf{K}_{\mathbf{x}_n}\tilde{H}_{\boldsymbol{\theta}_n,\mathbf{x}_n})(X_n))}_{M_{n+1}^{(\boldsymbol{\theta})}}$$

$$+ \beta_{n+1} \underbrace{((\mathbf{K}_{\mathbf{x}_{n+1}}\tilde{H}_{\boldsymbol{\theta}_{n+1},\mathbf{x}_{n+1}})(X_{n+1}) - (\mathbf{K}_{\mathbf{x}_n}\tilde{H}_{\boldsymbol{\theta}_n,\mathbf{x}_n})(X_{n+1}))}_{r_n^{(\boldsymbol{\theta},1)}} \tag{35a}$$

$$+ \beta_{n+1} \underbrace{((\mathbf{K}_{\mathbf{x}_n}\tilde{H}_{\boldsymbol{\theta}_n,\mathbf{x}_n})(X_n) - (\mathbf{K}_{\mathbf{x}_{n+1}}\tilde{H}_{\boldsymbol{\theta}_{n+1},\mathbf{x}_{n+1}})(X_{n+1}))}_{r_n^{(\boldsymbol{\theta},2)}},$$

$$\mathbf{x}_{n+1} = \mathbf{x}_n + \gamma_{n+1}(\boldsymbol{\pi}(\mathbf{x}_n) - \mathbf{x}_n) + \gamma_{n+1}\underbrace{(q_{\mathbf{x}_n}(X_{n+1}) - (\mathbf{K}_{\mathbf{x}_n}q_{\mathbf{x}_n})(X_n))}_{M_{n+1}^{(\mathbf{x})}}$$

$$+ \gamma_{n+1}\underbrace{([\mathbf{K}_{\mathbf{x}_n}q_{\mathbf{x}_n}](X_{n+1}) - [\mathbf{K}_{\mathbf{x}_n}q_{\mathbf{x}_{n+1}}](X_{n+1}))}_{r_n^{(\mathbf{x},1)}} \tag{35b}$$

$$+ \gamma_{n+1}\underbrace{((\mathbf{K}_{\mathbf{x}_n}q_{\mathbf{x}_n})(X_n) - (\mathbf{K}_{\mathbf{x}_{n+1}}q_{\mathbf{x}_{n+1}})(X_{n+1}))}_{r_n^{(\mathbf{x},2)}}.$$

such that

$$\boldsymbol{\theta}_{n+1} = \boldsymbol{\theta}_n + \beta_{n+1}\mathbf{h}(\boldsymbol{\theta}_n,\mathbf{x}_n) + \beta_{n+1}M_{n+1}^{(\boldsymbol{\theta})} + \beta_{n+1}r_n^{(\boldsymbol{\theta},1)} + \beta_{n+1}r_n^{(\boldsymbol{\theta},2)}, \tag{36a}$$

$$\mathbf{x}_{n+1} = \mathbf{x}_n + \gamma_{n+1}(\boldsymbol{\pi}(\mathbf{x}_n) - \mathbf{x}_n) + \gamma_{n+1}M_{n+1}^{(\mathbf{x})} + \gamma_{n+1}r_n^{(\mathbf{x},1)} + \gamma_{n+1}r_n^{(\mathbf{x},2)}. \tag{36b}$$

We can observe that (36) differs from the expression in Konda & Tsitsiklis (2004); Mokkadem & Pelletier (2006), which studied the two-timescale SA with Martingale difference noise. Here, due to the presence of the iterate-dependent Markovian noise and the application of the Poisson equation technique, we have additional non-vanishing terms $r_n^{(\boldsymbol{\theta},2)}, r_n^{(\mathbf{x},2)}$, which will be further examined in Lemma E.2. Additionally, when we apply the Poisson equation to the Martingale difference terms $M_{n+1}^{(\boldsymbol{\theta})}, M_{n+1}^{(\mathbf{x})}$, we find that there are some covariances that are also non-vanishing as in Lemma E.1. We will mention this again when we obtain those covariances. These extra non-zero noise terms make our analysis distinct from the previous ones since the key assumption (A4) in Mokkadem & Pelletier (2006) is not satisfied. We demonstrate that the long-term average performance of these terms can be managed so that they do not affect the final CLT result.

**Analysis of Terms** $M_{n+1}^{(\boldsymbol{\theta})}, M_{n+1}^{(\mathbf{x})}$

Consider the filtration $\mathcal{F}_n \triangleq \sigma(\boldsymbol{\theta}_0, \mathbf{x}_0, X_0, \cdots, \boldsymbol{\theta}_n, \mathbf{x}_n, X_n)$, it is evident that $M_{n+1}^{(\boldsymbol{\theta})}, M_{n+1}^{(\mathbf{x})}$ are Martingale difference sequences adapted to $\mathcal{F}_n$. Then, we have

$$\mathbb{E}\left[M_{n+1}^{(\mathbf{x})}(M_{n+1}^{(\mathbf{x})})^T \Big| \mathcal{F}_n\right]$$
$$= \mathbb{E}[q_{\mathbf{x}_n}(X_{n+1})q_{\mathbf{x}_n}(X_{n+1})^T|\mathcal{F}_n] + (\mathbf{K}_{\mathbf{x}_n}q_{\mathbf{x}_n})(X_n)\left((\mathbf{K}_{\mathbf{x}_n}q_{\mathbf{x}_n})(X_n)\right)^T \tag{37}$$
$$- \mathbb{E}[q_{\mathbf{x}_n}(X_{n+1})|\mathcal{F}_n]\left(\mathbf{K}_{\mathbf{x}_n}q_{\mathbf{x}_n})(X_n)\right)^T - (\mathbf{K}_{\mathbf{x}_n}q_{\mathbf{x}_n})(X_n)\mathbb{E}[q_{\mathbf{x}_n}(X_{n+1})^T|\mathcal{F}_n]$$
$$= \mathbb{E}[q_{\mathbf{x}_n}(X_{n+1})q_{\mathbf{x}_n}(X_{n+1})^T|\mathcal{F}_n] - (\mathbf{K}_{\mathbf{x}_n}q_{\mathbf{x}_n})(X_n)\left((\mathbf{K}_{\mathbf{x}_n}q_{\mathbf{x}_n})(X_n)\right)^T.$$

Similarly, we have

$$\mathbb{E}\left[M_{n+1}^{(\boldsymbol{\theta})}(M_{n+1}^{(\boldsymbol{\theta})})^T \Big| \mathcal{F}_n\right]$$
$$= \mathbb{E}[\tilde{H}_{\boldsymbol{\theta}_n,\mathbf{x}_n}(X_{n+1})\tilde{H}_{\boldsymbol{\theta}_n,\mathbf{x}_n}(X_{n+1})^T|\mathcal{F}_n] - (\mathbf{K}_{\mathbf{x}_n}\tilde{H}_{\boldsymbol{\theta}_n,\mathbf{x}_n})(X_n)\left((\mathbf{K}_{\mathbf{x}_n}\tilde{H}_{\boldsymbol{\theta}_n,\mathbf{x}_n})(X_n)\right)^T, \tag{38}$$

and

$$\mathbb{E}\left[M_{n+1}^{(\mathbf{x})}(M_{n+1}^{(\boldsymbol{\theta})})^T \Big| \mathcal{F}_n\right]$$
$$= \mathbb{E}[q_{\mathbf{x}_n}(X_{n+1})\tilde{H}_{\boldsymbol{\theta}_n,\mathbf{x}_n}(X_{n+1})^T|\mathcal{F}_n] - (\mathbf{K}_{\mathbf{x}_n}q_{\mathbf{x}_n})(X_n)\left((\mathbf{K}_{\mathbf{x}_n}\tilde{H}_{\boldsymbol{\theta}_n,\mathbf{x}_n})(X_n)\right)^T.$$

We now focus on $\mathbb{E}\left[M_{n+1}^{(\mathbf{x})}(M_{n+1}^{(\mathbf{x})})^T \Big| \mathcal{F}_n\right]$. Denote by

$$V_1(\mathbf{x},i) \triangleq \sum_{j\in\mathcal{N}}\mathbf{K}_{i,j}[\mathbf{x}]q_{\mathbf{x}}(j)q_{\mathbf{x}}(j)^T - (\mathbf{K}_{\mathbf{x}}q_{\mathbf{x}})(i)\left((\mathbf{K}_{\mathbf{x}}q_{\mathbf{x}})(i)\right)^T, \tag{39}$$

and let its expectation w.r.t the stationary distribution $\boldsymbol{\pi}(\mathbf{x})$ be $v_1(\mathbf{x}) \triangleq \mathbb{E}_{i\sim\boldsymbol{\pi}(\mathbf{x})}[V_1(\mathbf{x},i)]$, we can construct *another Poisson equation*, i.e.,

$$\mathbb{E}\left[M_{n+1}^{(\mathbf{x})}(M_{n+1}^{(\mathbf{x})})^T \Big| \mathcal{F}_n\right] - \sum_{X_n\in\mathcal{N}}\pi_{X_n}(\mathbf{x}_n)\mathbb{E}\left[M_{n+1}^{(\mathbf{x})}(M_{n+1}^{(\mathbf{x})})^T \Big| \mathcal{F}_n\right]$$
$$= V_1(\mathbf{x}_n, X_{n+1}) - v_1(\mathbf{x}_n)$$
$$= \varphi_{\mathbf{x}}^{(1)}(X_{n+1}) - (\mathbf{K}_{\mathbf{x}_n}\varphi_{\mathbf{x}_n}^{(1)})(X_{n+1}),$$

for some matrix-valued function $\varphi_{\mathbf{x}}^{(1)} : \mathcal{N} \to \mathbb{R}^{N \times N}$. Since $q_{\mathbf{x}}$ and $\mathbf{K}[\mathbf{x}]$ are continuous in $\mathbf{x}$, functions $V_1, v_1$ are also continuous in $\mathbf{x}$. Then, we can decompose (39) into

$$
V_1(\mathbf{x}_n, X_{n+1}) = \underbrace{v_1(\boldsymbol{\mu})}_{\mathbf{U}_{22}} + \underbrace{v_1(\mathbf{x}_n) - v_1(\boldsymbol{\mu})}_{\mathbf{D}_n^{(1)}} + \underbrace{\varphi_{\mathbf{x}_n}^{(1)}(X_{n+1}) - (\mathbf{K}_{\mathbf{x}_n}\varphi_{\mathbf{x}_n}^{(1)})(X_n)}_{\mathbf{J}_n^{(1,a)}}
$$
$$
+ \underbrace{(\mathbf{K}_{\mathbf{x}_n}\varphi_{\mathbf{x}_n}^{(1)})(X_n) - (\mathbf{K}_{\mathbf{x}_n}\varphi_{\mathbf{x}_n}^{(1)})(X_{n+1})}_{\mathbf{J}_n^{(1,b)}}.
\tag{40}
$$

Thus, we have

$$
\mathbb{E}[M_{n+1}^{(\mathbf{x})}(M_{n+1}^{(\mathbf{x})})^T | \mathcal{F}_n] = \mathbf{U}_{22} + \mathbf{D}_n^{(1)} + \mathbf{J}_n^{(1)},
\tag{41}
$$

where $\mathbf{J}_n^{(1)} = \mathbf{J}_n^{(1,a)} + \mathbf{J}_n^{(1,b)}$.

Following the similar steps above, we can decompose $\mathbb{E}\left[ M_{n+1}^{(\mathbf{x})}(M_{n+1}^{(\boldsymbol{\theta})})^T \middle| \mathcal{F}_n \right]$ and $\mathbb{E}\left[ M_{n+1}^{(\boldsymbol{\theta})}(M_{n+1}^{(\boldsymbol{\theta})})^T \middle| \mathcal{F}_n \right]$ as

$$
\mathbb{E}\left[ M_{n+1}^{(\mathbf{x})}(M_{n+1}^{(\boldsymbol{\theta})})^T \middle| \mathcal{F}_n \right] = \mathbf{U}_{21} + \mathbf{D}_n^{(2)} + \mathbf{J}_n^{(2)},
\tag{42a}
$$

$$
\mathbb{E}\left[ M_{n+1}^{(\boldsymbol{\theta})}(M_{n+1}^{(\boldsymbol{\theta})})^T \middle| \mathcal{F}_n \right] = \mathbf{U}_{11} + \mathbf{D}_n^{(3)} + \mathbf{J}_n^{(3)}.
\tag{42b}
$$

where $\mathbf{J}_n^{(2)} = \mathbf{J}_n^{(2,a)} + \mathbf{J}_n^{(2,b)}$ and $\mathbf{J}_n^{(3)} = \mathbf{J}_n^{(3,a)} + \mathbf{J}_n^{(3,b)}$. Here, we note that matrices $\mathbf{J}_n^i$ for $i = 1, 2, 3$ are in presence of the current CLT analysis of the two-timescale SA with Martingale difference noise. In addition, $\mathbf{U}_{11}, \mathbf{U}_{12}$ and $\mathbf{U}_{22}$ inherently include the information of the underlying Markov chain (with its eigenpair $(\lambda_i, \mathbf{u}_i)$), which is an extension of the previous works (Konda & Tsitsiklis, 2004; Mokkadem & Pelletier, 2006).

**Lemma E.1.** *For $M_{n+1}^{(\boldsymbol{\theta})}, M_{n+1}^{(\mathbf{x})}$ defined in (35) and their decomposition in (41) and (42), we have*

$$
\mathbf{U}_{11} = \sum_{i=1}^{N-1} \frac{1+\lambda_i}{1-\lambda_i} \mathbf{u}_i \mathbf{u}_i^T, \quad \mathbf{U}_{21} = \sum_{i=1}^{N-1} \frac{1+\lambda_i}{1-\lambda_i} \mathbf{u}_i \mathbf{u}_i^T \mathbf{H}, \quad \mathbf{U}_{11} = \sum_{i=1}^{N-1} \frac{1+\lambda_i}{1-\lambda_i} \mathbf{H}^T \mathbf{u}_i \mathbf{u}_i^T \mathbf{H},
\tag{43a}
$$

$$
\lim_{n\to\infty} \mathbf{D}_n^{(i)} = 0 \ \ a.s. \quad for \quad i = 1, 2, 3,
\tag{43b}
$$

$$
\lim_{n\to\infty} \gamma_n \mathbb{E}\left[ \left\| \sum_{k=1}^n \mathbf{J}_k^{(i)} \right\| \right] = 0, \quad for \quad i = 1, 2, 3.
\tag{43c}
$$

*Proof.* We now provide the properties of the four terms inside (41) as an example. Note that

$$
\mathbf{U}_{11} = \mathbb{E}_{i\sim\boldsymbol{\mu}}[V_1(\boldsymbol{\mu}, i)] = \sum_{i\in\mathcal{N}} \mu_i \left[ \sum_{j\in\mathcal{N}} \mathbf{P}(i,j) q_{\boldsymbol{\mu}}(j) q_{\boldsymbol{\mu}}(j)^T - (\mathbf{P}q_{\boldsymbol{\mu}})(i)\left((\mathbf{P}q_{\boldsymbol{\mu}})(i)\right)^T \right]
$$
$$
= \sum_{j\in\mathcal{N}} \mu_j q_{\boldsymbol{\mu}}(j) q_{\boldsymbol{\mu}}(j)^T - (\mathbf{P}q_{\boldsymbol{\mu}})(j)\left((\mathbf{P}q_{\boldsymbol{\mu}})(j)\right)^T.
$$

We can see that it has exactly the same structure as matrix $U$ in (27). Following the similar steps in deducing the explicit form of $U$ from (28) to (30), we get

$$
\mathbf{U}_{11} = \sum_{i=1}^{N-1} \frac{1+\lambda_i}{1-\lambda_i} \mathbf{u}_i \mathbf{u}_i^T.
\tag{44}
$$

By the almost sure convergence result $\mathbf{x}_n \to \boldsymbol{\mu}$ in Lemma 3.1, $v_1(\mathbf{x}_n) \to v_1(\boldsymbol{\mu})$ a.s. such that $\lim_{n\to\infty} \mathbf{D}_n^{(1)} = 0$ a.s.

We next prove that $\lim_{n\to\infty} \gamma_n \mathbb{E}\left[ \left\| \sum_{k=1}^n \mathbf{J}_k^{(1,a)} \right\| \right] = 0$ and $\lim_{n\to\infty} \gamma_n \mathbb{E}\left[ \left\| \sum_{k=1}^n \mathbf{J}_k^{(1,b)} \right\| \right] = 0$.

Since $\{\mathbf{J}_n^{(1,a)}\}$ is a Martingale difference sequence adapted to $\mathcal{F}_n$, with the Burkholder inequality in Lemma G.2 and $p = 1$, we show that

$$\mathbb{E}\left[\left\|\sum_{k=1}^{n}\mathbf{J}_k^{(1,a)}\right\|\right] \leq C_1\mathbb{E}\left[\sqrt{\left(\sum_{k=1}^{n}\left\|\mathbf{J}_k^{(1,a)}\right\|^2\right)}\right]. \tag{45}$$

By assumption A4, $\mathbf{x}_n$ is always within some compact set $\Omega$ such that $\sup_n\|\mathbf{J}_n^{(1,a)}\| \leq C_\Omega < \infty$ and for a given trajectory $\omega$ of $\mathbf{x}_n(\omega)$,

$$\gamma_n C_p\sqrt{\left(\sum_{k=1}^{n}\left\|\mathbf{J}_k^{(1,a)}\right\|^2\right)} \leq C_p C_\Omega \gamma_n\sqrt{n}, \tag{46}$$

and the last term decreases to zero in $n$ since $a > 1/2$.

For $\mathbf{J}_n^{(1,b)}$, we use Abel transformation and obtain

$$\sum_{k=1}^{n}\mathbf{J}_k^{(1,b)} = \sum_{k=1}^{n}((\mathbf{K}_{\mathbf{x}_k}\varphi_{\mathbf{x}_k}^{(1)})(X_{k-1}) - (\mathbf{K}_{\mathbf{x}_{k-1}}\varphi_{\mathbf{x}_{k-1}}^{(1)})(X_{k-1}))$$
$$+ (\mathbf{K}_{\mathbf{x}_0}\varphi_{\mathbf{x}_0}^{(1)})(X_0) - (\mathbf{K}_{\mathbf{x}_n}\varphi_{\mathbf{x}_n}^{(1)})(X_n).$$

Since $(\mathbf{K}_{\mathbf{x}}\varphi_{\mathbf{x}}^{(1)})(X)$ is continuous in $\mathbf{x}$, for $\mathbf{x}_n$ within a compact set $\Omega$ (assumption A4), it is local Lipschitz with a constant $L_\Omega$ such that

$$\|(\mathbf{K}_{\mathbf{x}_k}\varphi_{\mathbf{x}_k}^{(1)})(X_{k-1}) - \mathbf{K}_{\mathbf{x}_{k-1}}\varphi_{\mathbf{x}_{k-1}}^{(1)})(X_{k-1})\| \leq L_\Omega\|\mathbf{x}_k - \mathbf{x}_{k-1}\| \leq 2L_\Omega\gamma_k.$$

where the last inequality arises from (4b), i.e., $\mathbf{x}_k - \mathbf{x}_{k-1} = \gamma_k(\boldsymbol{\delta}_{X_k} - \mathbf{x}_{k-1})$ and $\|\boldsymbol{\delta}_{X_k} - \mathbf{x}_{k-1}\| \leq 2$ because $\mathbf{x}_n \in \mathrm{Int}(\Sigma)$. Also, $\|(\mathbf{K}_{\mathbf{x}_0}\varphi_{\mathbf{x}_0}^{(1)})(X_0)\| + \|(\mathbf{K}_{\mathbf{x}_n}\varphi_{\mathbf{x}_n}^{(1)})(X_n)\|$ are upper-bounded by some positive constant $C_\Omega'$. This implies that

$$\left\|\sum_{k=1}^{n}\mathbf{J}_k^{(1,b)}\right\| \leq C_\Omega' + 2L_\Omega\sum_{k=1}^{n}\gamma_k.$$

Note that

$$\gamma_n\left\|\sum_{k=1}^{n}\mathbf{J}_k^{(1,b)}\right\| \leq \gamma_n C_\Omega' + 2L_\Omega\gamma_n\sum_{k=1}^{n}\gamma_k \leq \gamma_n C_\Omega' + \frac{2L_\Omega}{a}n^{1-2a}, \tag{47}$$

where the last inequality is from $\sum_{k=1}^{n}\gamma_k < \frac{1}{a}n^{1-a}$. We observe that the last term in (47) is decreasing to zero in $n$ because $a > 1/2$.

Note that $\mathbf{J}_k^{(1)} = \mathbf{J}_k^{(1,a)} + \mathbf{J}_k^{(1,b)}$, by triangular inequality we have

$$\gamma_n\mathbb{E}\left[\left\|\sum_{k=1}^{n}\mathbf{J}_k^{(11)}\right\|\right] \leq \gamma_n\mathbb{E}\left[\left\|\sum_{k=1}^{n}\mathbf{J}_k^{(11,A)}\right\|\right] + \gamma_n\mathbb{E}\left[\left\|\sum_{k=1}^{n}\mathbf{J}_k^{(11,B)}\right\|\right]$$

$$\leq \gamma_n C_1\mathbb{E}\left[\sqrt{\left(\sum_{k=1}^{n}\left\|\mathbf{J}_k^{(11,A)}\right\|^2\right)}\right] + \gamma_n\mathbb{E}\left[\left\|\sum_{k=1}^{n}\mathbf{J}_k^{(11,B)}\right\|\right] \tag{48}$$

$$= \mathbb{E}\left[\gamma_n C_1\sqrt{\left(\sum_{k=1}^{n}\left\|\mathbf{J}_k^{(11,A)}\right\|^2\right)} + \gamma_n\left\|\sum_{k=1}^{n}\mathbf{J}_k^{(11,B)}\right\|\right],$$

where the second inequality comes from (45). By (46) and (47) we know that both terms in the last line of (48) are uniformly bounded by constants over time $n$ that depend on the set $\Omega$. Therefore, by dominated convergence theorem, taking the limit over the last line of (48) gives

$$\lim_{n\to\infty}\mathbb{E}\left[\gamma_n C_1\sqrt{\left(\sum_{k=1}^{n}\left\|\mathbf{J}_k^{(11,A)}\right\|^2\right)} + \gamma_n\left\|\sum_{k=1}^{n}\mathbf{J}_k^{(11,B)}\right\|\right]$$

$$= \mathbb{E}\left[\lim_{n\to\infty}\gamma_n C_1\sqrt{\left(\sum_{k=1}^{n}\left\|\mathbf{J}_k^{(11,A)}\right\|^2\right)} + \gamma_n\left\|\sum_{k=1}^{n}\mathbf{J}_k^{(11,B)}\right\|\right] = 0.$$

Therefore, we have

$$\lim_{n\to\infty} \gamma_n \mathbb{E}\left[\left\|\sum_{k=1}^n \mathbf{J}_k^{(1)}\right\|\right] = 0,$$

In sum, in terms of $\mathbb{E}[M_{n+1}^{(\mathbf{x})}(M_{n+1}^{(\mathbf{x})})^T | \mathcal{F}_n]$ in (41), we have $\mathbf{U}_{11}$ in (44), $\lim_{n\to\infty} \mathbf{D}_n^{(1)} = 0$ a.s. and $\lim_{n\to\infty} \gamma_n \mathbb{E}\left[\left\|\sum_{k=1}^n \mathbf{J}_k^{(1)}\right\|\right] = 0$.

We can apply the same steps as above for the other two terms $i = 2, 3$ in (42) and obtain the results. □

**Analysis of Terms $r_n^{(\boldsymbol{\theta},1)}, r_n^{(\boldsymbol{\theta},2)}, r_n^{(\mathbf{x},1)}, r_n^{(\mathbf{x},2)}$**

**Lemma E.2.** *For $r_n^{(\boldsymbol{\theta},1)}, r_n^{(\boldsymbol{\theta},2)}, r_n^{(\mathbf{x},1)}, r_n^{(\mathbf{x},2)}$ defined in (35), we have the following results:*

$$\|r_n^{(\boldsymbol{\theta},1)}\| = O(\gamma_n) = o(\sqrt{\beta_n}), \quad \sqrt{\gamma_n}\left\|\sum_{k=1}^n r_k^{(\boldsymbol{\theta},2)}\right\| = O(\sqrt{\gamma_n}) = o(1). \tag{49a}$$

$$\|r_n^{(\mathbf{x},1)}\| = O(\gamma_n) = o(\sqrt{\beta_n}), \quad \sqrt{\gamma_n}\left\|\sum_{k=1}^n r_k^{(\mathbf{x},2)}\right\| = O(\sqrt{\gamma_n}) = o(1). \tag{49b}$$

*Proof.* For $r_n^{(\boldsymbol{\theta},1)}$, note that

$$
\begin{aligned}
r_n^{(\boldsymbol{\theta},1)} &= (\mathbf{K}_{\mathbf{x}_{n+1}} \tilde{H}_{\boldsymbol{\theta}_{n+1},\mathbf{x}_{n+1}})(X_{n+1}) - (\mathbf{K}_{\mathbf{x}_n} \tilde{H}_{\boldsymbol{\theta}_n,\mathbf{x}_n})(X_{n+1}) \\
&= \sum_{j\in\mathcal{N}} \left( \mathbf{K}_{X_n,j}[\mathbf{x}_{n+1}] \tilde{H}_{\boldsymbol{\theta}_{n+1},\mathbf{x}_{n+1}}(j) - \mathbf{K}_{X_n,j}[\mathbf{x}_n] \tilde{H}_{\boldsymbol{\theta}_n,\mathbf{x}_n}(j) \right) \\
&\leq \sum_{j\in\mathcal{N}} L_{\mathcal{C}}(\|\boldsymbol{\theta}_{n+1} - \boldsymbol{\theta}_n\| + \|\mathbf{x}_{n+1} - \mathbf{x}_n\|) \\
&\leq N L_{\mathcal{C}}(C_{\mathcal{C}}\beta_{n+1} + 2\gamma_{n+1})
\end{aligned}
\tag{50}
$$

where the second last inequality is because $\mathbf{K}_{i,j}[\mathbf{x}]\tilde{H}_{\boldsymbol{\theta},\mathbf{x}}(j)$ is continuous in $\boldsymbol{\theta}, \mathbf{x}$ $\mathbf{K}[\boldsymbol{x}]$, which stems from continuous functions $\boldsymbol{K}[\mathbf{x}]$ and $\tilde{H}_{\boldsymbol{\theta},\mathbf{x}}$. The last inequality is from update rules (4) and $(\boldsymbol{\theta}_n, \mathbf{x}_n) \in \Omega$ for some compact subset $\Omega$ by assumption A4. Then, we have $\|r_n^{(\boldsymbol{\theta},1)}\| = O(\gamma_n) = o(\sqrt{\beta_n})$ because of $a > 1/2 \geq b/2$ by assumption A2.

We let $\nu_n \triangleq (\mathbf{K}_{\mathbf{x}_n} \tilde{H}_{\boldsymbol{\theta}_n,\mathbf{x}_n})(X_n)$ such that $r_n^{(\boldsymbol{\theta},2)} = \nu_n - \nu_{n+1}$. Note that $\sum_{k=1}^n r_k^{(\boldsymbol{\theta},2)} = \nu_1 - \nu_{n+1}$, and by assumption A4, $\|\nu_n\|$ is upper bounded by a constant dependent on the compact set, which leads to

$$\sqrt{\gamma_n}\left\|\sum_{k=1}^n r_k^{(\boldsymbol{\theta},2)}\right\| = \sqrt{\gamma_n}\|\nu_1 - \nu_{n+1}\| = O(\sqrt{\gamma_n}) = o(1).$$

Similarly, we can also obtain $\|r_n^{(\mathbf{x},1)}\| = o(\sqrt{\beta_n})$ and $\sqrt{\gamma_n}\left\|\sum_{k=1}^n r_k^{(\mathbf{x},2)}\right\| = O(\sqrt{\gamma_n}) = o(1)$. □

### E.2.2 EFFECT OF SRRW ITERATION ON SA ITERATION

In view of the almost sure convergence results in Lemma 3.1 and Lemma 3.2, for large enough $n$ so that both iterations $\boldsymbol{\theta}_n, \mathbf{x}_n$ are close to the equilibrium $(\boldsymbol{\theta}^*, \boldsymbol{\mu})$, we can apply the Taylor expansion to functions $\mathbf{h}(\boldsymbol{\theta}, \mathbf{x})$ and $\boldsymbol{\pi}[\mathbf{x}] - \mathbf{x}$ in (36) at the point $(\boldsymbol{\theta}^*, \boldsymbol{\mu})$, which results in

$$\mathbf{h}(\boldsymbol{\theta}, \mathbf{x}) = \mathbf{h}(\boldsymbol{\theta}^*, \boldsymbol{\mu}) + \nabla_{\boldsymbol{\theta}}\mathbf{h}(\boldsymbol{\theta}^*, \boldsymbol{\mu})(\boldsymbol{\theta} - \boldsymbol{\theta}^*) + \nabla_{\mathbf{x}}\mathbf{h}(\boldsymbol{\theta}^*, \boldsymbol{\mu})(\mathbf{x} - \boldsymbol{\mu}) + O(\|\boldsymbol{\theta} - \boldsymbol{\theta}^*\|^2 + \|\mathbf{x} - \boldsymbol{\mu}\|^2), \tag{51a}$$

$$\boldsymbol{\pi}[\mathbf{x}] - \mathbf{x} = \boldsymbol{\pi}[\boldsymbol{\mu}] - \boldsymbol{\mu} + \nabla_{\mathbf{x}}(\boldsymbol{\pi}(\mathbf{x}) - \mathbf{x})|_{\mathbf{x}=\boldsymbol{\mu}}(\mathbf{x} - \boldsymbol{\mu}) + O(\|\mathbf{x} - \boldsymbol{\mu}\|^2). \tag{51b}$$

With matrix $\mathbf{J}(\alpha)$, we have the following:

$$
\begin{aligned}
\mathbf{J}_{11} &= \nabla_{\boldsymbol{\theta}}\mathbf{h}(\boldsymbol{\theta}^*, \boldsymbol{\mu}) = \nabla\mathbf{h}(\boldsymbol{\theta}^*), \\
\mathbf{J}_{12}(\alpha) &= \nabla_{\mathbf{x}}\mathbf{h}(\boldsymbol{\theta}^*, \boldsymbol{\mu}) = -\alpha\mathbf{H}^T(\mathbf{P}^T + \mathbf{I}), \\
\mathbf{J}_{22}(\alpha) &= \nabla_{\mathbf{x}}(\boldsymbol{\pi}(\mathbf{x}) - \mathbf{x})|_{\mathbf{x}=\boldsymbol{\mu}} = 2\alpha\boldsymbol{\mu}\mathbf{1}^T - \alpha\mathbf{P}^T - (\alpha + 1)\mathbf{I}.
\end{aligned}
\tag{52}
$$

Then, (36) becomes

$$\boldsymbol{\theta}_{n+1} = \boldsymbol{\theta}_n + \beta_{n+1}(\mathbf{J}_{11}(\boldsymbol{\theta}_n - \boldsymbol{\theta}^*) + \mathbf{J}_{12}(\alpha)(\mathbf{x}_n - \boldsymbol{\mu}) + r_n^{(\boldsymbol{\theta},1)} + r_n^{(\boldsymbol{\theta},2)} + M_{n+1}^{(\boldsymbol{\theta})} + \eta_n^{(\boldsymbol{\theta})}), \quad (53a)$$

$$\mathbf{x}_{n+1} = \mathbf{x}_n + \gamma_{n+1}(\mathbf{J}_{22}(\alpha)(\mathbf{x}_n - \boldsymbol{\mu}) + r_n^{(\mathbf{x},1)} + r_n^{(\mathbf{x},2)} + M_{n+1}^{(\mathbf{x})} + \eta_n^{(\mathbf{x})}), \quad (53b)$$

where $\eta_n^{(\boldsymbol{\theta})} = O(\|\mathbf{x}_n\|^2 + \|\boldsymbol{\theta}_n\|^2)$ and $\eta_n^{(\mathbf{x})} = O(\|\mathbf{x}_n\|^2)$.

Then, inspired by Mokkadem & Pelletier (2006), we decompose iterates $\{\mathbf{x}_n\}$ and $\{\boldsymbol{\theta}_n\}$ into $\mathbf{x}_n = L_n^{(\boldsymbol{x})} + \Delta_n^{(\boldsymbol{x})}$ and $\boldsymbol{\theta}_n = L_n^{(\boldsymbol{\theta})} + R_n^{(\boldsymbol{\theta})} + \Delta_n^{(\boldsymbol{\theta})}$. Rewriting (53b) gives

$$\mathbf{x}_n - \boldsymbol{\mu} = \gamma_{n+1}^{-1}\mathbf{J}_{22}(\alpha)^{-1}(\mathbf{x}_{n+1} - \mathbf{x}_n) - \mathbf{J}_{22}(\alpha)^{-1}(r_n^{(\mathbf{x},1)} + r_n^{(\mathbf{x},2)} + M_{n+1}^{(\mathbf{x})} + \eta_n^{(\mathbf{x})}),$$

and substituting the above equation back in (53a) gives

$$
\begin{aligned}
\boldsymbol{\theta}_{n+1} - \boldsymbol{\theta}^* = \ & \boldsymbol{\theta}_n - \boldsymbol{\theta}^* + \beta_{n+1}\bigg(\mathbf{J}_{11}(\boldsymbol{\theta}_n - \boldsymbol{\theta}^*) + \gamma_{n+1}^{-1}\mathbf{J}_{12}(\alpha)\mathbf{J}_{22}(\alpha)^{-1}(\mathbf{x}_{n+1} - \mathbf{x}_n) \\
& - \mathbf{J}_{12}(\alpha)\mathbf{J}_{22}(\alpha)^{-1}(r_n^{(\mathbf{x},1)} + r_n^{(\mathbf{x},2)} + M_{n+1}^{(\mathbf{x})} + \eta_n^{(\mathbf{x})}) + r_n^{(\boldsymbol{\theta},1)} + r_n^{(\boldsymbol{\theta},2)} + M_{n+1}^{(\boldsymbol{\theta})} + \eta_n^{(\boldsymbol{\theta})}\bigg) \\
= \ & (\mathbf{I} + \beta_{n+1}\mathbf{J}_{11})(\boldsymbol{\theta}_n - \boldsymbol{\theta}^*) + [\beta_{n+1}\gamma_{n+1}^{-1}\mathbf{J}_{12}(\alpha)\mathbf{J}_{22}(\alpha)^{-1}(\mathbf{x}_{n+1} - \mathbf{x}_n)] \\
& + \beta_{n+1}(M_{n+1}^{(\boldsymbol{\theta})} - \mathbf{J}_{12}(\alpha)\mathbf{J}_{22}(\alpha)^{-1}M_{n+1}^{(\mathbf{x})}) \\
& + \beta_{n+1}(r_n^{(\boldsymbol{\theta},1)} + r_n^{(\boldsymbol{\theta},2)} + \eta_n^{(\boldsymbol{\theta})} - \mathbf{J}_{12}(\alpha)\mathbf{J}_{22}(\alpha)^{-1}(r_n^{(\mathbf{x},1)} + r_n^{(\mathbf{x},2)} + \eta_n^{(\mathbf{x})})),
\end{aligned}
$$
$$(54)$$

From (54) we can see the iteration $\{\boldsymbol{\theta}_n\}$ implicitly embeds the recursions of three sequences

- $\beta_{n+1}\gamma_{n+1}^{-1}\mathbf{J}_{12}(\alpha)\mathbf{J}_{22}(\alpha)^{-1}(\mathbf{x}_{n+1} - \mathbf{x}_n)$;
- $\beta_{n+1}(M_{n+1}^{(\boldsymbol{\theta})} - \mathbf{J}_{12}(\alpha)\mathbf{J}_{22}(\alpha)^{-1}M_{n+1}^{(\mathbf{x})})$;
- $\beta_{n+1}(r_n^{(\boldsymbol{\theta},1)} + r_n^{(\boldsymbol{\theta},2)} + \eta_n^{(\boldsymbol{\theta})} - \mathbf{J}_{12}(\alpha)\mathbf{J}_{22}(\alpha)^{-1}(r_n^{(\mathbf{x},1)} + r_n^{(\mathbf{x},2)} + \eta_n^{(\mathbf{x})})))$.

Let $u_n \triangleq \sum_{k=1}^n \beta_k$ and $s_n \triangleq \sum_{k=1}^n \gamma_k$. Below we define two iterations:

$$
\begin{aligned}
L_n^{(\boldsymbol{\theta})} &= e^{\beta_n \mathbf{J}_{11}} L_{n-1}^{(\boldsymbol{\theta})} + \beta_n(M_n^{(\boldsymbol{\theta})} - \mathbf{J}_{12}(\alpha)\mathbf{J}_{22}(\alpha)^{-1}M_n^{(\mathbf{x})}) \\
&= \sum_{k=1}^n e^{(u_n - u_k)\mathbf{J}_{11}}\beta_k(M_k^{(\boldsymbol{\theta})} - \mathbf{J}_{12}(\alpha)\mathbf{J}_{22}(\alpha)^{-1}M_k^{(\mathbf{x})})
\end{aligned}
\tag{55a}
$$

$$
\begin{aligned}
R_n^{(\boldsymbol{\theta})} &= e^{\beta_n \mathbf{J}_{11}} R_{n-1}^{(\boldsymbol{\theta})} + \beta_n \gamma_n^{-1}\mathbf{J}_{12}(\alpha)\mathbf{J}_{22}(\alpha)^{-1}(\mathbf{x}_n - \mathbf{x}_{n-1}) \\
&= \sum_{k=1}^n e^{(u_n - u_k)\mathbf{J}_{11}}\beta_k \gamma_k^{-1}\mathbf{J}_{12}(\alpha)\mathbf{J}_{22}(\alpha)^{-1}(\mathbf{x}_k - \mathbf{x}_{k-1})
\end{aligned}
\tag{55b}
$$

and a remaining term $\Delta_n^{(\boldsymbol{\theta})} \triangleq \boldsymbol{\theta}_n - \boldsymbol{\theta}^* - L_n^{(\boldsymbol{\theta})} - R_n^{(\boldsymbol{\theta})}$.

Similarly, for iteration $\mathbf{x}_n$, define the sequence $L_n^{(\mathbf{x})}$ such that

$$L_n^{(\mathbf{x})} = e^{\gamma_n \mathbf{J}_{22}(\alpha)} L_{n-1}^{(\mathbf{x})} + \gamma_n M_n^{(\mathbf{x})} = \sum_{k=1}^n e^{(s_n - s_k)\mathbf{J}_{22}(\alpha)}\gamma_k M_k^{(\mathbf{x})}, \tag{56}$$

and a remaining term

$$\Delta_n^{(\mathbf{x})} \triangleq \mathbf{x}_n - \boldsymbol{\mu} - L_n^{(\boldsymbol{\theta})} \tag{57}$$

The decomposition of $\boldsymbol{\theta}_n - \boldsymbol{\theta}^*$ and $\mathbf{x}_n - \boldsymbol{\mu}$ in the above form is also standard in the single-timescale SA literature (Delyon, 2000; Fort, 2015).

**Characterization of Sequences $\{L_n^{(\boldsymbol{\theta})}\}$ and $\{L_n^{(\mathbf{x})}\}$**

we set a Martingale $Z^{(n)} = \{Z_k^{(n)}\}_{k \geq 1}$ such that

$$Z_k^{(n)} = \begin{pmatrix} \beta_n^{-1/2}e^{u_n \mathbf{J}_{11}} & 0 \\ 0 & \gamma_n^{-1/2}e^{s_n \mathbf{J}_{22}(\alpha)} \end{pmatrix} \times \sum_{j=1}^k \begin{pmatrix} e^{-u_k \mathbf{J}_{11}}\beta_k(M_k^{(\boldsymbol{\theta})} - \mathbf{J}_{12}(\alpha)\mathbf{J}_{22}(\alpha)^{-1}M_k^{(\mathbf{x})}) \\ e^{-s_k \mathbf{J}_{22}(\alpha)}\gamma_k M_k^{(\mathbf{x})} \end{pmatrix}.$$

Then, the Martingale difference array $Z_k^{(n)} - Z_{k-1}^{(n)}$ becomes

$$Z_k^{(n)} - Z_{k-1}^{(n)} = \begin{pmatrix} \beta_n^{-1/2} e^{(u_n - u_k)\mathbf{J}_{11}} \beta_k (M_k^{(\boldsymbol{\theta})} - \mathbf{J}_{12}(\alpha)\mathbf{J}_{22}(\alpha)^{-1} M_k^{(\mathbf{x})}) \\ \gamma_n^{-1/2} e^{(s_n - s_k)\mathbf{J}_{22}(\alpha)} \gamma_k M_k^{(\mathbf{x})} \end{pmatrix}$$

and

$$\sum_{k=1}^n \mathbb{E}\left[ (Z_k^{(n)} - Z_{k-1}^{(n)})(Z_k^{(n)} - Z_{k-1}^{(n)})^T | \mathcal{F}_{k-1} \right] = \begin{pmatrix} A_{1,n} & A_{2,n} \\ A_{2,n}^T & A_{4,n} \end{pmatrix},$$

where, in view of decomposition of $M_n^{(\boldsymbol{\theta})}$ and $M_n^{(\mathbf{x})}$ in (41) and (42), respectively,

$$A_{1,n} = \beta_n^{-1} \sum_{k=1}^n \beta_k^2 e^{(u_n - u_k)\mathbf{J}_{11}} \bigg( \mathbf{U}_{22} + \mathbf{D}_k^{(1)} + \mathbf{J}_k^{(1)} - (\mathbf{U}_{21} + \mathbf{D}_k^{(2)} + \mathbf{J}_k^{(2)})(\mathbf{J}_{12}(\alpha)\mathbf{J}_{22}(\alpha)^{-1})^T$$

$$+ \mathbf{J}_{12}(\alpha)\mathbf{J}_{22}(\alpha)^{-1}(\mathbf{U}_{11} + \mathbf{D}_k^{(3)} + \mathbf{J}_k^{(3)})(\mathbf{J}_{12}(\alpha)\mathbf{J}_{22}(\alpha)^{-1})^T$$

$$- \mathbf{J}_{12}(\alpha)\mathbf{J}_{22}(\alpha)^{-1}(\mathbf{U}_{21} + \mathbf{D}_k^{(2)} + \mathbf{J}_k^{(2)})^T \bigg) e^{(u_n - u_k)(\mathbf{J}_{11})^T},$$

$$\tag{58a}$$

$$A_{2,n} = \beta_n^{-1/2}\gamma_n^{-1/2} \sum_{k=1}^n \beta_k \gamma_k e^{(u_n - u_k)\mathbf{J}_{11}}(\mathbf{U}_{21} - \mathbf{J}_{12}(\alpha)\mathbf{J}_{22}(\alpha)^{-1}\mathbf{U}_{11})e^{(s_n - s_k)\mathbf{J}_{22}(\alpha)^T}, \tag{58b}$$

$$A_{4,n} = \gamma_n^{-1} \sum_{k=1}^n \gamma_k^2 e^{(s_n - s_k)\mathbf{J}_{22}(\alpha)}(\mathbf{U}_{11} + \mathbf{D}_k^{(3)} + \mathbf{J}_k^{(3)})e^{(s_n - s_k)\mathbf{J}_{22}(\alpha)^T}. \tag{58c}$$

We further decompose $A_{1,n}$ into three parts:

$$A_{1,n} = \beta_n^{-1} \sum_{k=1}^n \bigg( \beta_k^2 e^{(u_n - u_k)\mathbf{J}_{11}}(\mathbf{U}_{22} - \mathbf{U}_{21}(\mathbf{J}_{12}(\alpha)\mathbf{J}_{22}(\alpha)^{-1})^T$$

$$- \mathbf{J}_{12}(\alpha)\mathbf{J}_{22}(\alpha)^{-1}\mathbf{U}_{12} + \mathbf{J}_{12}(\alpha)\mathbf{J}_{22}(\alpha)^{-1}\mathbf{U}_{11}(\mathbf{J}_{12}(\alpha)\mathbf{J}_{22}(\alpha)^{-1})^T)e^{(u_n - u_k)(\mathbf{J}_{11})^T} \bigg)$$

$$+ \beta_n^{-1} \sum_{k=1}^n \bigg( \beta_k^2 e^{(u_n - u_k)\mathbf{J}_{11}}(\mathbf{D}_k^{(1)} + \mathbf{J}_{12}(\alpha)\mathbf{J}_{22}(\alpha)^{-1}\mathbf{D}_k^{(3)}(\mathbf{J}_{12}(\alpha)\mathbf{J}_{22}(\alpha)^{-1})^T$$

$$- \mathbf{D}_k^{(2)}(\mathbf{J}_{12}(\alpha)\mathbf{J}_{22}(\alpha)^{-1})^T - \mathbf{J}_{12}(\alpha)\mathbf{J}_{22}(\alpha)^{-1}(\mathbf{D}_k^{(2)})^T)e^{(u_n - u_k)(\mathbf{J}_{11})^T} \bigg)$$

$$+ \beta_n^{-1} \sum_{k=1}^n \bigg( \beta_k^2 e^{(u_n - u_k)\mathbf{J}_{11}}(\mathbf{J}_k^{(1)} + \mathbf{J}_{12}(\alpha)\mathbf{J}_{22}(\alpha)^{-1}\mathbf{J}_k^{(3)}(\mathbf{J}_{12}(\alpha)\mathbf{J}_{22}(\alpha)^{-1})^T$$

$$- \mathbf{J}_k^{(2)}(\mathbf{J}_{12}(\alpha)\mathbf{J}_{22}(\alpha)^{-1})^T - \mathbf{J}_{12}(\alpha)\mathbf{J}_{22}(\alpha)^{-1}(\mathbf{J}_k^{(2)})^T)e^{(u_n - u_k)(\mathbf{J}_{11})^T} \bigg)$$

$$\triangleq A_{1,n}^{(a)} + A_{1,n}^{(b)} + A_{1,n}^{(c)}.$$

$$\tag{59}$$

Here, we define $\mathbf{U}_{\boldsymbol{\theta}}(\alpha) \triangleq \mathbf{U}_{22} - \mathbf{U}_{21}(\mathbf{J}_{12}(\alpha)\mathbf{J}_{22}(\alpha)^{-1})^T - \mathbf{J}_{12}(\alpha)\mathbf{J}_{22}(\alpha)^{-1}\mathbf{U}_{12} + \mathbf{J}_{12}(\alpha)\mathbf{J}_{22}(\alpha)^{-1}\mathbf{U}_{11}(\mathbf{J}_{12}(\alpha)\mathbf{J}_{22}(\alpha)^{-1})^T$. By (52) and (43a) in Lemma E.1, we have

$$\mathbf{U}_{\boldsymbol{\theta}}(\alpha) = \sum_{i=1}^{N-1} \frac{1}{(\alpha(1 + \lambda_i) + 1)^2} \cdot \frac{1 + \lambda_i}{1 - \lambda_i} \mathbf{H}^T \mathbf{u}_i \mathbf{u}_i^T \mathbf{H}. \tag{60}$$

Then, we have the following lemma.

**Lemma E.3.** *For $A_{1,n}^{(a)}, A_{1,n}^{(b)}, A_{1,n}^{(c)}$ defined in (59), we have*

$$\lim_{n\to\infty} A_{1,n}^{(a)} = \mathbf{V}_{\boldsymbol{\theta}}(\alpha), \quad \lim_{n\to\infty} \|A_{1,n}^{(b)}\| = 0, \quad \lim_{n\to\infty} \|A_{1,n}^{(c)}\| = 0, \tag{61}$$

*where $\mathbf{V}_{\boldsymbol{\theta}}(\alpha)$ is the solution to the Lyapunov equation*

$$\left( \mathbf{J}_{11} + \frac{\mathbb{1}_{\{b=1\}}}{2}\boldsymbol{I} \right)\mathbf{V}_{\boldsymbol{\theta}}(\alpha) + \mathbf{V}_{\boldsymbol{\theta}}(\alpha)\left( \mathbf{J}_{11} + \frac{\mathbb{1}_{\{b=1\}}}{2}\boldsymbol{I} \right)^T + \mathbf{U}_{\boldsymbol{\theta}}(\alpha) = 0.$$

*Proof.* First, from Lemma G.4, we have for some $c, T > 0$ such that

$$\|A_{1,n}^{(b)}\| \leq \beta_n^{-1} \sum_{k=1}^{n} \left\| \mathbf{D}_k^{(1)} + \mathbf{J}_{12}(\alpha)\mathbf{J}_{22}(\alpha)^{-1}\mathbf{D}_k^{(3)}(\mathbf{J}_{12}(\alpha)\mathbf{J}_{22}(\alpha)^{-1})^T - \mathbf{D}_k^{(2)}(\mathbf{J}_{12}(\alpha)\mathbf{J}_{22}(\alpha)^{-1})^T \right.$$
$$\left. - \mathbf{J}_{12}(\alpha)\mathbf{J}_{22}(\alpha)^{-1}(\mathbf{D}_k^{(2)})^T \right\| \cdot \beta_k^2 c^2 e^{-2T(u_n - u_k)}.$$

Applying Lemma G.6, together with $\mathbf{D}_n^{(i)} \to 0$ a.s. in Lemma E.1, gives

$$\limsup_n \|A_{1,n}^{(b)}\|$$
$$\leq \frac{1}{C(b,p)} \limsup_n \|(\mathbf{D}_n^{(1)} + \mathbf{J}_{12}(\alpha)\mathbf{J}_{22}(\alpha)^{-1}\mathbf{D}_n^{(4)}(\mathbf{J}_{12}(\alpha)\mathbf{J}_{22}(\alpha)^{-1})^T$$
$$- \mathbf{D}_n^{(2)}(\mathbf{J}_{12}(\alpha)\mathbf{J}_{22}(\alpha)^{-1})^T - \mathbf{J}_{12}(\alpha)\mathbf{J}_{22}(\alpha)^{-1}\mathbf{D}_n^{(3)})\|$$
$$= 0.$$

We now consider $\|A_{1,n}^{(c)}\|$. Set

$$\Xi_n \triangleq \sum_{k=1}^{n} \left( \mathbf{J}_n^{(1)} + \mathbf{J}_{12}(\alpha)\mathbf{J}_{22}(\alpha)^{-1}\mathbf{J}_k^{(3)}(\mathbf{J}_{12}(\alpha)\mathbf{J}_{22}(\alpha)^{-1})^T \right.$$
$$\left. - \mathbf{J}_k^{(2)}(\mathbf{J}_{12}(\alpha)\mathbf{J}_{22}(\alpha)^{-1})^T - \mathbf{J}_{12}(\alpha)\mathbf{J}_{22}(\alpha)^{-1}(\mathbf{J}_k^{(2)})^T \right),$$

we can rewrite $A_{1,n}^{(c)}$ as

$$A_{1,n}^{(c)} = \beta_n^{-1} \sum_{k=1}^{n} \beta_k^2 e^{(u_n - u_k)\mathbf{J}_{11}} (\Xi_k - \Xi_{k-1}) e^{(u_n - u_k)(\mathbf{J}_{11})^T}.$$

By the Abel transformation, we have

$$A_{1,n}^{(c)} = \beta_n \Xi_n + \beta_n^{-1} \sum_{k=1}^{n-1} \left[ \beta_k^2 e^{(u_n - u_k)\mathbf{J}_{11}} \Xi_k e^{(u_n - u_k)(\mathbf{J}_{11})^T} \right.$$
$$\left. - \beta_{k+1}^2 e^{(u_n - u_{k+1})\mathbf{J}_{11}} \Xi_k e^{(u_n - u_{k+1})(\mathbf{J}_{11})^T} \right]. \tag{62}$$

We know from Lemma E.1 that $\beta_n \Xi_n \to 0$ a.s. because $\Xi_n = o(\gamma_n^{-1})$. Besides,

$$\|\beta_k e^{(u_n - u_k)\mathbf{J}_{11}} - \beta_{k+1} e^{(u_n - u_{k+1})\mathbf{J}_{11}}\|$$
$$= \|(\beta_k - \beta_{k+1})e^{(u_n - u_k)\mathbf{J}_{11}} + \beta_{k+1}e^{(u_n - u_k)\mathbf{J}_{11}}(\mathbf{I} - e^{-\beta_{k+1}\mathbf{J}_{11}})\|$$
$$\leq C_1 \beta_k^2 e^{-(u_n - u_k)T}$$

for some constant $C_1 > 0$ because $\beta_n - \beta_{n+1} \leq C_2 \beta_n^2$ and $\|\mathbf{I} - e^{-\beta_{k+1}\mathbf{J}_{11}}\| \leq C_3 \beta_{k+1}$. Moreover,

$$\|\beta_k e^{(u_n - u_k)\mathbf{J}_{11}}\| + \|\beta_{k+1}e^{(u_n - u_{k+1})\mathbf{J}_{11}}\|$$
$$\leq \beta_k \|e^{(u_n - u_k)\mathbf{J}_{11}}\| + \beta_k \|e^{(u_n - u_k)\mathbf{J}_{11}}\| \cdot \|e^{-\beta_{k+1}\mathbf{J}_{11}}\|$$
$$\leq C_4 \beta_k e^{-(u_n - u_k)T}.$$

Using Lemma G.7 on (62) gives

$$\|A_{1,n}^{(c)}\| \leq C_1 C_4 \beta_n^{-1} \sum_{k=1}^{n-1} \beta_k^2 e^{-2(u_n - u_k)T} \|\beta_k \Xi_k\| + \|\beta_n \Xi_n\|.$$

Applying Lemma G.6 again gives

$$\limsup_n \|A_{1,n}^{(c)}\| \leq C_5 \limsup_n \|\beta_n \Xi_n\| = 0$$

for some constant $C_5 > 0$.

Finally, we provide an existing lemma below.

**Lemma E.4** (Mokkadem & Pelletier (2005) Lemma 4). *For a sequence with decreasing step size* $\beta_n = (n+1)^{-b}$ *for* $b \in (1/2, 1]$, $u_n = \sum_{k=1}^{n} \beta_k$, *a positive semi-definite matrix* $\Gamma$ *and a Hurwitz matrix* $\mathbf{Q}$, *which is given by*

$$\beta_n^{-1} \sum_{k=1}^{n} \beta_n^2 e^{(u_n - u_k)\mathbf{Q}} \Gamma e^{(u_n - u_k)\mathbf{Q}^T},$$

*we have*

$$\lim_{n \to \infty} \beta_n^{-1} \sum_{k=1}^{n} \beta_n^2 e^{(u_n - u_k)\mathbf{Q}} \Gamma e^{(u_n - u_k)\mathbf{Q}^T} = \mathbf{V}$$

*where* $\mathbf{V}$ *is the solution of the Lyapunov equation*

$$\left( \mathbf{Q} + \frac{\mathbb{1}_{\{b=1\}}}{2}\mathbf{I} \right) \mathbf{V} + \mathbf{V} \left( \mathbf{Q}^T + \frac{\mathbb{1}_{\{b=1\}}}{2}\mathbf{I} \right) + \Gamma = 0.$$

Then, $\lim_{n \to \infty} A_{1,n}^{(a)} = \mathbf{V}_{\boldsymbol{\theta}}(\alpha)$ is a direct application of Lemma E.4. $\qquad \square$

We can follow the similar steps in Lemma E.3 to obtain

$$\lim_{n \to \infty} A_{4,n} = \mathbf{V}_{\mathbf{x}}(\alpha),$$

where $\mathbf{V}_{\mathbf{x}}(\alpha)$ is in the form of (32).

The last step is to show $\lim_{n \to \infty} A_{2,n} = 0$. Note that

$$\|A_{2,n}\| = O \left( \beta_n^{-1/2} \gamma_n^{-1/2} \sum_{k=1}^{n} \beta_k \gamma_k \|e^{(u_n - u_k)\mathbf{J}_{11}}\| \|e^{(s_n - s_k)\mathbf{J}_{22}(\alpha)^T}\| \right)$$

$$= O \left( \beta_n^{-1/2} \gamma_n^{-1/2} \sum_{k=1}^{n} \beta_k \gamma_k e^{-(u_n - u_k)T} e^{-(s_n - s_k)T'} \right)$$

$$= O \left( \beta_n^{-1/2} \gamma_n^{-1/2} \sum_{k=1}^{n} \beta_k \gamma_k e^{-(s_n - s_k)T'} \right),$$

where the second equality is from Lemma G.4. Then, we use Lemma G.6 with $p = 0$ to obtain

$$\sum_{k=1}^{n} \beta_k \gamma_k e^{-(s_n - s_k)T'} = O(\beta_n) \tag{63}$$

Additionally, since $\beta_n = o(\gamma_n)$, we have

$$\beta_n^{-1/2} \gamma_n^{-1/2} \sum_{k=1}^{n} \beta_k \gamma_k^{-1/2} \gamma_k^{3/2} e^{-(s_n - s_k)T'} = O(\beta_n^{1/2} \gamma_n^{-1/2}) = o(1).$$

Then, it follows that $\lim_{n \to \infty} A_{2,n} = 0$. Therefore, we obtain

$$\lim_{n \to \infty} \sum_{k=1}^{n} \mathbb{E} \left[ (Z_k^{(n)} - Z_{k-1}^{(n)})(Z_k^{(n)} - Z_{k-1}^{(n)})^T | \mathcal{F}_{k-1} \right] = \begin{pmatrix} \mathbf{V}_{\boldsymbol{\theta}}(\alpha) & 0 \\ 0 & \mathbf{V}_{\mathbf{x}}(\alpha) \end{pmatrix}.$$

Now, we turn to verifying the conditions in Theorem G.3. For some $\tau > 0$, we have

$$\sum_{k=1}^{n} \mathbb{E} \left[ \|Z_k^{(n)} - Z_{k-1}^{(n)}\|^{2+\tau} | \mathcal{F}_{k-1} \right]$$

$$= O \left( \beta_n^{-(1+\frac{\tau}{2})} \sum_{k=1}^{n} \beta_k^{2+\frac{\tau}{2}} \beta_k^{\frac{\tau}{2}} e^{-(2+\tau)(u_n - u_k)T} + \gamma_n^{-(1+\frac{\tau}{2})} \sum_{k=1}^{n} \gamma_k^{2+\frac{\tau}{2}} \gamma_k^{\frac{\tau}{2}} e^{-(2+\tau)(s_n - s_k)T'} \right) \tag{64}$$

$$= O \left( \beta_n^{\frac{\tau}{2}} + \gamma_n^{\frac{\tau}{2}} \right)$$

where the last equality comes from Lemma G.6. Since (64) also holds for $\tau = 0$, we have

$$\sum_{k=1}^{n} \mathbb{E}\left[\|Z_k^{(n)} - Z_{k-1}^{(n)}\|^2 | \mathcal{F}_{k-1}\right] = O(1) < \infty.$$

Therefore, all the conditions in Theorem G.3 are satisfied and its application then gives

$$Z^{(n)} = \begin{pmatrix} \sqrt{\beta_n^{-1}} L_n^{(\boldsymbol{\theta})} \\ \sqrt{\gamma_n^{-1}} L_n^{(\mathbf{x})} \end{pmatrix} \xrightarrow[dist.]{n \to \infty} N\left(0, \begin{pmatrix} \mathbf{V}_{\boldsymbol{\theta}}(\alpha) & 0 \\ 0 & \mathbf{V}_{\mathbf{x}}(\alpha) \end{pmatrix}\right). \tag{65}$$

Furthermore, we have the following lemma about the strong convergence rate of $\{L_n^{(\boldsymbol{\theta})}\}$ and $\{L_n^{(\mathbf{x})}\}$.

**Lemma E.5.**

$$\|L_n^{(\boldsymbol{\theta})}\| = O\left(\sqrt{\beta_n \log(u_n)}\right) \quad a.s. \tag{66a}$$

$$\|L_n^{(\mathbf{x})}\| = O\left(\sqrt{\gamma_n \log(s_n)}\right) \quad a.s. \tag{66b}$$

*Proof.* This proof follows Pelletier (1998, Lemma 1). We only need the special case of Pelletier (1998, Lemma 1) that fits our scenario; e.g., we let the two types of step sizes therein to be the same. Specifically, we attach the following lemma.

**Lemma E.6** (Pelletier (1998) Lemma 1). *Consider a sequence*

$$L_{n+1} = e^{u_n \mathbf{H}} \sum_{k=1}^{n} e^{-u_k \mathbf{H}} \beta_k M_{k+1},$$

*where $\beta_n = n^{-b}$, $1/2 < b \leq 1$, and $\{M_n\}$ is a Martingale difference sequence adapted to the filtration $\mathcal{F}$ such that, almost surely, $\limsup_n \mathbb{E}[\|M_{n+1}\|^2|\mathcal{F}_n] \leq M^2$ and there exists $\tau \in (0, 2)$, $b(2 + \tau) > 2$, such that $\sup_n \mathbb{E}[\|M_{n+1}\|^{2+\tau}|\mathcal{F}_n] < \infty$. Then, almost surely,*

$$\limsup_n \frac{\|L_n\|}{\sqrt{\beta_n \log(u_n)}} \leq C_M, \tag{67}$$

*where $C_M$ is a constant dependent on $M$.*

By assumption A4, the iterates $(\boldsymbol{\theta}_n, \mathbf{x}_n)$ are bounded within a compact subset $\Omega$. Recall the form of $M_{n+1}^{(\boldsymbol{\theta})}, M_{n+1}^{(\mathbf{x})}$ defined in (35), it comprises the functions $\tilde{H}_{\boldsymbol{\theta}_n, \mathbf{x}_n}(i)$ and $(\mathbf{K}_{\mathbf{x}_n} \tilde{H}_{\boldsymbol{\theta}_n, \mathbf{x}_n})(i)$, which in turn include the function $H(\boldsymbol{\theta}, i)$. We know that $H(\boldsymbol{\theta}, i)$ is bounded for $\boldsymbol{\theta}$ in some compact set $\mathcal{C}$. Thus, for any $(\boldsymbol{\theta}_n, \mathbf{x}_n) \in \Omega$ for some compact set $\Omega$, $M_{n+1}^{(\boldsymbol{\theta})}, M_{n+1}^{(\mathbf{x})}$ are bounded and we denote by $c_{\boldsymbol{\theta}}$ and $c_{\mathbf{x}}$ as their upper bounds, i.e., $\mathbb{E}[\|M_{n+1}^{(\boldsymbol{\theta})}\|^2|\mathcal{F}_n] \leq c_\Omega^{(\boldsymbol{\theta})}$ and $\mathbb{E}[\|M_{n+1}^{(\mathbf{x})}\|^2|\mathcal{F}_n] \leq c_\Omega^{(\mathbf{x})}$. We only need to replace the upper bound $c$ in Lemma E.6 by $c_\Omega^{(\boldsymbol{\theta})}$ for the sequence $\{L_n^{(\boldsymbol{\theta})}\}$ (resp. $c_\Omega^{(\mathbf{x})}$ for the sequence $\{L_n^{(\mathbf{x})}\}$), i.e.,

$$\limsup_n \frac{\|L_n^{(\boldsymbol{\theta})}\|}{\sqrt{\beta_n \log(u_n)}} \leq C_\Omega^{(\boldsymbol{\theta})}, \tag{68a}$$

$$\limsup_n \frac{\|L_n^{(\mathbf{x})}\|}{\sqrt{\gamma_n \log(s_n)}} \leq C_\Omega^{(\mathbf{x})}, \tag{68b}$$

such that $\|L_n^{(\boldsymbol{\theta})}\| = O(\sqrt{\beta_n \log(u_n)})$ a.s. and $\|L_n^{(\mathbf{x})}\| = O(\sqrt{\gamma_n \log(s_n)})$ a.s. which completes the proof. □

Note that we have $\mathbf{x}_n - \boldsymbol{\mu}$ and $L_n^{(\mathbf{x})}$ weakly converge to the same Gaussian distribution from Remark E.1 and (65). Then, $\gamma_n^{-1/2} \Delta_n^{(\mathbf{x})}$ weakly converges to zero, implying that $\gamma_n^{-1/2} \Delta_n^{(\mathbf{x})}$ converges to zero with probability 1. Therefore, together with $\{\gamma_n\}$ being strictly positive, we have

$$\Delta_n^{(\mathbf{x})} = o(\sqrt{\gamma_n}) \quad a.s. \tag{69}$$

**Characterization of Sequences $\{R_n^{(\boldsymbol{\theta})}\}$ and $\{\Delta_n^{(\boldsymbol{\theta})}\}$**

We first consider the sequence $\{R_n^{(\boldsymbol{\theta})}\}$. We assume a positive real-valued bounded sequence $\{w_n\}$ under the same conditions as in Mokkadem & Pelletier (2006, Definition 1), i.e.,

**Definition E.1.** In the case $b < 1$, $\frac{w_n}{w_{n+1}} = 1 + o(\beta_n)$, which also implies $\frac{w_n}{w_{n+1}} = 1 + o(\gamma_n)$.

In the case $b = 1$, there exist $\epsilon \geq 0$ and a nondecreasing slowly varying function $l(n)$ such that $w_n = n^{-\epsilon} l(n)$. When $\epsilon = 0$, we require function $l(n)$ to be bounded. $\qquad\square$

Since $\|\mathbf{x}_n - \boldsymbol{\mu}\| = o(1)$ by a.s. convergence result, we can assume that there exists $\{w_n\}$ such that $\|\mathbf{x}_n - \boldsymbol{\mu}\| = O(w_n)$. Then, from (55b), we can use the Abel transformation and obtain

$$R_n^{(\boldsymbol{\theta})} = \beta_n \gamma_n^{-1} \mathbf{J}_{12}(\alpha)\mathbf{J}_{22}(\alpha)^{-1}(\mathbf{x}_n - \boldsymbol{\mu}) - e^{(u_n - u_1)\mathbf{J}_{11}}\beta_1\gamma_1^{-1}\mathbf{U}_{11}\mathbf{J}_{12}(\alpha)\mathbf{J}_{22}(\alpha)^{-1}(\mathbf{x}_1 - \boldsymbol{\mu})$$

$$+ e^{u_n\mathbf{J}_{11}}\sum_{k=1}^{n-1}\left(e^{-u_k\mathbf{J}_{11}}\beta_k\gamma_k^{-1} - e^{-u_{k+1}\mathbf{J}_{11}}\beta_{k+1}\gamma_{k+1}^{-1}\right)\mathbf{J}_{12}(\alpha)\mathbf{J}_{22}(\alpha)^{-1}(\mathbf{x}_{k+1} - \boldsymbol{\mu}),$$

where the last term on the RHS can be rewritten as

$$W_n = \sum_{k=1}^{n-1} e^{(u_n - u_{k+1})\mathbf{J}_{11}}\beta_{k+1}\gamma_{k+1}^{-1}\left(e^{\beta_{k+1}\mathbf{J}_{11}}\beta_k\beta_{k+1}^{-1}\gamma_k^{-1}\gamma_{k+1} - \mathbf{I}\right)\mathbf{J}_{12}(\alpha)\mathbf{J}_{22}(\alpha)^{-1}(\mathbf{x}_{k+1} - \boldsymbol{\mu}).$$

Using Lemma G.6 on $W_n$ gives $\|W_n\| = O(\gamma_n^{-1}\|e^{\beta_n\mathbf{J}_{11}} - \boldsymbol{I}\|\|\mathbf{x}_n - \boldsymbol{\mu}\|) = O(\gamma_n^{-1}\beta_n\omega_n)$. Then, it follows that for some $T > 0$,

$$\|R_n^{(\boldsymbol{\theta})}\| = O\left(\beta_n\gamma_n^{-1}\omega_n + \|e^{u_n\mathbf{J}_{11}}\|\right) = O(\beta_n\gamma_n^{-1}\omega_n + e^{-u_nT}) \tag{70}$$

with the application of Lemma G.4 to the second equality.

Then, we shift our focus on $\{\Delta_n^{(\boldsymbol{\theta})}\}$. Specifically, we take (54), (55a), and (56) back to $\Delta_n^{(\boldsymbol{\theta})} = \boldsymbol{\theta}_n - \boldsymbol{\theta}^* - L_n^{(\boldsymbol{\theta})} - R_n^{(\boldsymbol{\theta})}$, and obtain

$$
\begin{aligned}
\Delta_{n+1}^{(\boldsymbol{\theta})} =& (\mathbf{I} + \beta_{n+1}\mathbf{J}_{11})(\boldsymbol{\theta}_n - \boldsymbol{\theta}^*) \\
&+ \beta_{n+1}(r_n^{(\boldsymbol{\theta},1)} + r_n^{(\boldsymbol{\theta},2)} + \eta_n^{(\boldsymbol{\theta})} - \mathbf{J}_{12}(\alpha)\mathbf{J}_{22}(\alpha)^{-1}(r_n^{(\mathbf{x},1)} + r_n^{(\mathbf{x},2)} + \eta_n^{(\mathbf{x})})) \\
&- e^{\beta_{n+1}\mathbf{J}_{11}}L_n^{(\boldsymbol{\theta})} - e^{\beta_{n+1}\mathbf{J}_{11}}R_n^{(\boldsymbol{\theta})} \\
=& (\mathbf{I} + \beta_{n+1}\mathbf{J}_{11})(\boldsymbol{\theta}_n - \boldsymbol{\theta}^*) \\
&+ \beta_{n+1}(r_n^{(\boldsymbol{\theta},1)} + r_n^{(\boldsymbol{\theta},2)} + \eta_n^{(\boldsymbol{\theta})} - \mathbf{J}_{12}(\alpha)\mathbf{J}_{22}(\alpha)^{-1}(r_n^{(\mathbf{x},1)} + r_n^{(\mathbf{x},2)} + \eta_n^{(\mathbf{x})})) \\
&- (\mathbf{I} + \beta_{n+1}\mathbf{J}_{11} + O(\beta_{n+1}^2))L_n^{(\boldsymbol{\theta})} - (\mathbf{I} + \beta_{n+1}\mathbf{J}_{11} + O(\beta_{n+1}^2))R_n^{(\boldsymbol{\theta})} \\
=& (\mathbf{I} + \beta_{n+1}\mathbf{J}_{11})\Delta_n^{(\boldsymbol{\theta})} + O(\beta_{n+1}^2)(L_n^{(\boldsymbol{\theta})} + R_n^{(\boldsymbol{\theta})})) \\
&+ \beta_{n+1}(r_n^{(\boldsymbol{\theta},1)} + r_n^{(\boldsymbol{\theta},2)} + \eta_n^{(\boldsymbol{\theta})} - \mathbf{J}_{12}(\alpha)\mathbf{J}_{22}(\alpha)^{-1}(r_n^{(\mathbf{x},1)} + r_n^{(\mathbf{x},2)} + \eta_n^{(\mathbf{x})})),
\end{aligned}
\tag{71}
$$

where the second equality is by taking the Taylor expansion $e^{\beta_{n+1}\mathbf{J}_{11}} = \mathbf{I} + \beta_{n+1}\mathbf{J}_{11} + O(\beta_{n+1}^2)$.

Define $\Phi_{k,n} \triangleq \prod_{j=k+1}^{n}(\mathbf{I} + \beta_j\mathbf{J}_{11})$ and by convention $\Phi_{n,n} = \mathbf{I}$. Then, we rewrite (71) as

$$
\begin{aligned}
\Delta_{n+1}^{(\boldsymbol{\theta})} =& \sum_{k=1}^{n} \Phi_{k,n}\beta_{k+1}\left(O(\beta_{k+1})L_k^{(\boldsymbol{\theta})} + O(\beta_{k+1})R_k^{(\boldsymbol{\theta})}\right) \\
&+ \sum_{k=1}^{n} \Phi_{k,n}\beta_{k+1}(r_k^{(\boldsymbol{\theta},1)} + r_k^{(\boldsymbol{\theta},2)} + \eta_k^{(\boldsymbol{\theta})} - \mathbf{J}_{12}(\alpha)\mathbf{J}_{22}(\alpha)^{-1}(r_k^{(\mathbf{x},1)} + r_k^{(\mathbf{x},2)} + \eta_k^{(\mathbf{x})})) \\
=& \sum_{k=1}^{n} \Phi_{k,n}\beta_{k+1}\left(O(\beta_{k+1})L_k^{(\boldsymbol{\theta})} + O(\beta_{k+1})R_k^{(\boldsymbol{\theta})}\right) \\
&+ \sum_{k=1}^{n} \Phi_{k,n}\beta_{k+1}(r_k^{(\boldsymbol{\theta},1)} + \eta_k^{(\boldsymbol{\theta})} - \mathbf{J}_{12}(\alpha)\mathbf{J}_{22}(\alpha)^{-1}(r_k^{(\mathbf{x},1)} + \eta_k^{(\mathbf{x})})) \\
&+ \sum_{k=1}^{n} \Phi_{k,n}\beta_{k+1}(r_k^{(\boldsymbol{\theta},2)} - \mathbf{J}_{12}(\alpha)\mathbf{J}_{22}(\alpha)^{-1}r_k^{(\mathbf{x},2)}).
\end{aligned}
$$

$$\tag{72}$$

From (72), we can indeed decompose $\Delta_{n+1}^{(\boldsymbol{x})}$ into two parts $\Delta_{n+1}^{(\boldsymbol{\theta})} = \Delta_{n+1}^{(\boldsymbol{\theta},1)} + \Delta_{n+1}^{(\boldsymbol{\theta},2)}$, where

$$
\Delta_{n+1}^{(\boldsymbol{\theta},1)} \triangleq \sum_{k=1}^{n} \Phi_{k,n} \beta_{k+1} \left( O(\beta_{k+1}) L_k^{(\boldsymbol{\theta})} + O(\beta_{k+1}) R_k^{(\boldsymbol{\theta})} \right)
$$
$$
+ \sum_{k=1}^{n} \Phi_{k,n} \beta_{k+1} (r_k^{(\boldsymbol{\theta},1)} + \eta_k^{(\boldsymbol{\theta})} - \mathbf{J}_{12}(\alpha) \mathbf{J}_{22}(\alpha)^{-1} (r_k^{(\mathbf{x},1)} + \eta_k^{(\mathbf{x})})), \tag{73a}
$$

$$
\Delta_{n+1}^{(\boldsymbol{\theta},2)} \triangleq \sum_{k=1}^{n} \Phi_{k,n} \beta_{k+1} (r_k^{(\boldsymbol{\theta},2)} - \mathbf{J}_{12}(\alpha) \mathbf{J}_{22}(\alpha)^{-1} r_k^{(\mathbf{x},2)}). \tag{73b}
$$

This term $\Delta_{n+1}^{(\boldsymbol{\theta},1)}$ shares the same recursive form as in the sequence defined in Mokkadem & Pelletier (2006, Lemma 6), which is given below.

**Lemma E.7** (Mokkadem & Pelletier (2006) Lemma 6). *For $\Delta_{n+1}^{(\boldsymbol{\theta},1)}$ in the form of (73a), assume $\|\mathbf{x}_n - \boldsymbol{\mu}\| = O(\omega_n)$ and $\|\Delta_n^{(\mathbf{x})}\| = O(\delta_n)$ for the sequences $\omega_n, \delta_n$ defined in (E.1). Then, we have*

$$
\|\Delta_{n+1}^{(\boldsymbol{\theta},1)}\| = O(\beta_n^2 \gamma_n^{-2} \omega_n^2 + \beta_n \gamma_n^{-1} \delta_n) + o(\sqrt{\beta_n}) \quad a.s.
$$

Since we already have $\Delta_n^{(\mathbf{x})} = o(\sqrt{\gamma_n})$ in (69), together with Lemma E.7, we have

$$
\|\Delta_{n+1}^{(\boldsymbol{\theta},1)}\| = O(\beta_n^2 \gamma_n^{-2} \omega_n^2) + o(\beta_n \gamma_n^{-1/2}) + o(\sqrt{\beta_n}) = O(\beta_n^2 \gamma_n^{-2} \omega_n^2) + o(\sqrt{\beta_n})
$$

where the second equality comes from $o(\beta_n \gamma_n^{-1/2}) = o(\beta_n^{1/2} (\beta_n \gamma_n^{-1})^{1/2}) = o(\beta_n^{1/2})$.

We now focus on $\Delta_{n+1}^{(\boldsymbol{\theta},2)}$. Define a sequence

$$
\Psi_n \triangleq \sum_{k=1}^{n} r_k^{(\boldsymbol{\theta},2)} - \mathbf{J}_{12}(\alpha) \mathbf{J}_{22}(\alpha)^{-1} r_k^{(\mathbf{x},2)}, \tag{74}
$$

and we have

$$
\beta_{n+1}^{-1/2} \sum_{k=1}^{n} \Phi_{k,n} \beta_{k+1} (r_k^{(\boldsymbol{\theta},2)} - \mathbf{J}_{12}(\alpha) \mathbf{J}_{22}(\alpha)^{-1} r_k^{(\mathbf{x},2)})
$$
$$
= \beta_{n+1}^{-1/2} \sum_{k=1}^{n} \Phi_{k,n} \beta_{k+1} (\Psi_k - \Psi_{k-1})
$$
$$
= \beta_{n+1}^{1/2} \Psi_n + \beta_{n+1}^{-1/2} \sum_{k=1}^{n-1} (\beta_k \Phi_{k,n} - \beta_{k+1} \Phi_{k+1,n}) \Psi_k
$$

where the last equality comes from the Abel transformation. Note that

$$
\|\beta_k \Phi_{k,n} - \beta_{k+1} \Phi_{k+1,n}\| \le \beta_{k+1} \|\Phi_{k,n} - \Phi_{k+1,n}\| + (\beta_k - \beta_{k+1}) \|\Phi_{k,n}\|
$$
$$
\le \beta_{k+1} \|\Phi_{k+1,n}\| \beta_k \|\mathbf{J}_{11}\| + C_7 \beta_k^2 \|\Phi_{k,n}\|
$$
$$
\le C_8 \beta_k^2 e^{-(u_n - u_k)T}
$$

for some constant $C_7, C_8 > 0$, where the last inequality is from Lemma G.4 and $\|\Phi_{k+1,n}\| \le C_9 \|\Phi_{k,n}\|$ for some constant $C_9 > 0$ that depends on $e^{\beta_0 T}$. Then,

$$
\beta_{n+1}^{-1/2} \left\| \sum_{k=1}^{n} \Phi_{k,n} \beta_{k+1} (r_k^{(\boldsymbol{\theta},2)} - \mathbf{J}_{12}(\alpha) \mathbf{J}_{22}(\alpha)^{-1} r_k^{(\mathbf{x},2)}) \right\|
$$
$$
\le \|\beta_{n+1}^{1/2} \Psi_n\| + \left( \frac{\beta_{n+1}}{\beta_n} \right)^{1/2} \beta_n^{-1/2} \sum_{k=1}^{n} \|\beta_k \Phi_{k,n} - \beta_{k+1} \Phi_{k+1,n}\| \|\Psi_k\|
$$
$$
\le \|\beta_{n+1}^{1/2} \Psi_n\| + C_8 \left( \frac{\beta_{n+1}}{\beta_n} \right)^{1/2} \beta_n^{-1/2} \sum_{k=1}^{n} \beta_k^{3/2} e^{-(u_n - u_k)T} \|\beta_k^{1/2} \Psi_k\|.
$$

By Lemma E.1, we have $\beta_n^{1/2}\Psi_n \to 0$ a.s. such that by Lemma G.6, it follows that

$$\limsup_n \beta_{n+1}^{-1/2}\left\|\sum_{k=1}^n \Phi_{k,n}\beta_{k+1}(r_k^{(\boldsymbol{\theta},2)} - \mathbf{J}_{12}(\alpha)\mathbf{J}_{22}(\alpha)^{-1}r_k^{(\mathbf{x},2)})\right\| \leq \frac{\limsup_n \|\beta_n^{1/2}\Psi_n\|}{C(T,1/2)} = 0.$$

Therefore, we have

$$\Delta_{n+1}^{(\boldsymbol{\theta},2)} = \sum_{k=1}^n \Phi_{k,n}\beta_{k+1}(r_k^{(\boldsymbol{\theta},2)} - \mathbf{J}_{12}(\alpha)\mathbf{J}_{22}(\alpha)^{-1}r_k^{(\mathbf{x},2)}) = o(\sqrt{\beta_n}). \tag{75}$$

Consequently, $\Delta_{n+1}^{(\boldsymbol{\theta})} = O(\beta_n^2\gamma_n^{-2}\omega_n^2) + o(\sqrt{\beta_n})$ almost surely.

Now we are dealing with $\mathbf{x}_n - \boldsymbol{\mu}$ and its related sequence $\omega_n$. Note that by Lemma E.5 and (69), we have almost surely,

$$\begin{aligned}
\|\mathbf{x}_n - \boldsymbol{\mu}\| &= O(\|L_n^{(\mathbf{x})}\| + \|\Delta_n^{\mathbf{x}}\|) \\
&= O(\sqrt{\gamma_n \log(s_n)} + o(\sqrt{\gamma_n})) \\
&= O(\sqrt{\gamma_n \log(s_n)}).
\end{aligned} \tag{76}$$

Thus, we can set $\omega_n \equiv O(\sqrt{\gamma_n \log(s_n)})$ such that $\|R_n^{(\boldsymbol{\theta})}\|$ in (70) can be written as

$$\|R_n^{(\boldsymbol{\theta})}\| = O(n^{a/2-b}\sqrt{log(s_n)} + e^{-u_n T}),$$

and

$$\|\Delta_{n+1}^{(\boldsymbol{\theta})}\| = O(n^{a-2b}log(s_n)) + o(\sqrt{\beta_n}).$$

In view of assumption A2 and $\beta_n = o(\gamma_n)$, $a/2 - b < -b/2$ and $a - 2b < -b$, there exists a $c > b/2$ such that almost surely,

$$\|R_n^{(\boldsymbol{\theta})}\| = O(n^{-s}), \quad \|\Delta_{n+1}^{(\boldsymbol{\theta})}\| = o(\sqrt{\beta_n}).$$

Therefore, $\beta_n^{-1/2}(R_n^{(\boldsymbol{\theta})} + \Delta_{n+1}^{(\boldsymbol{\theta})}) \to 0$ almost surely. This completes the proof of Scenario 2.

### E.3 CASE (III): $\gamma_n = o(\beta_n)$

For $\gamma_n = o(\beta_n)$, we can see that the roles of $\boldsymbol{\theta}_n$ and $\mathbf{x}_n$ are flipped, i.e., $\boldsymbol{\theta}_n$ is now on fast timescale while $\mathbf{x}_n$ is on slow timescale.

We still decompose $\mathbf{x}_n$ as $\mathbf{x}_n - \boldsymbol{\mu} = L_n^{(\mathbf{x})} + \Delta_n^{(\mathbf{x})}$, where $L_n^{(\mathbf{x})}, \Delta_n^{(\mathbf{x})}$ are defined in (56) and (57), respectively. Since $\mathbf{x}_n$ is independent of $\boldsymbol{\theta}_n$, the results of $L_n^{(\mathbf{x})}$ and $\Delta_n^{(\mathbf{x})}$ remain the same, i.e., almost surely, $L_n^{(\mathbf{x})} = O(\sqrt{\gamma_n \log(s_n)})$ from Lemma E.5 and $\Delta_n^{(\mathbf{x})} = o(\sqrt{\gamma_n})$ from (69). Then, we define sequences $\hat{L}_n^{(\boldsymbol{\theta})}$ and $\hat{R}_n^{(\boldsymbol{\theta})}$ as follows.

$$\hat{L}_n^{(\boldsymbol{\theta})} \triangleq e^{\beta_n \mathbf{J}_{11}}\hat{L}_{n-1}^{(\boldsymbol{\theta})} + \beta_n M_n^{(\boldsymbol{\theta})} = \sum_{k=1}^n e^{(u_n - u_k)\mathbf{J}_{11}}\beta_k M_k^{(\boldsymbol{\theta})}, \tag{77a}$$

$$\hat{R}_n^{(\boldsymbol{\theta})} \triangleq e^{\beta_n \mathbf{J}_{11}}\hat{R}_{n-1}^{(\boldsymbol{\theta})} + \beta_n \mathbf{J}_{12}(\alpha)(L_{n-1}^{(\mathbf{x})} + R_{n-1}^{(\mathbf{x})}) = \sum_{k=1}^n e^{(u_n - u_k)\mathbf{J}_{11}}\beta_k \mathbf{J}_{12}(\alpha)(L_{k-1}^{(\mathbf{x})} + R_{k-1}^{(\mathbf{x})}). \tag{77b}$$

Moreover, the remaining term $\hat{\Delta}_n^{(\boldsymbol{\theta})} \triangleq \boldsymbol{\theta}_n - \boldsymbol{\theta}^* - \hat{L}_n^{(\boldsymbol{\theta})} - \hat{R}_n^{(\boldsymbol{\theta})}$.

The proof outline is the same as in the previous scenario:

- We first show $\beta_n^{-1/2}\hat{\Delta}_n^{(\boldsymbol{\theta})}$ weakly converges to the distribution $N(0, \mathbf{V}_{\boldsymbol{\theta}}^{(3)})(\alpha)$;

- We analyse $\hat{L}_n^{(\boldsymbol{\theta})}$ and $\hat{R}_n^{(\boldsymbol{\theta})}$ to ensure that these two terms decrease faster than the CLT scale $\beta_n^{-1/2}$, i.e., $\lim_{n\to\infty}\beta_n^{-1/2}(\hat{L}_n^{(\boldsymbol{\theta})} - \hat{R}_n^{(\boldsymbol{\theta})}) = 0$;

- With above two steps, we can show that $\beta_n^{-1/2}(\boldsymbol{\theta}_n - \boldsymbol{\theta}^*)$ weakly converges to the distribution $N(0, \mathbf{V}_{\boldsymbol{\theta}}^{(3)})(\alpha)$.

## Analysis of $\hat{L}_n^{(\boldsymbol{\theta})}$

We first focus on $\hat{L}_n^{(\boldsymbol{\theta})}$ and follow similar steps as we did when we analysed $L_n^{(\boldsymbol{\theta})}$ in the previous scenario. We set a Martingale $Z^{(n)} = \{Z_k^{(n)}\}_{k \geq 1}$ such that

$$Z_k^{(n)} = \beta_n^{-1/2} \sum_{k=1}^{n} e^{(u_n - u_k)\mathbf{J}_{11}} \beta_k M_k^{(\boldsymbol{\theta})}.$$

Then,

$$A_n \triangleq \sum_{k=1}^{n} \mathbb{E}\left[ (Z_k^{(n)} - Z_{k-1}^{(n)})(Z_k^{(n)} - Z_{k-1}^{(n)})^T \,\middle|\, \mathcal{F}_{k-1} \right].$$

Following the similar steps in (59) to decompose $M_k^{(\boldsymbol{\theta})}$ with (42b), we have

$$
\begin{aligned}
A_n &= \beta_n^{-1} \sum_{k=1}^{n} \beta_k^2 e^{(u_n - u_k)\mathbf{J}_{11}} \left( \mathbf{U}_{11} + \mathbf{D}_k^{(3)} + \mathbf{J}_k^{(3)} \right) e^{(u_n - u_k)\mathbf{J}_{11}^T} \\
&= \underbrace{\beta_n^{-1} \sum_{k=1}^{n} \beta_k^2 e^{(u_n - u_k)\mathbf{J}_{11}} \mathbf{U}_{11} e^{(u_n - u_k)\mathbf{J}_{11}^T}}_{A_n^{(a)}} + \underbrace{\beta_n^{-1} \sum_{k=1}^{n} \beta_k^2 e^{(u_n - u_k)\mathbf{J}_{11}} \mathbf{D}_k^{(3)} e^{(u_n - u_k)\mathbf{J}_{11}^T}}_{A_n^{(b)}} \\
&\quad + \underbrace{\beta_n^{-1} \sum_{k=1}^{n} \beta_k^2 e^{(u_n - u_k)\mathbf{J}_{11}} \mathbf{J}_k^{(3)} e^{(u_n - u_k)\mathbf{J}_{11}^T}}_{A_n^{(c)}}
\end{aligned}
\tag{78}
$$

Since $A_n^{(a)}, A_n^{(b)}, A_n^{(c)}$ share similar forms as in Lemma E.3, we follow the same steps as the proof therein, with the application of Lemma E.1. To avoid repetition, we omit the proof and directly give the following lemma.

**Lemma E.8.** *For $A_n^{(a)}, A_n^{(b)}, A_n^{(c)}$ defined in* (78), *we have*

$$\lim_{n \to \infty} A_n^{(a)} = \mathbf{V}_{\boldsymbol{\theta}}^{(3)}(\alpha), \quad \lim_{n \to \infty} \|A_n^{(b)}\| = 0, \quad \lim_{n \to \infty} \|A_n^{(c)}\| = 0, \tag{79}$$

*where* $\mathbf{V}_{\boldsymbol{\theta}}^{(3)}(\alpha)$ *is the solution to the Lyapunov equation*

$$\mathbf{J}_{11}\mathbf{V} + \mathbf{V}\mathbf{J}_{11}^T + \mathbf{U}_{11} = 0.$$

Note that here we don't have the term $\frac{\mathbb{1}_{\{b=1\}}}{2}\boldsymbol{I}$ in above lemma, compared to Lemma E.3, because in the case of $\gamma_n = o(\beta_n)$, $b < 1$ such that $\mathbb{1}_{\{b=1\}} = 0$. Then, applying Lemma G.1 to derive the closed form of $\mathbf{V}_{\boldsymbol{\theta}}^{(3)}(\alpha)$ gives

$$\mathbf{V}_{\boldsymbol{\theta}}^{(3)}(\alpha) = \int_0^{\infty} e^{t\nabla_{\boldsymbol{\theta}}\mathbf{h}(\boldsymbol{\theta}^*)} \mathbf{U}_{11} e^{t\nabla_{\boldsymbol{\theta}}\mathbf{h}(\boldsymbol{\theta}^*)} dt.$$

Thus, it follows that

$$\lim_{n \to \infty} \sum_{k=1}^{n} \mathbb{E}\left[ (Z_k^{(n)} - Z_{k-1}^{(n)})(Z_k^{(n)} - Z_{k-1}^{(n)})^T | \mathcal{F}_{k-1} \right] = \mathbf{V}_{\boldsymbol{\theta}}^{(3)}(\alpha).$$

Again, we use the Martingale CLT result in Theorem G.3 and have the following result.

$$Z^n = \beta_n^{-1/2} \hat{L}_n^{(\boldsymbol{\theta})} \xrightarrow[\text{dist.}]{n \to \infty} N\left( 0, \mathbf{V}_{\boldsymbol{\theta}}^{(3)}(\alpha) \right).$$

Moreover, similar to the tighter upper bound of $L_n^{(\mathbf{x})}$ proved in Lemma E.5, we utilize the tighter upper bound Lemma E.6 in the proof thereof, and obtain $\hat{L}_n^{(\boldsymbol{\theta})} = O(\sqrt{\beta_n \log(u_n)})$.

**Analysis of $\hat{R}_n^{(\boldsymbol{\theta})}$**

Next, we turn to the term $\hat{R}_n^{(\boldsymbol{\theta})}$ in (77b). Taking the norm gives the following inequality for some constant $C, T > 0$ by applying Lemma G.4,

$$\|\hat{R}_n^{(\boldsymbol{\theta})}\| \le C \sum_{k=1}^{n} e^{-(u_n - u_k)T} \beta_k (\|L_{k-1}^{(\mathbf{x})}\| + \|R_{k-1}^{(\mathbf{x})}\|).$$

Using Lemma G.6 gives

$$\sum_{k=1}^{n} e^{-(u_n - u_k)T} \beta_k (\|L_{k-1}^{(\mathbf{x})}\| + \|R_{k-1}^{(\mathbf{x})}\|) = O(\|L_{k-1}^{(\mathbf{x})}\| + \|R_{n-1}^{(\mathbf{x})}\|).$$

Thus, $\beta_n^{-1/2} \|\hat{R}_n^{(\boldsymbol{\theta})}\| = o(\sqrt{\gamma_n \beta_n^{-1}}) + O\left(\sqrt{\gamma_n \beta_n^{-1} \log(s_n)}\right)$. Since $\gamma_n = o(\beta_n)$, $\gamma_n \beta_n^{-1} = (n+1)^{b-a}$, where $b - a < 0$. Then, there exists some $s > 0$ such that $b - a < -s < 0$. Together with $\log(s_n) = O(\log(n))$, we have $O\left(\sqrt{\gamma_n \beta_n^{-1} \log(s_n)}\right) = O(\sqrt{n^{-s} \log(n)}) = o(1)$. Therefore, we have

$$\lim_{n \to \infty} \beta_n^{-1/2} \hat{R}_n^{(\boldsymbol{\theta})} = 0.$$

**Analysis of $\hat{\Delta}_n^{(\boldsymbol{\theta})}$**

Lastly, let's focus on the term $\hat{\Delta}_n^{(\boldsymbol{\theta})}$. We have

$$
\begin{aligned}
\hat{\Delta}_{n+1}^{(\boldsymbol{\theta})} &= \boldsymbol{\theta}_{n+1} - \boldsymbol{\theta}^* - \hat{L}_{n+1}^{(\boldsymbol{\theta})} - \hat{R}_{n+1}^{(\boldsymbol{\theta})} \\
&= \boldsymbol{\theta}_n - \boldsymbol{\theta}^* + \beta_{n+1}\left(\mathbf{J}_{11}(\boldsymbol{\theta}_n - \boldsymbol{\theta}^*) + \mathbf{J}_{12}(\alpha)(\mathbf{x}_n - \boldsymbol{\mu}) + M_{n+1}^{(\boldsymbol{\theta})} + r_n^{(\boldsymbol{\theta},1)} + r_n^{(\boldsymbol{\theta},2)} + \eta_n^{(\boldsymbol{\theta})}\right) \\
&\quad - e^{\beta_{n+1}\mathbf{J}_{11}} \hat{L}_n^{(\boldsymbol{\theta})} - \beta_{n+1} M_{n+1}^{(\boldsymbol{\theta})} - e^{\beta_{n+1}\mathbf{J}_{11}} \hat{R}_n^{(\boldsymbol{\theta})} - \beta_{n+1}\mathbf{J}_{12}(\alpha)(L_n^{(\mathbf{x})} + R_n^{(\mathbf{x})}) \\
&= (\mathbf{I} + \beta_{n+1}\mathbf{J}_{11})(\boldsymbol{\theta}_n - \boldsymbol{\theta}^*) + \beta_{n+1}\mathbf{J}_{12}(\alpha)\Delta_n^{(\mathbf{x})} + \beta_{n+1}(r_n^{(\boldsymbol{\theta},1)} + r_n^{(\boldsymbol{\theta},2)} + \eta_n^{(\boldsymbol{\theta})}) \\
&\quad - (\mathbf{I} + \beta_{n+1}\mathbf{J}_{11} + O(\beta_{n+1}^2))(\hat{L}_n^{(\boldsymbol{\theta})} + \hat{R}_n^{(\boldsymbol{\theta})}) \\
&= (\mathbf{I} + \beta_{n+1}\mathbf{J}_{11})\hat{\Delta}_n^{(\boldsymbol{\theta})} + \beta_{n+1}\mathbf{J}_{12}(\alpha)\Delta_n^{(\mathbf{x})} + \beta_{n+1}(r_n^{(\boldsymbol{\theta},1)} + r_n^{(\boldsymbol{\theta},2)} + \eta_n^{(\boldsymbol{\theta})}) \\
&\quad + O(\beta_{n+1}^2)(\hat{L}_n^{(\boldsymbol{\theta})} + \hat{R}_n^{(\boldsymbol{\theta})}).
\end{aligned}
$$

where the second equality is from (53a), the third equality stems from the approximation of $e^{\beta_{n+1}\mathbf{J}_{11}}$. Then, we again use the definition $\Phi_{k,n} \triangleq \prod_{j=k+1}^{n}(\mathbf{I} + \beta_j \mathbf{J}_{11})$ and reiterate the above equation as

$$
\begin{aligned}
\hat{\Delta}_{n+1}^{(\boldsymbol{\theta})} &= \sum_{k=1}^{n} \Phi_{k,n}\beta_{k+1}\left(O(\beta_{k+1})L_k^{(\boldsymbol{\theta})} + O(\beta_{k+1})R_k^{(\boldsymbol{\theta})}\right) \\
&\quad + \sum_{k=1}^{n} \Phi_{k,n}\beta_{k+1}\mathbf{J}_{12}(\alpha)\Delta_n^{(\mathbf{x})} + \sum_{k=1}^{n} \Phi_{k,n}\beta_{k+1}(r_k^{(\boldsymbol{\theta},1)} + \eta_k^{(\boldsymbol{\theta})}) \\
&\quad + \sum_{k=1}^{n} \Phi_{k,n}\beta_{k+1}r_k^{(\boldsymbol{\theta},2)} \\
&\triangleq \hat{\Delta}_{n+1}^{(\boldsymbol{\theta},1)} + \hat{\Delta}_{n+1}^{(\boldsymbol{\theta},2)},
\end{aligned}
$$

where $\hat{\Delta}_{n+1}^{(\boldsymbol{\theta},2)} = \sum_{k=1}^{n} \Phi_{k,n}\beta_{k+1}r_k^{(\boldsymbol{\theta},2)}$ and

$$
\begin{aligned}
\hat{\Delta}_{n+1}^{(\boldsymbol{\theta},1)} &= \sum_{k=1}^{n} \Phi_{k,n}\beta_{k+1}\left(O(\beta_{k+1})L_k^{(\boldsymbol{\theta})} + O(\beta_{k+1})R_k^{(\boldsymbol{\theta})}\right) \\
&\quad + \sum_{k=1}^{n} \Phi_{k,n}\beta_{k+1}(r_k^{(\boldsymbol{\theta},1)} + \eta_k^{(\boldsymbol{\theta})} + \mathbf{J}_{12}(\alpha)\Delta_n^{(\mathbf{x})}).
\end{aligned}
\tag{80}
$$

For $\hat{\Delta}_{n+1}^{(\boldsymbol{\theta},2)}$, we follow the same steps from (74) to (75), and obtain $\hat{\Delta}_{n+1}^{(\boldsymbol{\theta},2)} = o(\sqrt{\beta_n})$.

Next, we consider $\hat{\Delta}_{n+1}^{(\boldsymbol{\theta},1)}$ and want to show that $\hat{\Delta}_{n+1}^{(\boldsymbol{\theta},1)} = o(\sqrt{\beta_n})$. Again, we utilize Mokkadem & Pelletier (2006, Lemma 6) for $\hat{\Delta}_{n+1}^{(\boldsymbol{\theta},1)}$ and adapt the notation here for the case $\gamma_n = o(\beta_n)$.

**Lemma E.9.** *For $\hat{\Delta}_{n+1}^{(\boldsymbol{\theta},1)}$ in the form of (80), assume $\|\boldsymbol{\theta}_n - \boldsymbol{\theta}^*\| = O(\omega_n)$ and $\|\hat{\Delta}_n^{(\boldsymbol{\theta},1)}\| = O(\delta_n)$ for the sequences $\omega_n, \delta_n$ defined in (E.1). Then, we have*

$$\|\hat{\Delta}_{n+1}^{(\boldsymbol{\theta},1)}\| = O(\gamma_n^2 \beta_n^{-2} \omega_n^2 + \gamma_n \beta_n^{-1} \delta_n) + o(\sqrt{\gamma_n}) \quad a.s. \tag{81}$$

Now we need to further analyse $\delta_n$ and tighten its big O form, starting from $\delta_n \equiv 1$, so that we can finally obtain the big O form of $\|\hat{\Delta}_{n+1}^{(\boldsymbol{\theta},1)}\|$. The following steps are borrowed from the ideas in Mokkadem & Pelletier (2006, Section 2.3.2).

By almost sure convergence result $\lim_{n\to\infty} \boldsymbol{\theta}_n = \boldsymbol{\theta}^*$, we have $\lim_{n\to\infty} \Delta_n^{(\boldsymbol{\theta})} = 0$ a.s. such that we can first set $\delta_n \equiv 1$, and $\|\hat{\Delta}_{n+1}^{(\boldsymbol{\theta},1)}\| = O(\gamma_n^2 \beta_n^{-2} \omega_n^2 + \gamma_n \beta_n^{-1}) + o(\sqrt{\gamma_n})$. Then, we redefine

$$\delta_n \equiv O(\gamma_n^2 \beta_n^{-2} \omega_n^2 + \gamma_n \beta_n^{-1}) + o(\sqrt{\gamma_n}),$$

and notice that it still satisfies definition E.1. Then, reapplying this $\delta_n$ form to (81) gives

$$\|\hat{\Delta}_{n+1}^{(\boldsymbol{\theta},1)}\| = O(\gamma_n^2 \beta_n^{-2} \omega_n^2 + [\gamma_n \beta_n^{-1}]^2) + o(\sqrt{\gamma_n})$$

and by induction we have for all integers $k \geq 1$,

$$\|\hat{\Delta}_{n+1}^{(\boldsymbol{\theta},1)}\| = O(\gamma_n^2 \beta_n^{-2} \omega_n^2 + [\gamma_n \beta_n^{-1}]^k) + o(\sqrt{\gamma_n}).$$

Since $[\gamma_n \beta_n^{-1}]^k = n^{(b-a)k}$, there exists $k_0 > a/2(a-b)$ such that $[\gamma_n \beta_n^{-1}]^{k_0} = o(\sqrt{\gamma_n})$, and

$$\|\hat{\Delta}_{n+1}^{(\boldsymbol{\theta},1)}\| = O(\gamma_n^2 \beta_n^{-2} \omega_n^2) + o(\sqrt{\gamma_n}). \tag{82}$$

Then, as suggested in Mokkadem & Pelletier (2006, Section 2.3.2), we can choose $\omega_n = O(\sqrt{\beta_n \log(u_n)} + [\gamma_n \beta_n^{-1}]^k)$, which also satisfies definition E.1. Then,

$$\|\boldsymbol{\theta}_n - \boldsymbol{\theta}^*\| = \|\hat{L}_n^{(\boldsymbol{\theta})} + \hat{R}_n^{(\boldsymbol{\theta})} + \hat{\Delta}_n^{(\boldsymbol{\theta})}\|$$

$$= O\left(\sqrt{\beta_n \log(u_n)} + \sqrt{\gamma_n \beta_n^{-1} \log(s_n)} + \left([\gamma_n \beta_n^{-1}]^{k+1} + \gamma_n \beta_n^{-1}\sqrt{\beta_n \log(u_n)}\right)^2\right)$$

$$+ o(\sqrt{\beta_n} + \sqrt{\gamma_n})$$

$$= O(\sqrt{\beta_n \log(u_n)} + [\gamma_n \beta_n^{-1}]^{k+1}).$$

By induction, this holds for all $k \geq 1$ such that there exists $k_0$, $[\gamma_n \beta_n^{-1}]^{k_0} = o(\sqrt{\beta_n})$ and $\|\boldsymbol{\theta}_n - \boldsymbol{\theta}^*\| = O(\sqrt{\beta_n \log(u_n)})$. Equivalently, $\omega_n = \sqrt{\beta_n \log(u_n)}$. Therefore, from (82) we have

$$\|\hat{\Delta}_{n+1}^{(\boldsymbol{\theta},1)}\| = O(\gamma_n^2 \beta_n^{-1} \log(u_n)) + o(\sqrt{\gamma_n}) = o(\sqrt{\gamma_n}).$$

Together with $\|\hat{\Delta}_{n+1}^{(\boldsymbol{\theta},2)}\| = o(\sqrt{\beta_n})$, we have $\beta_n^{-1/2}\|\hat{\Delta}_{n+1}^{(\boldsymbol{\theta})}\| = o(\sqrt{\gamma_n \beta_n^{-1}}) + 1)$ such that

$$\lim_{n\to\infty} \beta_n^{-1/2} \hat{\Delta}_{n+1}^{(\boldsymbol{\theta})} = 0.$$

Thus, we have finished the proof according to the proof outline mentioned at the beginning of this part.

## F  DISCUSSION OF COVARIANCE ORDERING OF SA-SRRW

### F.1  PROOF OF PROPOSITION 3.4

For any $\alpha > 0$ and any vector $\mathbf{x} \in \mathbb{R}^d$, we have

$$\mathbf{x}^T \mathbf{V}_{\boldsymbol{\theta}}^{(1)}(\alpha)\mathbf{x} = \int_0^\infty \mathbf{x}^T e^{t(\nabla_{\boldsymbol{\theta}}\mathbf{h}(\boldsymbol{\theta}^*) + \frac{\mathbb{1}_{\{b=1\}}}{2}\boldsymbol{I})} \mathbf{U}_{\boldsymbol{\theta}}(\alpha) e^{t(\nabla_{\boldsymbol{\theta}}\mathbf{h}(\boldsymbol{\theta}^*) + \frac{\mathbb{1}_{\{b=1\}}}{2}\boldsymbol{I})^T} \mathbf{x} \, dt$$

where the first equality is from the form of $\mathbf{V}_{\boldsymbol{\theta}}^{(1)}(\alpha)$ in Theorem 3.3. Let $\mathbf{y} \triangleq e^{t(\nabla_{\boldsymbol{\theta}}\mathbf{h}(\boldsymbol{\theta}^*)+\frac{\mathbb{1}_{\{b=1\}}}{2}\boldsymbol{I})}\mathbf{x}$, with the dependence on variable $t$ left implicit. The matrix $\mathbf{U}_{\boldsymbol{\theta}}(\alpha)$, given explicitly in (11) positive semi definite, since $\lambda_i \in (-1,1)$ for all $i \in \{1, \cdots, N-1\}$. Thus, the terms $\mathbf{y}^T\mathbf{U}_{\boldsymbol{\theta}}(\alpha)\mathbf{y}$ inside the integral are non-negative, and it is enough to provide an ordering on $\mathbf{y}^T\mathbf{U}_{\boldsymbol{\theta}}(\alpha)\mathbf{y}$ with respect to $\alpha$.

For any $\alpha_2 > \alpha_1 > 0$,

$$
\begin{aligned}
\mathbf{y}^T\mathbf{U}_{\boldsymbol{\theta}}(\alpha_2)\mathbf{y} &= \sum_{i=1}^{N-1} \frac{1}{(\alpha_2(1+\lambda_i)+1)^2} \cdot \frac{1+\lambda_i}{1-\lambda_i}\mathbf{y}^T\mathbf{H}^T\mathbf{u}_i\mathbf{u}_i^T\mathbf{H}\mathbf{y} \\
&< \sum_{i=1}^{N-1} \frac{1}{(\alpha_1(1+\lambda_i)+1)^2} \cdot \frac{1+\lambda_i}{1-\lambda_i}\mathbf{y}^T\mathbf{H}^T\mathbf{u}_i\mathbf{u}_i^T\mathbf{H}\mathbf{y} = \mathbf{y}^T\mathbf{U}_{\boldsymbol{\theta}}(\alpha_1)\mathbf{y} \\
&< \sum_{i=1}^{N-1} \cdot\frac{1+\lambda_i}{1-\lambda_i}\mathbf{y}^T\mathbf{H}^T\mathbf{u}_i\mathbf{u}_i^T\mathbf{H}\mathbf{y} = \mathbf{y}^T\mathbf{U}_{\boldsymbol{\theta}}(0)\mathbf{y},
\end{aligned}
$$

where the inequality[13] is because $\alpha(1+\lambda_i) > 0$ for all $i \in \{1, \cdots, N\}$ and any $\alpha > 0$. In fact, the ordering is monotone in $\alpha$, and $\mathbf{y}^T\mathbf{U}_{\boldsymbol{\theta}}(\alpha_2)\mathbf{y}$ decreases at rate $1/\alpha^2$ as seen form its form in the equation above. This completes the proof.

### F.2 DISCUSSION REGARDING PROPOSITION 3.4 AND MSE ORDERING

We can use Proposition 3.4 to show that the MSE of SA iterates of (4c) driven by SRRW eventually becomes smaller than that SA iterates when the stochastic noise is driven by an *i.i.d.* sequence of random variables. The diagonal entries of $\mathbf{V}_{\boldsymbol{\theta}}^{(1)}(\alpha)$ are obtained by evaluating $\mathbf{e}_i^T\mathbf{V}_{\boldsymbol{\theta}}^{(1)}(\alpha)\mathbf{e}_i$, where $\mathbf{e}_i$ is the $i$'th standard basis vector.[14] These diagonal entries are the asymptotic variance corresponding to the element-wise iterate errors, and for large enough $n$, we have $\mathbf{e}_i^T\mathbf{V}_{\boldsymbol{\theta}}^{(1)}(\alpha)\mathbf{e}_i \approx \mathbb{E}[(\boldsymbol{\theta}_n - \boldsymbol{\theta}^*)_i^2]/\beta_n$ for all $i \in \{1, \cdots, D\}$. Thus, the trace of matrix $\mathbf{V}_{\boldsymbol{\theta}}^{(1)}(\alpha)$ approximates the scaled MSE, that is $\text{Tr}(\mathbf{V}_{\boldsymbol{\theta}}^{(1)}(\alpha)) = \sum_i \mathbf{e}_i^T\mathbf{V}_{\boldsymbol{\theta}}^{(1)}(\alpha)\mathbf{e}_i \approx \sum_i \mathbb{E}[(\boldsymbol{\theta}_n - \boldsymbol{\theta}^*)_i^2]/\beta_n = \mathbb{E}[\|\boldsymbol{\theta}_n - \boldsymbol{\theta}^*\|^2]/\beta_n$ for large $n$. Since all entries of $\mathbf{V}_{\boldsymbol{\theta}}^{(1)}(\alpha)$ go to zero as $\alpha$ increases, they get smaller than the corresponding term for the SA algorithm with *i.i.d.* input for large enough $\alpha$, which achieves a constant MSE in the similarly scaled limit, since the asymptotic covariance is not a function of $\alpha$. Moreover, the value of $\alpha$ only needs to be moderately large, since the asymptotic covariance terms decrease at rate $O(1/\alpha^2)$ as shown in Proposition 3.4.

### F.3 PROOF OF COROLLARY 3.5

We see that $\mathbf{V}_{\boldsymbol{\theta}}^{(3)}(\alpha) = \mathbf{V}_{\boldsymbol{\theta}}^{(3)}(0)$ for all $\alpha > 0$, because the form of $\mathbf{V}_{\boldsymbol{\theta}}^{(3)}(\alpha)$ in Theorem 3.3 is independent of $\alpha$. To prove that $\mathbf{V}_{\boldsymbol{\theta}}^{(1)}(\alpha) <_L \mathbf{V}_{\boldsymbol{\theta}}^{(3)}(0)$, it is enough to show that $\mathbf{V}_{\boldsymbol{\theta}}^{(1)}(0) = \mathbf{V}_{\boldsymbol{\theta}}^{(3)}(0)$, since $\mathbf{V}_{\boldsymbol{\theta}}^{(1)}(\alpha) <_L \mathbf{V}_{\boldsymbol{\theta}}^{(1)}(0)$ from Proposition 3.4. This is easily checked by substituting $\alpha = 0$ in 11, for which $\mathbf{U}_{\boldsymbol{\theta}}(0) = \mathbf{U}_{11}$. Substituting in the respective forms of $\mathbf{V}_{\boldsymbol{\theta}}^{(1)}(0)$ and $\mathbf{V}_{\boldsymbol{\theta}}^{(3)}(0)$ in Theorem 3.3, we get equivalence. This completes the proof.

## G BACKGROUND THEORY

### G.1 TECHNICAL LEMMAS

**Lemma G.1** (Solution to the Lyapunov Equation)**.** *If all the eigenvalues of matrix $\mathbf{M}$ have negative real part, then for every positive semi-definite matrix $\mathbf{U}$ there exists a unique positive semi-definite*

---

[13]The inequality may not be strict when $H$ is low rank, however it will always be true for some choice of $\mathbf{x}$, since $\mathbf{H}$ is not a zero matrix. Thus, the ordering derived still follows our definition of $<_L$ in Section 1, footnote 6.

[14]$D$-dimensional vector of all zeros except at the $i$'th position which is 1.

*matrix* $\mathbf{V}$ *satisfying the Lyapunov equation* $\mathbf{U} + \mathbf{MV} + \mathbf{VM}^T = \mathbf{0}$. *The explicit solution* $\mathbf{V}$ *is given as*

$$\mathbf{V} = \int_0^\infty e^{\mathbf{M}t} \mathbf{U} e^{(\mathbf{M}^T)t} dt. \tag{83}$$

Chellaboina & Haddad (2008, Theorem 3.16) states that for a positive definite matrix $\mathbf{U}$, there exists a positive definite matrix $\mathbf{V}$. The reason they focus on the positive definite matrix $\mathbf{U}$ is that they require the related autonomous ODE system to be asymptotically stable. However, in this paper we don't need this requirement. The same steps therein can be used to prove Lemma G.1 and show that if $\mathbf{U}$ is positive semi-definite, then $\mathbf{V}$ in the form of (83) is unique and also positive semi-definite.

**Lemma G.2** (Burkholder Inequality, Davis (1970), Hall et al. (2014) Theorem 2.10). *Given a Martingale difference sequence* $\{M_{i,n}\}_{i=1}^n$, *for* $p \geq 1$ *and some positive constant* $C_p$, *we have*

$$\mathbb{E}\left[\left\|\sum_{i=1}^n M_{i,n}\right\|^p\right] \leq C_p \mathbb{E}\left[\left(\sum_{i=1}^n \|M_{i,n}\|^2\right)^{p/2}\right] \tag{84}$$

**Theorem G.3** (Martingale CLT, Delyon (2000) Theorem 30). *If a Martingale difference array* $\{X_{n,i}\}$ *satisfies the following condition: for some* $\tau > 0$,

$$\sum_{k=1}^n \mathbb{E}\left[\|X_{n,k}\|^{2+\tau}|\mathcal{F}_{k-1}\right] \xrightarrow{\mathbb{P}} 0, \tag{85}$$

$$\sup_n \sum_{k=1}^n \mathbb{E}\left[\|X_{n,k}\|^2|\mathcal{F}_{k-1}\right] < \infty, \tag{86}$$

*and*

$$\sum_{k=1}^n \mathbb{E}\left[X_{n,k}X_{n,k}^T|\mathcal{F}_{k-1}\right] \xrightarrow{\mathbb{P}} \boldsymbol{V}, \tag{87}$$

*then*

$$\sum_{i=1}^n X_{n,i} \xrightarrow{dist.} N(0, \boldsymbol{V}). \tag{88}$$

$\square$

**Lemma G.4** (Duflo (1996) Proposition 3.I.2). *For a Hurwitz matrix* $\boldsymbol{H}$, *there exist some positive constants* $C, b$ *such that for any* $n$,

$$\left\|e^{\boldsymbol{H}n}\right\| \leq Ce^{-bn}. \tag{89}$$

**Lemma G.5** (Fort (2015) Lemma 5.8). *For a Hurwitz matrix* $\boldsymbol{A}$, *denote by* $-r$, $r > 0$, *the largest real part of its eigenvalues. Let a positive sequence* $\{\gamma_n\}$ *such that* $\lim_n \gamma_n = 0$. *Then for any* $0 < r' < r$, *there exists a positive constant* $C$ *such that for any* $k < n$,

$$\left\|\prod_{j=k}^n (\boldsymbol{I} + \gamma_j \boldsymbol{A})\right\| \leq Ce^{-r'\sum_{j=k}^n \gamma_j}. \tag{90}$$

**Lemma G.6** (Fort (2015) Lemma 5.9, Mokkadem & Pelletier (2006) Lemma 10). *Let* $\{\gamma_n\}$ *be a positive sequence such that* $\lim_n \gamma_n = 0$ *and* $\sum_n \gamma_n = \infty$. *Let* $\{\epsilon_n, n \geq 0\}$ *be a nonnegative sequence. Then, for* $b > 0$, $p \geq 0$,

$$\limsup_n \gamma_n^{-p} \sum_{k=1}^n \gamma_k^{p+1} e^{-b\sum_{j=k+1}^n \gamma_j} \epsilon_k \leq \frac{1}{C(b,p)} \limsup_n \epsilon_n \tag{91}$$

*for some constant* $C(b,p) > 0$.

*When* $p = 0$ *and define a positive sequence* $\{w_n\}$ *satisfying* $w_{n-1}/w_n = 1 + o(\gamma_n)$, *we have*

$$\sum_{k=1}^n \gamma_k e^{-b\sum_{j=k+1}^n \gamma_j} \epsilon_k = \begin{cases} O(w_n), & \text{if } \epsilon_n = O(w_n), \\ o(w_n), & \text{if } \epsilon_n = o(w_n). \end{cases} \tag{92}$$

**Lemma G.7** (Fort (2015) Lemma 5.10). *For any matrices* $A, B, C$,

$$\|ABA^T - CBC^T\| \leq \|A - C\|\|B\|(\|A\| + \|C\|). \tag{93}$$

## G.2 Asymptotic Results of Single-Timescale SA

Consider the stochastic approximation in the form of

$$z_{n+1} = z_n + \gamma_{n+1} G(z_n, X_{n+1}). \tag{94}$$

Let $\mathbf{K}_z$ be the transition kernel of the underlying Markov chain $\{X_n\}_{n \geq 0}$ with stationary distribution $\pi(z)$ such that $g(z) \triangleq \mathbb{E}_{X \sim \pi(z)}[G(z, X)]$ with domain $\mathcal{O} \subseteq \mathbb{R}^d$. Define an operator $\mathbf{K}_z f$ for any function $f : \mathcal{N} \to \mathbb{R}^D$ such that

$$(\mathbf{K}_z f)(i) = \sum_{j \in \mathcal{N}} f(j) \mathbf{K}_z(i, j). \tag{95}$$

Assume that

C1. W.p.1, the closure of $\{z_n\}_{n \geq 0}$ is a compact subset of $\mathcal{O}$.

C2. $\gamma_n = \gamma_0 / n^a, a \in (1/2, 1]$.

C3. Function $g$ is continuous on $\mathcal{O}$ and there exists a non-negative $C^1$ function $w$ and a compact set $\mathcal{K} \subset \mathcal{O}$ such that
- $\nabla w(z)^T g(z) \leq 0$ for all $z \in \mathcal{O}$ and $\nabla w(z)^T g(z) < 0$ if $z \notin \mathcal{K}$;
- the set $S \triangleq \{z \mid \nabla w(z)^T g(z) = 0\}$ is such that $w(S)$ has an empty interior;

C4. For every $z$, there exists a solution $m_z : \mathcal{N} \to \mathbb{R}^d$ for the following Poisson equation

$$m_z(i) - (\mathbf{K}_z m_z)(i) = G(z, i) - g(z) \tag{96}$$

for any $i \in \mathcal{N}$; for any compact set $\mathcal{C} \subset \mathcal{O}$,

$$\sup_{z \in \mathcal{C}, i \in \mathcal{N}} \|(\mathbf{K}_z m_z)(i)\| + \|m_z(i)\| < \infty \tag{97}$$

and there exist a continuous function $\phi_{\mathcal{C}}, \phi_{\mathcal{C}}(0) = 0$, such that for any $z, z' \in \mathcal{C}$,

$$\sup_{i \in \mathcal{N}} \|(\mathbf{K}_z m_z)(i) - (\mathbf{K}_{z'} m_{z'})(i)\| \leq \phi_{\mathcal{C}}(\|z - z'\|). \tag{98}$$

C5. Denote by $-r$ the largest real part of the eigenvalues of the Jacobian matrix $\nabla g(z^*)$ and assume $r > \frac{\mathbb{1}_{\{a=1\}}}{2}$.

C6. For every $z$, there exists a solution $Q_z : \mathcal{N} \to \mathbb{R}^{d \times d}$ for the following Poisson equation

$$Q_z(i) - (\mathbf{K}_z Q_z)(i) = F(z, i) - \mathbb{E}_{j \sim \pi(z)}[F(z, j)] \tag{99}$$

for any $i \in \mathcal{N}$, where

$$F(z, i) \triangleq \sum_{j \in \mathcal{N}} m_z(j) m_z(j)^T \mathbf{K}_z(i, j) - (\mathbf{K}_z m_z)(i)(\mathbf{K}_z m_z)(i)^T. \tag{100}$$

For any compact set $\mathcal{C} \subset \mathcal{O}$,

$$\sup_{z \in \mathcal{C}, i \in \mathcal{N}} \|Q_z(i)\| + \|(\mathbf{K}_z Q_z)(i)\| < \infty \tag{101}$$

and there exist $p, C_{\mathcal{C}} > 0$, such that for any $z, z' \in \mathcal{C}$,

$$\sup_{i \in \mathcal{N}} \|(\mathbf{K}_z Q_z)(i) - (\mathbf{K}_{z'} Q_{z'})(i)\| \leq C_{\mathcal{C}} \|z - z'\|^p. \tag{102}$$

**Theorem G.8** (Delyon et al. (1999) Theorem 2)**.** *Consider* (94) *and assume C1 - C4. Then, w.p.1,* $\limsup_n d(z_n, S) = 0$.

**Theorem G.9** (Fort (2015) Theorem 2.1 & Proposition 4.1)**.** *Consider* (94) *and assume C1 - C6. Then, given the condition that $z_n$ converges to one point $z^* \in S$, we have*

$$\gamma_n^{-1/2}(z_n - z^*) \xrightarrow[n \to \infty]{dist.} N(0, \mathbf{V}), \tag{103}$$

*where*

$$\mathbf{V}\left(\frac{\mathbb{1}_{\{b=1\}}}{2}\mathbf{I} + \nabla g(z^*)^T\right) + \left(\frac{\mathbb{1}_{\{b=1\}}}{2}\mathbf{I} + \nabla g(z^*)\right)\mathbf{V} + \mathbf{U} = 0, \tag{104}$$

*and*

$$\mathbf{U} \triangleq \sum_{i \in \mathcal{N}} \mu_i \left(m_{z^*}(i) m_{z^*}(i)^T - (\mathbf{K}_{z^*} m_{z^*})(i)(\mathbf{K}_{z^*} m_{z^*})(i)^T\right). \tag{105}$$

### G.3 Asymptotic Results of Two-Timescale SA

For the two-timescale SA with iterate-dependent Markov chain, we have the following iterations:

$$\mathbf{z}_{n+1} = \mathbf{z}_n + \beta_{n+1} G_1(\mathbf{z}_n, \mathbf{y}_n X_{n+1}), \tag{106a}$$

$$\mathbf{y}_{n+1} = \mathbf{y}_n + \gamma_{n+1} G_2(\mathbf{z}_n, \mathbf{y}_n, X_{n+1}), \tag{106b}$$

with the goal of finding the root $(\mathbf{z}^*, \mathbf{y}^*)$ such that

$$g_1(\mathbf{z}^*, \mathbf{y}^*) = \mathbb{E}_{X \sim \boldsymbol{\mu}}[G_1(\mathbf{z}^*, \mathbf{y}^*, X)] = 0, \quad g_2(\mathbf{z}^*, \mathbf{y}^*) = \mathbb{E}_{X \sim \boldsymbol{\mu}}[G_2(\mathbf{z}^*, \mathbf{y}^*, X)] = 0. \tag{107}$$

We present here a simplified version of the assumptions for single-valued functions $G_1, G_2$ that are necessary for the almost sure convergence result in Yaji & Bhatnagar (2020, Theorem 4). The original assumptions are intended for more general set-valued functions $G_1, G_2$.

(B1) The step sizes $\beta_n \triangleq n^{-b}$ and $\gamma_n \triangleq n^{-a}$, where $0.5 < a < b \leq 1$.

(B2) Assume the function $G_1(\mathbf{z}, \mathbf{y}, X)$ is continuous and differentiable with respect to $\mathbf{z}, \mathbf{y}$. There exists a positive constant $L_1$ such that $\|G_1(\mathbf{z}, \mathbf{y}, X)\| \leq L_1(1 + \|\mathbf{z}\| + \|\mathbf{y}\|)$ for every $\mathbf{z} \in \mathbb{R}^{d_1}, \mathbf{y} \in \mathbb{R}^{d_2}, X \in \mathcal{N}$. The same condition holds for the function $G_2$ as well.

(B3) Assume there exists a function $\rho : \mathbb{R}^{d_1} \to \mathbb{R}^{d_2}$ such that the following three properties hold: (i) $\|\rho(\mathbf{z})\| \leq L_2(1 + \|\mathbf{z}\|)$ for some positive constant $L_2$; (ii) the ODE $\dot{\mathbf{y}} = g_2(\mathbf{z}, \mathbf{y})$ has a globally asymptotically stable equilibrium $\lambda(\mathbf{z})$ such that $g_2(\mathbf{z}, \rho(\mathbf{z})) = 0$. Additionally, let $\hat{g}_1(\mathbf{z}) \triangleq g_1(\mathbf{z}, \rho(\mathbf{z}))$, there exists a set of disjoint roots $\Lambda \triangleq \{\mathbf{z}^* : \hat{g}_1(\mathbf{z}^*) = 0\}$, which is the set of globally asymptotically stable equilibria of the ODE $\dot{\mathbf{z}} = \hat{g}_1(\mathbf{z})$.

(B4) $\{X_n\}_{n \geq 0}$ is an iterate-dependent Markov process in finite state space $\mathcal{N}$. For every $n \geq 0, P(X_{n+1} = j | \mathbf{z}_m, \mathbf{y}_m, X_m, 0 \leq m \leq n) = P(X_{n+1} = j | \mathbf{z}_n, \mathbf{y}_n, X_n = i) = \mathbf{P}_{i,j}[\mathbf{z}_n, \mathbf{y}_n]$, where the transition kernel $\mathbf{P}[\mathbf{z}, \mathbf{y}]$ is continuous in $\mathbf{z}, \mathbf{y}$, and the Markov chain generated by $\mathbf{P}[\mathbf{z}, \mathbf{y}]$ is ergodic so that it admits a stationary distribution $\boldsymbol{\pi}(\mathbf{z}, \mathbf{y})$, and $\boldsymbol{\pi}(\mathbf{z}^*, \rho(\mathbf{z}^*)) = \boldsymbol{\mu}$.

(B5) $\sup_{n \geq 0}(\|\mathbf{z}_n\| + \|\mathbf{y}_n\|) < \infty$ a.s.

Yaji & Bhatnagar (2020) included assumptions A1 - A9 and A11 for the following Theorem G.10. We briefly show the correspondence of our assumptions (B1) - (B5) and theirs: (B1) with A5, (B2) with A1 and A2, (B3) with A9 and A11, (B4) with A3 and A4, and (B5) with A8. Given that our two-timescale SA framework (106) excludes additional noises (setting them to zero), A6 and A7 therein are inherently met.

**Theorem G.10** (Yaji & Bhatnagar (2020) Theorem 4). *Under Assumptions (B1) - (B5), iterations* $(\mathbf{z}_n, \mathbf{y}_n)$ *in* (106) *almost surely converge to a set of roots, i.e.,* $(\mathbf{z}_n, \mathbf{y}_n) \to \bigcup_{\mathbf{z}^* \in \Lambda}(\mathbf{z}^*, \rho(\mathbf{z}^*))$ *a.s.*

## H Additional Simulation Results

### H.1 Binary Classification on Additional Datasets

In this part, we perform the binary classification task as in Section 4 on additional datasets, i.e., *a9a* (with 123 features) and *splice* (with 60 features) from LIBSVM (Chang & Lin, 2011). Figure 4 provides the performance ordering of different $\alpha$ values, and we empirically demonstrate that the curves with $\alpha \geq 5$ still outperform the *i.i.d.* counterpart. Additionally, Figure 5 compare cases (i) - (iii) under both *a9a* and *splice* datasets, and case (i) consistently perform the best.

### H.2 Non-convex Linear Regression

We further test SGD-SRRW and SHB-SRRW algorithms with a non-convex function to demonstrate the efficiency of our SA-SRRW algorithm beyond the convex setting. In this task, we simulate the following linear regression problem in Khaled & Richtárik (2023) with non-convex regularization

$$\min_{\boldsymbol{\theta} \in \mathbb{R}^d} \left\{ f(\boldsymbol{\theta}) \triangleq \frac{1}{N} \sum_{i=1}^{N} l_i(\boldsymbol{\theta}) + \kappa \sum_{j=1}^{d} \frac{\boldsymbol{\theta}_j^2}{\boldsymbol{\theta}_j^2 + 1} \right\} \tag{108}$$

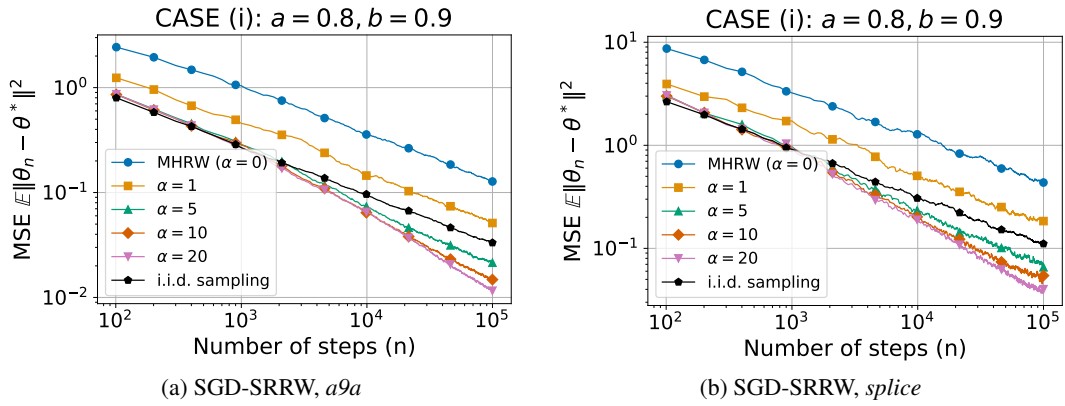

(a) SGD-SRRW, *a9a*    (b) SGD-SRRW, *splice*

Figure 4: Simulation results with various $\alpha$ values in *a9a* and *splice* datasets.

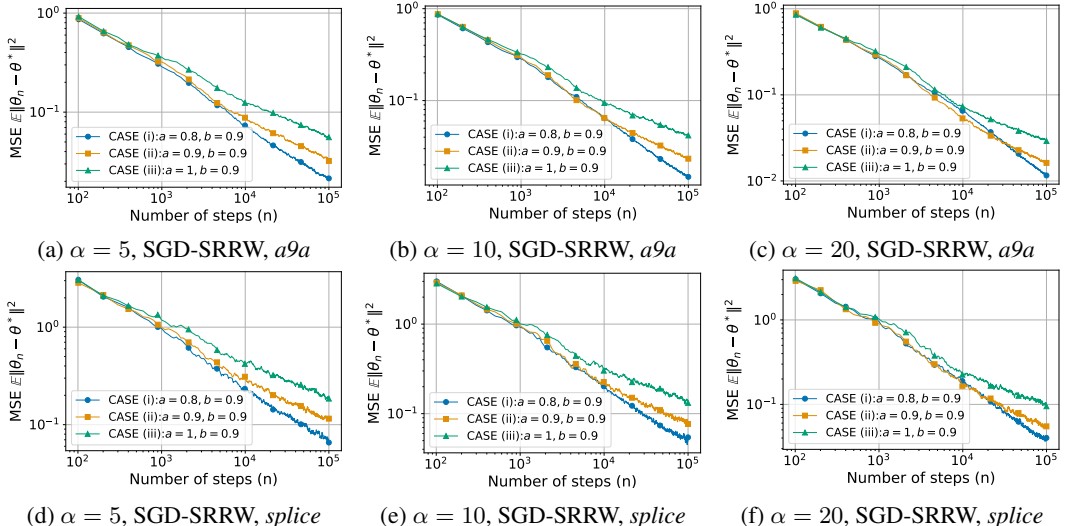

(a) $\alpha = 5$, SGD-SRRW, *a9a*    (b) $\alpha = 10$, SGD-SRRW, *a9a*    (c) $\alpha = 20$, SGD-SRRW, *a9a*

(d) $\alpha = 5$, SGD-SRRW, *splice*    (e) $\alpha = 10$, SGD-SRRW, *splice*    (f) $\alpha = 20$, SGD-SRRW, *splice*

Figure 5: Performance comparison among cases (i) - (iii) for $\alpha \in \{5, 10, 20\}$ in *a9a* and *splice* datasets.

where the loss function $l_i(\boldsymbol{\theta}) = \|\mathbf{s}_i^T \boldsymbol{\theta} - y_i\|^2$ and $\kappa = 1$, with the data points $\{(\mathbf{s}_i, y_i)\}_{i \in \mathcal{N}}$ from the *ijcnn1* dataset of LIBVIM (Chang & Lin, 2011). We still perform the optimization over the wikiVote graph, as done in Section 4.

The numerical results for the non-convex linear regression taks are presented in Figures 6 and 7, where each experiment is repeated 100 times. Figures 6a and 6b show that the performance ordering across different $\alpha$ values is still preserved for both algorithms over almost all time, and curves for $\alpha \geq 5$ outperform that of the *i.i.d.* sampling (in black) under the graph topological constraints. Additionally, among the three cases examined at identical $\alpha$ values, Figures 7a - 7c confirm that case (i) performs consistently better than the other two cases, implying that case (i) can even become the best choice for non-convex distributed optimization tasks.

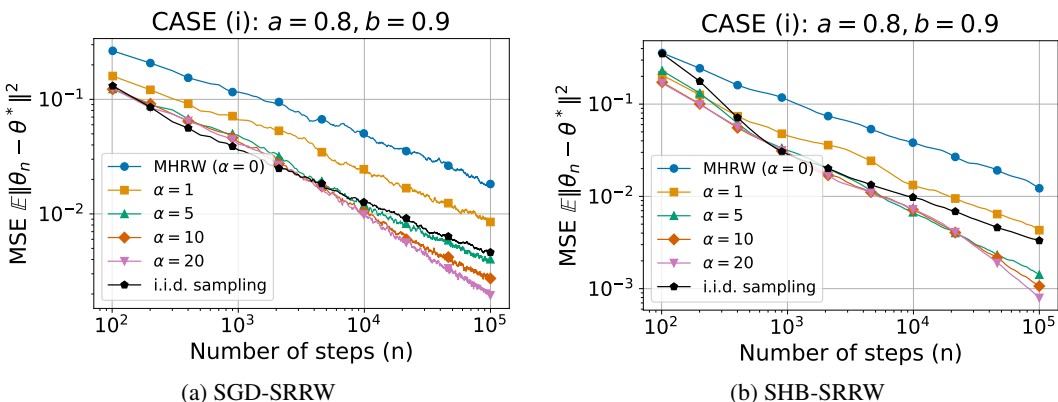

Figure 6: Simulation results for non-convex linear regression under case (i) with various $\alpha$ values.

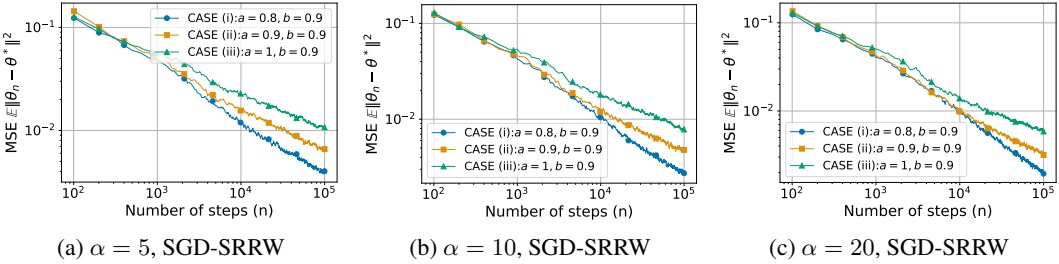

Figure 7: Performance comparison among cases (i) - (iii) for non-convex regression.

