# OpenReview forum: "Accelerating Distributed Stochastic Optimization via Self-Repellent Random Walks"
_ICLR.cc/2024/Conference — ICLR 2024 oral_

### Official Review · Reviewer_DxLx · 2023-10-28

**Soundness:** 3 good
**Presentation:** 3 good
**Contribution:** 3 good
**Rating:** 8
**Confidence:** 3

**Summary:**

This paper studied a family of distributed stochastic optimization algorithms where gradients
are sampled by a token traversing a network of agents using the Self-Repellent Radom Walk (SRRW).

This paper generalized the previous SRRW by introducing an additional iterate to update the empirical distribution so the resulting algorithm shares a similar pattern with the two-time-scale stochastic approximation (SA).

The author then provides some theoretical understanding of the proposed algorithm, including
- (1) almost surely convergence of two iterates
- (2) central limit theorem (CLT) of parameter $\theta_n$
- (3) order analysis on the asymptotic variance matrices for three different time-scales choices.

**Strengths:**

The paper is well-written and solid. I like reading the paper.

The paper contains rich theoretical analysis, which investigates deeply about the properties of proposed methods. The acceleration effect of the proposed method is indeed new and interesting (at least to me).

Most of the proof in the appendix seems to be correct, but I didn’t check very carefully.

The simulation validates the theoretical finding in the paper.

**Weaknesses:**

A concern in my mind is that the paper assumes $X$ takes values in a finite state space.
The decentralized optimization where only a single agent is active each time is a particular example.
Are there any other examples where $X$ takes finite value?

In my opinion, for a general SA method, the randomness or the noise could be various.
Note that the Poisson method used in the proof is not limited to discrete randomness.
Is it possible to extend the analysis to a more general setting where $X$ could be a continuous random variable?
If not, where is the main difficulty?

**Questions:**

1. Under the paragraph below (3), the author tries to explain why SRRW gets its name. More specifically, they said `` if node $j$ has been visited more often, so far, the entry $x_j$ becomes larger (than target value $\mu_j$ )’’. I didn’t understand the logic behind it. If node $j$ has been visited more often, I think $x_j$, as the coordinate of a distribution vector, should be close to $\mu_j$. I can’t tell whether $x_j$ is larger than $\mu_j$ or not.  However, I acknowledge that the nonlinear kernel in (3) would encourage the algorithm to visit the states that are less visited, based on its expression.

2. The newly introduced iterate in (4b) generalizes the original algorithm and allows us to treat the resulting algorithm as a two-time-scale SA. However, I didn’t see much motivation to introduce the new iterate. Could the author elaborate more on the motivation?

3. If we set $\alpha=\infty$, from (6), the variance matrix would be zero. Could I say, in this case, $x_n$ actually converges to $\mu$ at a faster rate than $\gamma_n$ so that the CLT degenerates to zero? How to explain the intuition behind it?

4. From the simulation, it seems the larger $\alpha$, the faster convergence. Is there any suggestion for a practical choice of $\alpha$?

5. If the $\alpha$ is also updated, for example, we use $\alpha_t$ at the iterate $t$ that increases at a carefully selected rate in advance (or perhaps could be updated using a third iterate). Can we accelerate the convergence rate of $\theta_n$ so that its mean square error is faster than $\beta_n$?

I will increase my point to 8 if most of my questions are addressed.

------------------------------------ Post rebuttal --------------

Thanks for the authors' clarification.  Most of my concerns and questions are well addressed, so I increase my point to 8.
However, there are still two points I don't agree with the author.

One point I disagree with is that I think when $\alpha \to \infty$, the kernel (3) is still well defined.
In this case, I think $K_{ij}(x) = 1$ if and only if $x_j/\mu_j = \arg\min_{l} x_l/\mu_l $.
As you can see, this kernel is reduced to a deterministic transition.
I guess this reduction makes the system no longer random so that the CLT degenerates.
One property of this generation is that the variance matrix would converge to zero.
This actually is a good property from my perspective because it means that one could get a convergence rate faster than the step size.

Let's do a simpler thought experiment. Let $X_t \sim \sqrt{\beta_t} \cdot N(0, c_t )$.
If the variance $c_t$ is non-zero (so that $c_t \to 1$ w.l.o.g.), then the MSE of $X_t$ is dominated by the variance and is of order $\beta_t$.
If the variance is (nearly) zero (say $c_t \to 0$), then the MSE is dominated by a higher-order term and we have $E |X_t|^2$ converge faster than $\beta_t$.

Back to this paper, I guess the counterpart of $c_t$ in this paper is $1/\alpha$. So, if we let $\alpha \to \infty$, we would have $c_t \to 0$. This is the reason why I ask whether increasing $\alpha$ would fasten the convergence. The author's response works only when the optimal $\alpha^{\star}$ exists and is finite. However, the particular case I am interested in is when $\alpha^{\star} = \infty$.
I think that exploring the case where  $\alpha^{\star} = \infty$ detailedly is worth a more detailed examination.
It is fine not to discuss this in this paper.

---

> ### Author Response · Authors · 2023-11-17
> **Response to Reviewer DxLx (1/2)**
>
> We appreciate the reviewer for the detailed comments. Please find our replies to your questions below.
>
> >### (Q1). Under the paragraph below (3), the author tries to explain why SRRW gets its name. More specifically, they said ''if node $j$ has been visited more often so far, the entry $x_j$ becomes larger (than target value $\\mu_j$ )''. I didn’t understand the logic behind it. If node $j$ has been visited more often, I think $x_j$, as the coordinate of a distribution vector, should be close to $\\mu_j$. I can’t tell whether $x_j$ is larger than $\\mu_j$ or not. However, I acknowledge that the nonlinear kernel in (3) would encourage the algorithm to visit the states that are less visited, based on its expression.
>
> Answer: Your interpretation ''$x_j$ should be close to $\\mu_j$’ is indeed accurate in the context of long-term behavior, where the empirical distribution $\\mathbf{x}$ converges towards the target distribution $\\mu$ almost surely. However, the point of confusion seems to arise from our description 'if node $j$ has been visited more often so far'. To clarify, when saying 'node $j$ has been visited more often', we mean that the node $j$ has been visited more often than other nodes (and we have improved this explanation in the revision). In this context, according to the definition of empirical distribution, the entry $[\\mathbf{x}\_n]\_j$ – representing the proportion of visits to node $j$ at time $n$ – can be (temporarily) larger than its target value $\\mu_j$. In this case, node $j$ will be chosen with smaller probability at time $n+1$ by the SRRW kernel with $\\mathbf{x}\_n$ in place of $\\mathbf{x}$ in (3). Note that, eventually, from the SRRW convergence result, all these $[\\mathbf{x}\_n]\_j$ will converge to the target $\\mu\_j$ as n increases, as you pointed out already. In some sense, SRRW tries to exploit such temporal fluctuation (while keeping the stationary behavior the same) to 'force' the random walker to reduce such temporal fluctuation (deviation of $\\mathbf{x}$ at time $n$ from the target $\\mu$), which translates to smaller sampling variance.
>
> >### (Q2). The newly introduced iterate in (4b) generalizes the original algorithm and allows us to treat the resulting algorithm as a two-time-scale SA. However, I didn’t see much motivation to introduce the new iterate. Could the author elaborate more on the motivation?
>
> Answer: We thank the reviewer for commenting on the SRRW iterate (4b). This SRRW iterate generalizes the step size $\\gamma_n$ from $1/(n+1)$ to a general form $1/(n+1)^a$  for $a\\in(0.5,1]$, which is crucial for enhancing the algorithm’s flexibility and performance. This generalization allows for the exploration of the step sizes $\\beta_n=1/(n+1)^b =o(\\gamma_n)$ (case(i)), which is suggested as the best algorithmic choice among all three cases in simulation. Without this generalization, i.e., if we stick to the original step size of $\\gamma_n=1/(n+1)$ as in [Doshi et al. 2023], the case (i) becomes vacuous as it is impossible to choose the step size $\\beta_n=o(1/n)$ (such step size becomes summable, fundamentally violating the condition for the step size in any stochastic approximation setting). To reflect this response, we have added a new footnote #7 in the revision to elaborate on the motivation for the generalized step size in the SRRW iterates.
>
> >Doshi, V., Hu, J., & Eun, D. Y. (2023). Self-Repellent Random Walks on General Graphs--Achieving Minimal Sampling Variance via Nonlinear Markov Chains. International Conference on Machine Learning. PMLR, 2023.
>
> >### (Q3). If we set $\\alpha=\\infty$, from (6), the variance matrix would be zero. Could I say, in this case, $\\mathbf{x}\_n$ actually converges to $\\mu$ at a faster rate than $\\gamma\_n$ so that the CLT degenerates to zero? How to explain the intuition behind it?
>
> Answer: It is crucial to clarify that our CLT results, as presented in Theorem 3.3, are specifically tailored for any 'given’ $\\alpha < \infty$, where $\\mathbf{x}_n$ converges to $\\mu$ at the exact asymptotic rate $\\gamma_n$ and the resulting covariance tends to zero for larger $\\alpha$ without actually reaching it. It's important to understand that our analysis does not extend to the case of $\\alpha=\\infty$ since the SRRW kernel in (3) is undefined. Consequently, the behavior of $\\mathbf{x}\_n$ in such a setting, including its convergence rate to $\\mu$ and the resulting asymptotic covariance, falls outside the scope of our theoretical framework.

---

> ### Author Response · Authors · 2023-11-17
> **Response to Reviewer DxLx (2/2)**
>
> >### (Q4). From the simulation, it seems the larger $\\alpha$, the faster convergence. Is there any suggestion for a practical choice of $\\alpha$?
>
> Answer: In practice, we recommend using moderate values of $\\alpha$, e.g., $\\alpha=5$. Since covariance in case (i) decreases at a rate of $O(1/\\alpha^2)$, even moderate values of $\\alpha$ are sufficient to achieve small errors, e.g., $\\alpha=5$ in Figure 2(c). Moreover, it’s important to consider the potential drawbacks of setting $\\alpha$ too large. As illustrated in Figure 2(b), larger $\\alpha$ values could slow down convergence in the early stage of the training process. For example, when $\\alpha$ is set to $10$ or $20$, performance is slightly inferior to $\\alpha=5$ until $n\\approx 5×10^4$.
>
> >### (Q5). If the $\\alpha$ is also updated, for example, we use $\\alpha_n$ at the iterate $n$ that increases at a carefully selected rate in advance (or perhaps could be updated using a third iterate). Can we accelerate the convergence rate of $\\theta_n$ so that its mean square error is faster than $\\beta_n$?
>
> Answer: Assume $\\alpha\_n$ is updated through a third iterate and converges to some finite value $\\alpha^*$, we speculate that the asymptotic rate of MSE cannot go faster than $\\beta\_n$. Our current analysis, as detailed in Theorem 3.3, addresses scenarios with a fixed $\\alpha$ value. Here, the influence of the SRRW sequence $\\{X\_n\\}$ is encapsulated within the matrix $\\mathbf{U}$, which contributes to the asymptotic covariance $\\mathbf{V}\_{\\theta}$ of $\\theta\_n$. Introducing a time-varying $\\alpha\_n$ adds a new dimension to the system, potentially transforming the SA-SRRW algorithm, which we analyze via *two-timescale* SA setting, into a *three-timescale* SA. The CLT analysis of such a complex scenario remains unexplored and presents an interesting avenue for future research. Intuitively speaking, in the asymptotic regime, as $\\alpha\_n$ already approaches $\\alpha^*$, the behavior of sequence $\\{X\_n\\}$ would likely be similar to that driven by SRRW with constant $\\alpha^*$. This means that while varying $\\alpha_n$ impacts the asymptotic covariance $\\mathbf{V}\_{\\theta}$ through matrix $\\mathbf{U}$, the convergence rate of $\\theta\_n$, determined by $\\beta\_n^{-1/2}$, would remain unchanged. Hence, even with a dynamic $\\alpha\_n$, the MSE of $\\theta\_n-\\theta^*$ is still expected to converge at a rate tied to $\\beta_n$, according to our current theoretical framework.
>
> >### (Q6): Are there any other examples where $X$ takes finite value? In my opinion, for a general SA method, the randomness or the noise could be various. Note that the Poisson method used in the proof is not limited to discrete randomness. Is it possible to extend the analysis to a more general setting where $X$ could be a continuous random variable? If not, where is the main difficulty?
>
> Answer: Our SA-SRRW algorithm is versatile and not limited to decentralized optimization employed with token algorithm only. It is equally applicable to scenarios in stochastic optimization where the entire dataset is fully accessible. This application is particularly prevalent in the machine learning domain, such as in training neural networks using SGD or other accelerated algorithms. In these instances, the dataset forms a *complete graph with self-loops* so that $X$ can be independently and identically sampled from the dataset with marginal distribution $\\mu$, implying that these instances can also be transformed into the decentralized optimization in the broader sense. Here, the baseline Markov chain $\\mathbf{P}$, as indicated in the SRRW kernel (3), becomes a rank-1 matrix $\\mathbf{1}\\mathbf{\\mu}^T$.
>
> Meanwhile, $X$ cannot be a continuous random variable. The limitation here is not related to the Poisson method used in our proofs but is inherent to the structure of the SRRW employed in the SA method. Specifically, the concept of self-repellence in SRRW requires counting the visits to each state $X$ to form the empirical measure $\\mathbf{x}\_n$ at time $n$. This count is crucial for designing the SRRW kernel $\\mathbf{K}[\\mathbf{x}\_n]$. In a continuous state space, such a counting mechanism becomes infeasible since the chance of sampling the same $X$ in the continuous state space is measure zero, posing a significant challenge to the design and implementation of a continuous version of SRRW in our algorithm.

---

### Official Review · Reviewer_81Sf · 2023-10-31

**Soundness:** 3 good
**Presentation:** 3 good
**Contribution:** 2 fair
**Rating:** 6
**Confidence:** 4

**Summary:**

This paper deals with distributed stochastic optimization, with tokens sampled by self-repelling Markov chain. The authors proved the law of large numbers and central limit theorem for the optimization iterate errors, which are refinements of recent work of Doshi et al. The proposed algorithm achieves $1/\alpha^2$-rate, and the theoretical results are corroborated by the empircal study.

**Strengths:**

The paper studies rigorously a distributed stochastic optimization using self-repelling random walks. The paper is well-written, and I enjoyed reading it. I have also checked most theoretical results, and they appear to be correct. The rate $1/\alpha^2$ also appears to be new.

**Weaknesses:**

The paper is mostly motivated by the recent progress of Dochi et al., and it is not clear what is the "main" novelty of this paper compared to the previous work (fundamentally). It seems to me that most results are refinements of Dochi et al. (while I agree that the setting is slightly different.) The authors may add a paragraph to highlight the main "technical" novelty of this paper.

Also the parameter $\alpha$ measures the "heaviness" of the self-repelling random walk, which is basically a hyper-parameter. It seems to be a bit strange to quantify the errors using $Poly(1/\alpha)$ (provided that one sends $\alpha \to \infty$ and I do have concerns on the real application in this regime). Also the idea of using self-repelling random walk to accelerate optimization is not new, see https://arxiv.org/abs/2005.04507 (in which it was shown to outperform many existing algorithms.) The authors may think of citing the work and some references therein.

**Questions:**

See the Weakness.

**Details Of Ethics Concerns:**

Not available.

---

> ### Author Response · Authors · 2023-11-17
> **Response to Reviewer 81Sf (1/2)**
>
> Thank you for your comments and for the time and effort you put into reading, understanding and evaluating our paper. We now answer the question posed by the reviewer.
>
> >### (Q1). The paper is mostly motivated by the recent progress of Doshi et al., and it is not clear what is the "main" novelty of this paper compared to the previous work (fundamentally). It seems to me that most results are refinements of Doshi et al. (while I agree that the setting is slightly different.) The authors may add a paragraph to highlight the main "technical" novelty of this paper.
>
> Answer: While Doshi et al.’s analysis is foundational, it primarily utilizes existing CLT results for *single-timescale* SA with controlled Markov noise. Our work, in contrast, provides a first of its kind CLT result for the more complex *two-timescale* SA with controlled Markov noise (step-size cases (i) and (iii) fall into this category) marking a significant advancement. This CLT result is obtained via our original analysis detailed in Appendices D.2 and D.3, and does not build upon the analysis in Doshi et. al.. Without this, we cannot theoretically show the asymptotic covariance of the SA iterates $\\theta_n$ and prove that case (i) achieves $O(1/\\alpha^2)$ rate for its covariance and outperforms case (iii). The significance of this analysis is underscored right after Theorem 3.3 in our original paper, where we outline the technical challenges encountered when dealing with controlled Markov noise. In response to your valuable feedback, we have revised Contribution #2 in the introduction to delineate our technical innovations more clearly.
>
> >### (Q2). Also the parameter $\\alpha$ measures the "heaviness" of the self-repelling random walk, which is basically a hyper-parameter. It seems to be a bit strange to quantify the errors using $Poly(1/\\alpha)$ (provided that one sends $\\alpha\\to\\infty$ and I do have concerns on the real application in this regime).
>
> Answer: Thank you for highlighting the concerns regarding the hyper-parameter $\\alpha$. We would like to clarify its role and implications in our analysis in the following three aspects:
> - **Role of $\\alpha$ in Asymptotic Covariance**: Our CLT results in Theorem 3.3 enable us to quantify how the asymptotic covariance of the error varies with $\\alpha\\geq 0$, for any given finite $\\alpha$. It is important to note that CLT analysis does not extend to the case of $\\alpha\\to\\infty$, even though the form of the asymptotic covariance matrix allows one to evaluate it at $\\alpha\ = \\infty$. This is because setting $\\alpha$ to infinity would disrupt the continuity of the SRRW kernel $\\mathbf{K}[\\mathbf{x}]$ as stated in (3), and it would no longer be well-defined for any $\\mathbf{x}\\in\\text{Int}{\\Sigma}$. The continuity of $\\mathbf{K}[\\mathbf{x}]$ with respect to $\\mathbf{x}$ is essential in our CLT analysis.
>
> - **Practical Implications of $\\alpha$**: In real-world applications, using large values of $\\alpha$ is not necessary. Our findings show that the asymptotic covariance in case (i) decreases at a rate of $O(1/\\alpha^2)$. This suggests that even moderate values of $\\alpha$ are sufficient to achieve small errors. For instance, in our empirical results shown in Figure 2(c), setting $\\alpha$ to $5$ already yields satisfactory performance. Increasing $\\alpha$ from $5$ to $20$ does not result in significant improvements in mean square error (MSE), as the MSE was already small enough for $\\alpha=5$.
>
> - **Drawbacks of Excessively Large $\\alpha$**: It's also important to consider the potential drawbacks of setting $\\alpha$ too large. As illustrated in Figure 2(b), larger $\\alpha$ values can actually slow down convergence in the initial stages of the training process. For example, when $\\alpha$ is set to $10$ or $20$, performance is slightly inferior to $\\alpha=5$ until $n\\approx 5×10^4$.

---

> > ### Comment · Reviewer_81Sf · 2023-11-21
> >
> > I would thank the authors for the detailed response. The score is increased to 6.

---

> ### Author Response · Authors · 2023-11-17
> **Response to Reviewer 81Sf (2/2)**
>
> >### (Q3). Also the idea of using self-repelling random walk to accelerate optimization is not new, see https://arxiv.org/abs/2005.04507 (in which it was shown to outperform many existing algorithms.) The authors may think of citing the work and some references therein.
>
> Answer: We appreciate the reviewer for directing our attention to Guo et al.'s work using 'self-repelling random walk’ to accelerate optimization. We have included this reference in footnote #3 of our revised manuscript to acknowledge its relevance to our research domain.
>
> Our study never claims to be the first paper to incorporate the concept of `self-repellence’. Additionally, our study diverges significantly from Guo et al.'s work in its application and theoretical framework. The novelty of our work lies in the strategic control of the noise sequence $\\{X_n\\}$ in the token algorithm (4c) within the context of distributed learning adhering to arbitrary graphical constraints (general graphs), which is not a consideration in Guo et al.’s study where the underlying graph is more of 1-D nature (increasing vs. decreasing, instead of choosing one of neighbors in an arbitrary graph). Our approach also leverages SRRW from the Markov Chain Monte Carlo (MCMC) literature in controlling the noise sequence $\\{X\_n\\}$ in the function $H(\\theta\_n,X\_{n+1})$ as outlined in (4c). This is markedly different from Guo et al.’s approach, which integrates self-repellence into gradient descent iterates themselves for escaping saddle points.
>
> Finally, It’s important to note that our version of self-repellence and that of Guo et al. are not directly comparable, as they serve different purposes within their respective algorithms, e.g., one is about the perturbation $\\theta_n$, and the other is about the choice of $X_{n+1}$ in the function $H(\\theta\_n,X\_{n+1})$. However, an intriguing possibility is the combination of both versions of self-repellence approaches to improve algorithmic performance.

---

### Official Review · Reviewer_MZm1 · 2023-11-01

**Soundness:** 4 excellent
**Presentation:** 4 excellent
**Contribution:** 3 good
**Rating:** 8
**Confidence:** 4

**Summary:**

The present paper consider a family of stochastic algorithms of the following form. There exists a finite space $\mathcal{N}$ and some kind of "random walk" $\{X_n\}_{n\in\N}$ over this set. One then takes $\theta_0\in\mathbb{R}^d$ and a function $H:\mathbb{R}^d\times \mathcal{N}\to \mathbb{R}^d$ and sets

$$\theta_{n+1} = \theta_n + \beta_n\,H(\theta_n,X_n).$$

One particular example is when $X_n\sim \mu$ and $H(\theta,i) = -\nabla f(\theta,i)$ for some $f$. Then the above is basically a form of stochastic gradient descent for the function

$$F(\theta) :=\sum_i \mu_i\,f(\theta,i).$$

The authors say that $X_n$ may be thought of as a "token", so they are considering token distributed optimization methods.

The authors are especially interested in the case where $X_n$ is a stochastic process called "self-repellent random walk". Basically, this kind of RW have asymptotic distribution $\mu$, but they are designed to be less likely to jump to states that have already been visited many times. A recent paper by Doshi et al. showed that such random walks lead to smaller-variance Monte Carlo estimates of expectations with respect to $\mu$. In fact, the variance in this case decreases like $\alpha^{-1}$, where $\alpha>0$ measures the amount of self-repulsion.

The present paper builds on this to show a remarkable result. Namely, under suitable conditions, $\theta_n$ converges to a fixed point $\theta_*$ (in expectation) of the iteration, and $\theta_n - \theta_*$ has variance decrease when $\alpha$ grows. In fact, the variance goes down like $1/\alpha^2$ for large $\alpha$. In particular, this implies that self-repellent walks are "even better" for optimization or fixed point computations than for Monte-Carlo integration. However, this main result requires on tuning $\beta_n$ to be smaller than a parameter describing the rate of change of the "self-repulsion measure". In fact, the authors show that, when this is not the case, it may be that a large $\alpha$ will not help at all. The paper's proofs build on Doshi et al and rely on stochastic approximation methods.

**Strengths:**

The paper describes a surprising phenomenon -- even considering the paper by Doshi et al. It gives precise asymptotic results. Their small simulation study observe gains in finite time. The paper is very clear and convincing.

**Weaknesses:**

All results are asymptotic. It is not clear (theoretically) whether there are gains for finite time, before eg. the token can visit most elements of $\mathcal{N}$. The analysis requires some *a priori* assumptions about the ODE invoked for stochastic approximation. The analysis is heavily indebted to Doshi et al.

**Questions:**

I only have one very open ended question: is there any hope of getting finite sample bounds?

---

> ### Author Response · Authors · 2023-11-17
> **Response to Reviewer MZm1**
>
> Our sincere thanks for the in-depth review of our paper. We now answer the question posed by the reviewer.
>
> >### (Q1). I only have one very open ended question: is there any hope of getting finite sample bounds?
>
> Answer: We think it’s very unlikely to obtain the finite sample bound for our SA-SRRW algorithm. In the new appendix B in the revision, we briefly demonstrate that the state-of-the-art non-asymptotic analyses all require globally Lipschitz mean field function for both iterates to derive the finite sample bounds. However, for our SA-SRRW algorithm (4), the mean field function $\\boldsymbol{\\pi}(\\mathbf{x})-\\mathbf{x}$ in the SRRW iterates (4b) is only locally Lipschitz continuous in $\\mathbf{x}\\in\\text{Int}(\\Sigma)$ (i.e., the interior of the probability simplex) and we provide a detailed explanation of the local Lipschitzness in appendix B. There does not exist a uniformly bounded Lipschitz constant for $\\boldsymbol{\\pi}(\\mathbf{x})-\\mathbf{x}$ in terms of $\\mathbf{x}\\in\\text{Int}(\\Sigma)$ . Thus, in this context, our SA-SRRW algorithm is unlikely to obtain finite sample bounds. Indeed, we believe that deriving any usable finite sample bound in this general area of two-timescale SA with controlled Markov noise for non-globally-Lipschitz kernels, would pose as one of important future direction in the literature, and clearly much beyond the scope of our paper now.

---

### Official Review · Reviewer_XrEW · 2023-11-06

**Soundness:** 3 good
**Presentation:** 3 good
**Contribution:** 3 good
**Rating:** 6
**Confidence:** 4

**Summary:**

This paper introduces a new distributed stochastic optimization algorithm, where the random walk guiding the sampling of gradients across the network uses Self-Repellent Random Walk (SRRW), a recently introduced family of nonlinear Markov chain. For MCMC sampling on a graph, SRRW has been shown to achieve a $O(1/\alpha)$ decrease in the asymptotic variance [Doshi et al, ICML 2023], where $\alpha$ is a hyperparameter of the chain; roughly speaking, given any base Markov chain, SRRW works by preferring transitions to states that were less visited in the past, and the larger the $\alpha$, the stronger this preference. Building on this insight, the authors in this paper use SRRW as the "token" gradient sampler in a distributed stochastic optimization setting.

**Strengths:**

The main contributions are as follows. Very interestingly, the authors show that given the "right" step-size setting (i.e. when the step-size for the optimization parameter $\theta_n$, $\beta_n$, is $o(\gamma_n)$, where $\gamma_n$ is the step-size for the update of the weighted empirical distribution $x_n$), the asymptotic variance of the $\theta_n$ (defined as $\frac{\theta_n - \theta^*}{\sqrt{\beta_n}}$) actually decays with the rate $O(1/\alpha^2),$ which is better than the $ O(1/\alpha)$decay rate for the (un)weighted empirical distribution $x_n$ in [Doshi et al., 2013]. An important result the authors show is that the asymptotic step-size ratio between $\beta_n$ and $\gamma_n$ is key; when $\gamma_n = \beta_n$ or wheren $\gamma_n = o(\beta_n)$, the asymptotic variance is only $O(1/\alpha)$. The key technical tool that achieves this is a novel analysis of two-time scale SA with controlled Markov noise. The theoretical results are also well-supported by simulation results.

Overall, this is a well-written paper and the idea of adding SRRW to distributed stochastic optimization is nice. In addition, the theoretical result about the influence of step-size ratio on the asymptotical covariance decay rate is also very interesting.

**Weaknesses:**

see questions below:

**Questions:**

There are some questions that I hope the authors can address.
1) Could the authors comment a little bit more on the stability of the iterates? This is assumed to always hold, but as the authors point out, practical tricks are required to ensure this, in which case the theoretical analysis might also need tweaks.
2) Are there any tradeoffs in picking $\alpha$? Should $\alpha$ always be as large as possible, or will that affect other aspects of the algorithm, e.g. stability?
3) While the analysis is on asymptotic covariance, could the authors comment on the likely finite-time convergence rate behavior of their SRRW-SA algorithm, and whether improvements (due to the SRRW) can be also shown in terms of finite-time convergence rates?
4) The authors suggest that $\beta_n = o(\gamma_n)$ is the best step-size choice if a $O(1/\alpha^2)$ decay rate in the asymptotic covariance is desired. However, if faster convergence is to be desired, a larger learning rate (i.e. larger $\beta_n$) should always be picked, i.e. we would want to pick the maximal possible step-size required by the assumptions in the paper, i.e. $\beta_n = O(1/n^{0.5+\epsilon})$ for some $\epsilon > 0$ as close to 0 as possible, in which case $\gamma_n$ can at best match the step-size of $\beta_n$, meaning that we can only get $O(1/\alpha)$ in the asymtotic covariance decay again. Could the authors comment on this possible issue?
5) It would be nice if the authors provided some intuition as to why the $\beta_n = o(\gamma_n)$ step-size ratio can yield the $O(1/\alpha^2)$ rate. Moreover, it would be nice if the authors spelled out precisely the dependence on the difference between the step-size exponents in their asymptotic rates (i.e. if  $\beta_n = n^{-b}$ and $\alpha_n = n^{-a}$, how the rate actually depends on the difference between $a$ and $b$). If this difference does not affect the asymptotic rate, it would be nice to provide insight into how this difference might play out in practice.

---

> ### Author Response · Authors · 2023-11-17
> **Response to Reviewer XrEW (1/3)**
>
> We thank the reviewer for their detailed comments. Following are our detailed responses to the questions posed.
>
>
> >### (Q1). Could the authors comment a little bit more on the stability of the iterates? This is assumed to always hold, but as the authors point out, practical tricks are required to ensure this, in which case the theoretical analysis might also need tweaks.
>
> Answer: We appreciate the reviewer's question regarding the stability of the iterates in our SA-SRRW algorithm, and we recognize the potential impact of practical tricks on our theoretical analysis. The stability of SRRW iterates $\\mathbf{x}_n$ in (4b) is always ensured through the truncation method in [1], where the detailed steps have been shown in [2]’s Appendix E with $\\gamma_n=1/(n+1)$. This analysis can be extended to generalized step size $\\gamma_n = 1/(n+1)^a$ for $a \\in (0.5,1]$ as well since it satisfies the step size assumption (A1) in [1].
>
> However, the stability of the SA iterates $\\theta_n$ presents a more complex challenge. This is due to its dependency on the SRRW iterates $\\mathbf{x}_n$. Currently, there is a lack of comprehensive stability analysis for $\\theta_n$ within the domain of two-timescale SA with controlled Markov noise. Even the latest result in this area, which provides only the almost sure convergence (no CLT result therein), still requires the stability assumption [3]. This situation mirrors the challenges faced in single-timescale SA literature, where establishing stability conditions often requires significant analytical effort, e.g., Chapter 4 in [4], [1], [5], [6].
>
> Therefore, while we ensure the stability of the SRRW iterates using established methods, the stability of $\\theta_n$ under the two-timescale framework remains an open question. Addressing this would not only require modifying our theoretical analysis to accommodate practical algorithmic adjustments but also contribute significantly to the broader field of two-timescale SA, which we acknowledge as an important aspect of future work.
>
> >[1]. Andrieu, C., Moulines, É., & Priouret, P. Stability of stochastic approximation under verifiable conditions. SIAM Journal on control and optimization, 44(1), 283-312, 2005.
> >
> >[2]. Doshi, V., Hu, J., & Eun, D. Y. Self-Repellent Random Walks on General Graphs--Achieving Minimal Sampling Variance via Nonlinear Markov Chains. International Conference on Machine Learning. PMLR, 2023.
> >
> >[3]. Yaji, V. G., & Bhatnagar, S. Stochastic recursive inclusions in two timescales with nonadditive iterate-dependent Markov noise. Mathematics of Operations Research, 45(4), 1405-1444, 2020.
> >
> >[4]. Borkar, V. Stochastic Approximation: A Dynamical Systems Viewpoint: Second Edition. Texts and Readings in Mathematics. Hindustan Book Agency, 2022.
> >
> >[5]. Fort, G., Moulines, E., Schreck, A., and Vihola, M. Convergence of markovian stochastic approximation with discontinuous dynamics. SIAM Journal on Control and Optimization, 54(2):866–893, 2016.
> >
> >[6]. Yaji, V. G., & Bhatnagar, S. Analysis of stochastic approximation schemes with set valued maps in the absence of a stability guarantee and their stabilization. IEEE Transactions on Automatic Control,65(3), 1100–1115, 2020.
>
> >### (Q2). Are there any tradeoffs in picking $\\alpha$? Should $\\alpha$ always be as large as possible, or will that affect other aspects of the algorithm, e.g. stability?
>
> Answer: While a larger $\\alpha$ does indeed favor a smaller asymptotic covariance as per our CLT, there are practical considerations that suggest a more balanced approach to choosing $\\alpha$. In practice, a moderate value of $\\alpha$ often strikes the right balance between reducing asymptotic covariance and maintaining efficient convergence in the initial phases of training. For instance, as observed in our simulations (Figure 2(b)), we noticed that when $\\alpha$ is set to larger values like 10 or 20, the performance initially lags behind that of $\\alpha=5$ until about $n\\approx 5×10^4$ iterations. In addition, asymptotic covariance in case (i) decreases at a rate of $O(1/\\alpha^2)$, implying that even moderate values of $\\alpha$ are sufficient to achieve small errors, e.g., $\\alpha=5$ in Figure 2(c).
>
> As explained in the previous response, the stability of SRRW iterates can always be guaranteed through truncation method in [1] for any $\\alpha<\\infty$. Thus, although we do not have the stability of $\\theta_n$, due to the robust behavior of SRRW iterates, we speculate that large $\\alpha$ is unlikely to compromise the stability of $\\theta_n$.

---

> ### Author Response · Authors · 2023-11-17
> **Response to Reviewer XrEW (2/3)**
>
> >### (Q3). While the analysis is on asymptotic covariance, could the authors comment on the likely finite-time convergence rate behavior of their SRRW-SA algorithm, and whether improvements (due to the SRRW) can be also shown in terms of finite-time convergence rates?
>
> Answer: The current non-asymptotic analysis of two-timescale SA with controlled Markov noise cannot be applied to our SA-SRRW algorithm. Specifically, all the state-of-the-art non-asymptotic analysis requires that the mean field functions of both iterates are globally Lipschitz [7-9]. This globally Lipschitzness condition is in place even for finite-time bounds in the singe-timescale SA [10-12]. However, the mean field function $\\boldsymbol{\\pi}(\\mathbf{x})-\\mathbf{x}$ of SRRW iterates (4b) exhibits only locally Lipschitz continuity, where $\\mathbf{x}$ is in the interior of the probability simplex. This is due to the polynomial form $(x_i/μ_i)^{-\\alpha}$ inside $\\boldsymbol{\\pi}(\\mathbf{x})$, which was elaborated on in Appendix D [2] as a necessary condition to ensure scale invariance and local information reliance. This distinction is critical as it places our SA-SRRW algorithm outside the existing non-asymptotic analysis. To address this, we include additional insights in the paragraph following (4) in our revised manuscript and provide a detailed discussion in the new appendix B.
>
> >[7]. Doan, T. T. Finite-time convergence rates of nonlinear two-time-scale stochastic approximation under markovian noise. arXiv preprint arXiv:2104.01627, 2021.
> >
> >[8]. Zeng, S., Doan, T. T., and Romberg, J. A two-time-scale stochastic optimization framework with applications in control and reinforcement learning. arXiv preprint arXiv:2109.14756, 2021.
> >
> >[9]. Doan, T. T. Nonlinear two-time-scale stochastic approximation convergence and finite-time performance. IEEE Transactions on Automatic Control, 2022.
> >
> >[10]. Chen, Z., Maguluri, S. T., Shakkottai, S., & Shanmugam, K. Finite-sample analysis of stochastic approximation using smooth convex envelopes. arXiv preprint arXiv:2002.00874, 2020.
> >
> >[11]. Sun, T., Sun, Y., & Yin, W. On markov chain gradient descent. Advances in neural information processing systems, 31, 2018.
> >
> >[12]. Even, M. Stochastic gradient descent under markovian sampling schemes. International Conference on Machine Learning, 2023.
>
> >### (Q4). The authors suggest that $\\beta_n=o(\\gamma_n)$ is the best step-size choice if a $O(1/\\alpha^2)$ decay rate in the asymptotic covariance is desired. However, if faster convergence is to be desired, a larger learning rate (i.e. larger $\\beta_n$) should always be picked, i.e. we would want to pick the maximal possible step-size required by the assumptions in the paper, i.e. $\\beta_n=O(1/n^{0.5+\\epsilon})$ for some $\\epsilon>0$ as close to 0 as possible, in which case $\\gamma_n$ can at best match the step-size of $\\beta_n$, meaning that we can only get $O(1/\\alpha)$ in the asymptotic covariance decay again. Could the authors comment on this possible issue?
>
> Answer: In our paper, $\\beta_n=o(\\gamma_n)$ is shown to be the best step-size choice in terms of achieving smaller asymptotic error (captured via asymptotic covariance). These errors are smallest for choices of $\\beta_n=O(1/n)$, and not $O(1/n^{0.5+\\epsilon})$ for some close-to-zero $\\epsilon$, since the step sizes are also the scaling factors under which the finite asymptotic variance is characterized. Thus, larger learning rates actually lead to larger asymptotic variances as well; even though the initial convergence (to a level set around the solution) could be quicker, larger learning rates result in larger variances (larger level sets) for large enough time. This is also demonstrated by the extreme case of constant step size, where, without additional averaging, the SGD iterates may converge quickly, but only to a level set (neighborhood) around the solution, and not to the solution itself, and it is well known that the size of such neighborhood around the solution critically depends on the (asymptotic) variance of the iterates. Therefore, selecting the largest possible learning rate is not necessarily the most desirable.

---

> ### Author Response · Authors · 2023-11-17
> **Response to Reviewer XrEW (3/3)**
>
> >### (Q5). It would be nice if the authors provided some intuition as to why the $\\beta_n=o(\\gamma_n)$ step-size ratio can yield the $O(1/\\alpha^2)$ rate. Moreover, it would be nice if the authors spelled out precisely the dependence on the difference between the step-size exponents in their asymptotic rates (i.e. if $\\beta_n=n^{-b}$ and $\\gamma_n=n^{-a}$, how the rate actually depends on the difference between $a$ and $b$). If this difference does not affect the asymptotic rate, it would be nice to provide insight into how this difference might play out in practice.
>
> Answer: For $\\beta_n=o(\\gamma_n)$ in case (i), the impact of the SRRW iterates on the SA iterates is primarily reflected in the correlation terms $\\mathbf{J}\_{12}(\\alpha)$ and $\\mathbf{J}\_{22}(\\alpha)$, as detailed in equation (55) and Lemma D.3 in Appendix D.2.2. These terms play a pivotal role in shaping the matrix $\\mathbf{U}\_{\\theta}(\\alpha)$ as follows:
>
> $$\\mathbf{U}\_{\\theta}(\\alpha)=\\mathbf{U}\_{22}-\\mathbf{U}_{21} (\\mathbf{J}\_{12}(\\alpha) \\mathbf{J}\_{22}(\\alpha)^{-1})^T-\\mathbf{J}\_{12}(\\alpha)\\mathbf{J}\_{22}(\\alpha)^{-1}\\mathbf{U}\_{12}+\\mathbf{J}\_{12}(\\alpha) \\mathbf{J}\_{22}(\\alpha)^{-1}\mathbf{U}\_{11}(\\mathbf{J}\_{12}(\\alpha) \\mathbf{J}\_{22}(\\alpha)^{-1})^T,$$
>
> where $\\mathbf{U}\_{ij}$ is defined in (9) for $i,j\\in\\{1,2\\}$. Through algebraic computations, we derive the $O(1/\\alpha^2)$ rate for matrix $\\mathbf{U}\_{\\theta}(\\alpha)$ and in turn for the asymptotic covariance $\\mathbf{V}\_{\\theta}^{(1)}(\\alpha)$.
>
> Regarding the difference in step-size exponents ($a$ and $b$ in $\\beta\_n=n^{-b}$ and $\\gamma\_n=n^{-a}$), our main CLT results in Theorem 3.3 indicate that for all three cases, the asymptotic covariances remain the same irrespective of the variations in $a$ and $b$. The key change lies in the asymptotic rates $\\beta\_n^{1/2}$ and $\\gamma\_n^{1/2}$ of the iterates. In practice, we indeed observe that for a fixed value $b$, the convergence speed in the initial training phase is affected by the different value of $a$. We take $b=0.9$ as an example. Choosing $a$ close to $0.5$ results in the slowest convergence since the SRRW iterates $\\mathbf{x}\_n$ drastically vary in the initial period, potentially deviating from the target distribution $\boldsymbol{\\mu}$ and introducing huge bias to the SA iterates $\\theta\_n$. As $a$ increases to $0.8$, we note the fastest convergence in case (i) in our simulation setup. However, further increases in $a$ (beyond $0.8$) gradually slow down the convergence since such choice of $a$ would get closer to cases (ii) and (iii). This empirical observation suggests that in practice, as $a$ increases in the range $(0.5,b]$, the convergence becomes faster but then decelerates beyond a certain point close to $b$ in the initial period. This necessitates careful fine-tuning of the step size in case (i) to achieve optimal performance.

---

### Author Response · Authors · 2023-11-21
**Reminder to All Reviewers**

Dear Reviewers,

We would like to remind you that the rebuttal period will close in one day. During this phase, we have responded to the reviews and comments/concerns you have provided. Your engagement is crucial in this part of the process. Your insights and expertise are invaluable to ensuring the improvement of our paper. We greatly appreciate your commitment and time dedicated to this important phase.

Thank you once again for your valuable reviews.

Best regards,

Submission #2003 Authors

---

### Meta-Review · Area_Chair_mKr3 · 2023-12-06

**Metareview:**

The paper describes an interesting and surprising phenomenon that appears in the context of an important problem. The consensus of the reviewers is that it should be accepted.

**Justification For Why Not Higher Score:**

N/A

**Justification For Why Not Lower Score:**

The contribution is nice and the phenomenon described (acceleration of distributed optimization with repelling random walks) is something the broader ML community should be aware of.

---

### Decision · Program_Chairs · 2024-01-16

Accept (oral)